# Atomic reconstruction for realizing stable solar-driven reversible hydrogen storage of magnesium hydride

Xiaoyue Zhang[1], Shunlong Ju [ORCID][1], Chaoqun Li[1], Jiazheng Hao[2,3], Yahui Sun[1], Xuechun Hu[1], Wei Chen[1], Jie Chen[2,3], Lunhua He [ORCID][2,4,5], Guanglin Xia [ORCID][1] ✉, Fang Fang [ORCID][1] ✉, Dalin Sun[1] & Xuebin Yu [ORCID][1] ✉

Reversible solid-state hydrogen storage of magnesium hydride, traditionally driven by external heating, is constrained by massive energy input and low systematic energy density. Herein, a single phase of $Mg_2Ni(Cu)$ alloy is designed via atomic reconstruction to achieve the ideal integration of photothermal and catalytic effects for stable solar-driven hydrogen storage of $MgH_2$. With the intra/inter-band transitions of $Mg_2Ni(Cu)$ and its hydrogenated state, over 85% absorption in the entire spectrum is achieved, resulting in the temperature up to 261.8 °C under 2.6 W cm$^{-2}$. Moreover, the hydrogen storage reaction of $Mg_2Ni(Cu)$ is thermodynamically and kinetically favored, and the imbalanced distribution of the light-induced hot electrons within CuNi and $Mg_2Ni(Cu)$ facilitates the weakening of Mg-H bonds of $MgH_2$, enhancing the "hydrogen pump" effect of $Mg_2Ni(Cu)/Mg_2Ni(Cu)H_4$. The reversible generation of $Mg_2Ni(Cu)$ upon repeated dehydrogenation process enables the continuous integration of photothermal and catalytic roles stably, ensuring the direct action of localized heat on the catalytic sites without any heat loss, thereby achieving a 6.1 wt.% $H_2$ reversible capacity with 95% retention under 3.5 W cm$^{-2}$.

Hydrogen is a potential clean energy carrier for developing a carbon-neutral system and of great significance for realizing a global transition to a sustainable energy economy. Unfortunately, the absence of a safe and efficient hydrogen storage method remains a major bottleneck for the extensive utilization of hydrogen energy[1–4]. In comparison with hydrogen storage in pressurized tanks or cryogenic containers, the reversible storage of hydrogen into solid-state light-weight metal hydrides holds the intrinsic advantages of high safety, low-cost, and exceptional hydrogen storage capacity[5–8]. Among them, magnesium hydride ($MgH_2$) that has a gravimetric and volumetric hydrogen density of 7.6 wt.% and 110 kg m$^{-3}$, respectively, and excellent reversibility, is particularly favored[9–12]. Large energy input, however, is required to drive hydrogen storage reaction of metal hydrides due to their high thermodynamic stability and activation energy[13–15]. Particularly, a theoretical operating temperature of ~280 °C should be supplied to realize reversible hydrogen storage of $MgH_2$ and the extra high kinetic barrier would lead to even higher operating temperature of 400 °C[16,17]. As a result, the reversible hydrogen storage of $MgH_2$ mainly relies on external energy input by intricate electric heating apparatus that would unavoidably not only decrease the hydrogen storage capacity of the whole system but also induce high energy consumption and costs for their practical large-scale applications.

In order to solve this critical issue, we have recently proposed the concept of solar-driven reversible hydrogen storage of metal hydrides

[1]Department of Materials Science, Fudan University, Shanghai, China. [2]Spallation Neutron Source Science Center, Dongguan, China. [3]Institute of High Energy Physics, Chinese Academy of Sciences, Beijing, China. [4]Beijing National Laboratory for Condensed Matter Physics, Institute of Physics, Chinese Academy of Sciences, Beijing, PR China. [5]Songshan Lake Materials Laboratory, Dongguan, PR China. ✉e-mail: xiaguanglin@fudan.edu.cn; f_fang@fudan.edu.cn; yuxuebin@fudan.edu.cn

by using solar energy as the sustainable and unlimited energy source[18,19]. It is realized by coupling the photothermal effect of light adsorber (i.e., MXene and Cu) that could generate heat for elevating the temperature of $MgH_2$ and catalytic effect of thermochemical catalysts (i.e., $Ti/TiH_x$) that could decrease the operating temperature of $MgH_2$ by reducing kinetic barriers. Undoubtedly, the match of the photothermal effect and the catalytic effect is a prerequisite for realizing solar-driven reversible hydrogen storage of $MgH_2$. The light adsorber and thermal catalysts are physically separated and the phase separation of thermochemical catalysts, light adsorber, and $MgH_2$, however, inevitably leads to the delay of the transfer of thus-generated heat from photothermal materials towards catalytic sites and the loss of heat, which results in poor catalytic activity and light-to-heat conversion efficiency. Moreover, both the photothermal effect of MXene and Cu and the catalytic effect of $Ti/TiH_x$ is limited and hence an extremely high light intensity of $4.0 \, W \, cm^{-2}$ is required for driving reversible hydrogen storage reaction of $MgH_2$.

Theoretically, it is an ideal solution to enhance solar-driven hydrogen storage performance of $MgH_2$ by introducing a single-component phase that simultaneously holds photothermal and catalytic effects. Herein, an in situ alloying reaction between $MgH_2$ and CuNi is developed to achieve the integration of the photothermal effect with full-spectrum light absorption and the catalytic effect that could significantly lower the apparent activation energy of hydrogen storage reaction. Under UV–vis irradiation, the strong localized surface plasmon resonance (LSPR) effect of CuNi alloys would generate localized heat at the catalytic sites of CuNi alloys, leading to superior initial $H_2$ desorption performance than that driven by traditional heating. Together with the initial dehydrogenation process, the in situ alloying reaction between Mg and CuNi results in simultaneous atomic reconstruction, leading to the formation of $Mg_2Ni(Cu)$ ternary alloy that has the metallic properties and the combined effect of intra/inter-band transitions. Therefore, an over 85% absorption in the entire spectrum could be achieved for $MgH_2$ catalyzed by CuNi after the initial dehydrogenation process, which, due to the promoted light-to-heat conversion efficiency, results in a high surface temperature of $261.8 \, °C$ under $2.6 \, W \, cm^{-2}$, $15 \, °C$ higher than that before cycling under identical condition. On the other hand, in comparison with $MgH_2$, the hydrogen storage reaction of $Mg_2Ni(Cu)$ is thermodynamically and kinetically favored resulting from the weakening of Mg–H bonds and the low migration barrier of hydrogen atoms in both $Mg_2Ni(Cu)$ and $Mg_2Ni(Cu)H_4$, which in turn provides a facile pathway for the spontaneous breaking of Mg–H bonds of $MgH_2$. More importantly, attributed to the smaller work function of Cu than that of Ni and the larger work function of Cu than that of Mg, the hot electrons generated by the LSPR effect of Cu could be transferred to Ni inside of both CuNi and $Mg_2Ni(Cu)$ alloys, leading to an increase in electron density around Ni during light irradiation. The uneven distribution of light-induced hot electrons within CuNi and $Mg_2Ni(Cu)$ alloy contributes to the effective weakening of Mg–H bonds of $MgH_2$ and hence enhances the "hydrogen pump" effect of $Mg_2Ni(Cu)/Mg_2Ni(Cu)H_4$, which improves cycling hydrogenation and dehydrogenation of $MgH_2$ and significantly reduces the apparent activation energy required for driving hydrogen storage in $MgH_2$. As a result, the in situ atomic reconstruction of $Mg_2Ni(Cu)$ enables the integration of photothermal and catalytic roles in a single-component phase, which allows the direct action of localized photothermal heat on the catalytic sites without any heat loss, resulting in the complete dehydrogenation of $MgH_2$ with a reversible capacity of 6.1 wt.% within 15 min under $3.5 \, W \, cm^{-2}$. More importantly, the stable reversible generation of $Mg_2Ni(Cu)$ during cycling hydrogen storage process results in well-preserved photothermal and catalytic effects in improving repeated hydrogenation and dehydrogenation of $MgH_2$, delivering a reversible $H_2$ storage capacity of 6.1 wt.% with a capacity retention of 95% after ten cycles using solar energy.

## Results

### Preparation and characterization of CuNi alloys

CuNi bimetallic alloys are first synthesized via a one-step solvothermal calcination method as shown in Supplementary Fig. 1. Nitrate salts containing Cu and Ni ions are added and evenly dispersed in a mixed aqueous solution of glycerol and propanediol, which is then hydrothermally treated at $150 \, °C$ for 3 h. After washing and thermal reduction, CuNi alloys encapsulated in carbon layers could be obtained. By changing the appropriate amount of the precursors, the ratio of Cu and Ni atoms in the thus-obtained alloy could be facile controlled, resulting in the synthesis of $Cu_1Ni_x$ (Cu: Ni atomic ratio of 1:$x$), Cu, and Ni, respectively. Inductively coupled plasma-optical emission spectroscopy (ICP-OES) indicates that the atomic ratios of $Cu_1Ni_1$, $Cu_1Ni_2$, and $Cu_1Ni_3$ are 1:1.03, 1:2.01, and 1:3.01 (Supplementary Table 1), respectively, which demonstrates the successful synthesis of CuNi alloys with controllable atomic ratios by facile adjusting the ratio of precursors.

According to X-ray diffraction (XRD) results (Fig. 1a), all the as-synthesized catalysts exhibit three distinct diffraction peaks, indicating the formation of typical face-centered-cubic (FCC) crystal phase with (111), (200), and (220) planes. The diffraction peaks of CuNi alloys show no splitting and only slight shifts in the region between the diffraction peaks of Cu (PDF#00-004-0836) and Ni (PDF#04-003-6597) could be observed for CuNi alloys, which is attributed to the slight change of crystal plane spacing caused by various Cu/Ni molar ratio[20]. Among them, the diffraction peaks of $Cu_1Ni_1$ are located at 43.86°, 51.09°, and 75.17°, respectively, matching the standard diffraction peaks of $Cu_{0.5}Ni_{0.5}$ (PDF#04-002-1345) ideally, thereby confirming the successful synthesis of $Cu_1Ni_1$ alloys. Moreover, the presence of Cu and Ni in CuNi alloys is confirmed in X-ray photoelectron spectroscopy (XPS) survey spectra (Supplementary Fig. 2a). High-resolution Cu $2p$ and Ni $2p$ XPS spectra (Fig. 1b, c and Supplementary Fig. 2c, d) demonstrate that Cu and Ni are present mainly in their zero-valent state, consistent with the XRD results. Particularly, the binding energies of $Cu^0$ and $Ni^0$ in CuNi alloys exhibit shifts compared to pure Cu and Ni due to the electronegativity difference, indicating the electronic interactions inside of CuNi alloys[21,22]. The weak peaks of $Cu^{2+}$ and $Ni^{2+}$ could be attributed to the slight oxidation of CuNi alloys during measurement (Supplementary Table 2) and no chemical interaction between the outer carbon layer and CuNi alloys could be observed in the high-resolution C $1s$ XPS spectra (Supplementary Fig. 2b).

Scanning electron microscopy (SEM) image (Fig. 1d) illustrates that $Cu_1Ni_1$ is composed of densely packed spherical particles with an average particle size of approximately 20 nm (Fig. 1g). High-resolution transmission electron microscopy (HRTEM) images (Fig. 1e and Supplementary Fig. 3a–d) confirm that $Cu_1Ni_1$ alloys are wrapped by a carbon layer of 3 nm with a weight percent of approximately 9% as evidenced by thermogravimetric analysis (TGA) (Supplementary Fig. 5a). The lattice spacing inside of $Cu_1Ni_1$ is calculated to be 0.206 nm, corresponding to the (111) crystal plane of $Cu_1Ni_1$. The selected area electron diffraction (SAED) pattern in Fig. 1f confirms the polycrystal structure of $Cu_1Ni_1$ alloys and the diffraction rings could be well recorded to (111), (200), and (220) crystal planes. In addition, the lattice spacing measured from HRTEM images (Supplementary Fig. 4a–d) presents a lattice spacing (0.206 nm) of $Cu_1Ni_1$ (111) that is between the lattice spacing (0.210 nm) of Cu (111) and the lattice spacing (0.203 nm) of Ni (111) with a periodic pattern, implying the successful synthesis of $Cu_1Ni_1$ alloys wrapped with ultrathin carbon layers. Furthermore, energy dispersive X-ray spectroscopy (EDS) elemental mapping and line scanning of two adjacent particles (Fig. 1h–j and Supplementary Fig. 3e–j) results confirm the homogeneous distribution of Cu and Ni, which coincides well with the morphology of particles, further validating the uniform synthesis of CuNi alloys. In addition, the morphology, particle size (Supplementary Fig. 6a–f), and carbon content (Supplementary Fig. 5b, c) of $Cu_1Ni_2$, $Cu_1Ni_3$, and Ni are

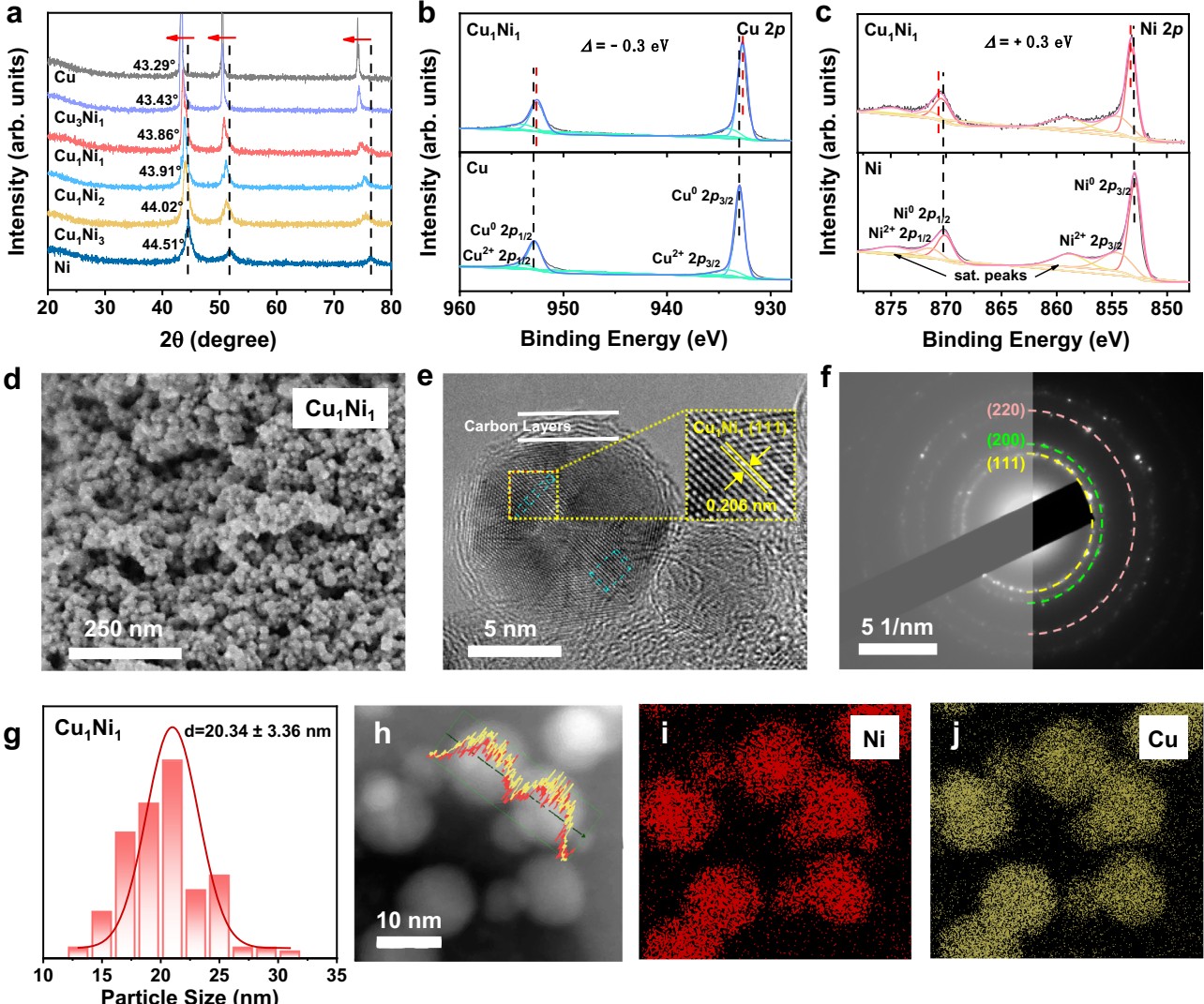

**Fig. 1 | Characterization of the properties and structures of CuNi alloys. a** XRD patterns of the as-synthesized of CuNi alloys, including Ni and Cu for comparison. The red arrows represent the direction of characteristic peak shifts with increasing Cu content. High-resolution **b** Cu *2p* and **c** Ni *2p* XPS spectra of $Cu_1Ni_1$, with Cu and Ni for comparison. **d** SEM and **e** HRTEM images with the measured lattice spacing inset of $Cu_1Ni_1$. **f** SAED pattern of $Cu_1Ni_1$ with the raw pattern in lightened region on the left side for comparison. **g** the corresponding particle size distribution of $Cu_1Ni_1$ alloys with Gaussian fitting. **h**–**j** EDS elemental mapping images with line scanning of $Cu_1Ni_1$ alloys. Source data are provided as a source data file.

similar to those of $Cu_1Ni_1$, while the particle size of Cu is approximately 50 nm (Supplementary Fig. 6g, h).

**Solar-driven hydrogen storage performance of MgH₂**

The hydrogen storage performance using conventional electrical heating method demonstrates that, in comparison to $MgH_2$ catalyzed by Ni nanoparticles, $MgH_2$ under the catalysis of CuNi alloys exhibits superior hydrogen storage performance of further decreased operating temperature with lower kinetic barriers and enhanced kinetics of $H_2$ desorption and absorption. The specific experimental results with detailed analysis could be found in the Supplementary Information (Supplementary Figs. 7–10). Attributed to the decrease of $H_2$ desorption activation energy to 80 kJ mol⁻¹, the peak temperature of $H_2$ desorption from $MgH_2$ catalyzed by CuNi is reduced down to approximately 245 °C, and complete $H_2$ desorption could be achieved within 30 min at 250 °C. Moreover, under a hydrogen pressure of 1 MPa at 250 °C, 6.1 wt.% $H_2$ could be recharged within 10 min, and 95% reversible capacity is maintained after ten cycles of $H_2$ absorption and desorption. Although the activation energy of heat-driven reversible $H_2$ storage of $MgH_2$ catalyzed by CuNi has been lowered tremendously, it still requires external thermal energy input towards realizing the operating temperature of over 200 °C. Therefore, attempting to use inexhaustible solar energy to drive $H_2$ absorption and desorption of CuNi-catalyzed $MgH_2$ is a highly promising and environmentally-friendly alternative. To investigate the solar-driven $H_2$ storage performance of $MgH_2$, solar energy serves as the only energy input source without external heating, which aligns with our goal of completely replacing the electric heating with solar energy, via the apparatus shown in Supplementary Fig. 11.

After the ball-milling process of $MgH_2$ with the addition of catalysts with a weight percent of 10%, an obvious color difference could be observed (Fig. 2a). Pure $MgH_2$ is essentially white and it turns into yellowish-brown, and becomes darker and finally dark-black after the addition of Ni, CuNi alloys with the increasing Cu content, and Cu, respectively, indicating the increase of light absorption potential. It could be directly supported by the ultraviolet–visible–near-infrared (UV–vis–NIR) absorption spectra (Fig. 2b), which demonstrates that $MgH_2$ catalyzed by Cu has the highest overall absorbance capability and the absorbance of $MgH_2$ after the addition of CuNi alloys increases as the relative Cu content in these catalysts increases, where $Cu_1Ni_2$

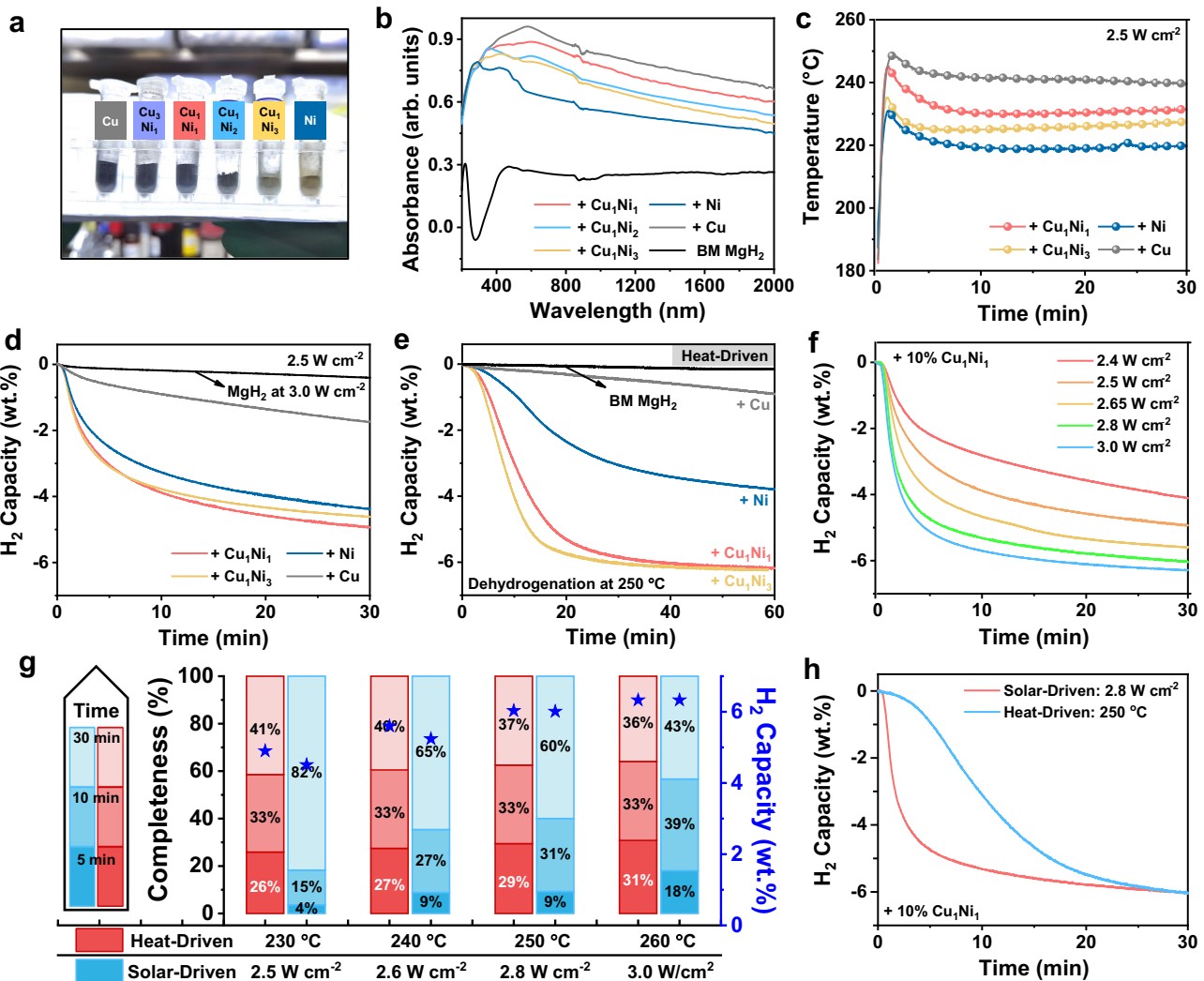

**Fig. 2 | The initial solar-driven dehydrogenation performance of MgH₂ catalyzed by CuNi alloy. a** The color, **b** UV–vis–NIR absorption spectra, **c** the response of temperature to light irradiation, **d** the corresponding solar-driven H₂ desorption curves of MgH₂, and **e** the heat-driven isothermal dehydrogenation curves at 250 °C of MgH₂ under the catalysis of CuNi alloys, Ni, and Cu, respectively, including ball-milled MgH₂ (BM MgH₂) for comparison. **f** H₂ desorption curves of MgH₂ catalyzed by Cu₁Ni₁ under different light intensities. **g** Completeness ratio over time using solar energy and direct heating with normalized H₂ release capacity of MgH₂ under the catalysis of Cu₁Ni₁ at varying light intensities (or corresponding temperatures). The pentagrams represent H₂ capacity. **h** H₂ desorption curves of MgH₂ catalyzed by Cu₁Ni₁ using solar energy (light intensity: 2.8 W cm⁻², corresponding to a temperature of 250 °C) and electrical heating (250 °C), respectively. Source data are provided as a source data file.

falls between those of Cu₁Ni₃ and Cu₁Ni₁. Notably, MgH₂ under the catalysis of Cu exhibits a characteristic absorption peak from 400 to 800 nm due to the LSPR effect of Cu nanoparticles (Supplementary Fig. 12a)[23]. By comparison, a slight red-shift of the LSPR characteristic peak of Cu is observed for MgH₂ catalyzed by Cu₁Ni₁ owing to the size changes of nanocrystals[24]. The FDTD simulations further demonstrate that plasmonic "hot spot" regions could be clearly observed around particles in both Cu and CuNi alloys, which validates the strong LSPR-induced localized electric field of Cu and CuNi (Supplementary Fig. 12e–g). In addition, the relatively weaker absorption observed in the NIR region also increases with higher Cu content, which could be attributed to the scattering effect induced by the LSPR coupling of Cu particles (Fig. 2b and Supplementary Fig. 12b)[25–27].

Herein, with the consideration of maximizing the differences brought by photothermal effects under similar catalytic performance, Cu₁Ni₁ and Cu₁Ni₃ are selected as representatives in the following. Correspondingly, under the same intensity (2.5 W cm⁻² and 3.0 W cm⁻²) of solar irradiation, the surface temperature of MgH₂ also increases with the addition of catalysts that has more relative content of Cu

(Fig. 2c and Supplementary Fig. 12a). These results validate that, due to the strong LSPR effect of metal nanoparticles of Cu and Ni, the introduction of CuNi alloys endows MgH₂ with overall stronger light absorption and hence better photothermal effect under solar irradiation.

As expected, induced by this photothermal effect, the surface temperature of MgH₂ catalyzed by Cu₁Ni₁ reaches 230.1 °C within 10 min under a solar irradiation intensity of 2.5 W cm⁻², which could drive the H₂ desorption of MgH₂ catalyzed by Cu₁Ni₁ (Fig. 2c), achieving a rapid H₂ desorption of 4.9 wt.% within 30 min (Fig. 2d). In comparison to Cu₁Ni₁ alloys, MgH₂ catalyzed by Cu₁Ni₃ exhibits the same significant decrease in dehydrogenation temperature and faster kinetics in traditional heat-driven experiments compared to MgH₂ catalyzed by Cu₁Ni₁ (Fig. 2e and Supplementary Fig. 7a). However, under 2.5 W cm⁻² illumination, MgH₂ catalyzed by Cu₁Ni₃ only reaches a surface temperature of 224.8 °C, corresponding to a H₂ desorption capacity of 4.6 wt.%, which is even 0.3 wt.% lower than that of MgH₂ catalyzed by Cu₁Ni₁ (Fig. 2d). In addition, the difference of the dehydrogenation kinetics of MgH₂ catalyzed by Cu₁Ni₁ and Cu₁Ni₃ becomes

almost negligible under the solar irradiation of 2.5 W cm$^{-2}$ and 3.0 W cm$^{-2}$ (Fig. 2d and Supplementary Fig. 13b). This result suggests that the higher content of Cu in CuNi alloys, which brings about a stronger photothermal effect, leads to further improvement of H$_2$ desorption performance under solar irradiation compared to those driven by direct heating, which provides direct evidence to the crucial role of photothermal effect in enhancing solar-driven hydrogen storage of MgH$_2$.

Upon the elevation of the irradiation intensity, the surface temperature of MgH$_2$ catalyzed by Cu$_1$Ni$_1$ would increase, which is capable of promoting the degree and rate of H$_2$ desorption from MgH$_2$ (Fig. 2f and Supplementary Fig. 14a, b). The dehydrogenation instantaneous rate peaks within the initial 100 seconds under solar irradiation, and the relationship between initial H$_2$ release rates and light intensities is exponential (Supplementary Fig. 14c, d), consistent with the Arrhenius empirical equation[28,29], which suggests that the photothermal effect primarily operates as a plasma heating. Comparing the amount of H$_2$ desorption at various temperatures and light intensities (Supplementary Fig. 14e), the H$_2$ desorption performance under solar energy is similar to that driven by thermal heating, that is, H$_2$ release from MgH$_2$ through a photothermal pathway induced by the light-to-heat conversion ability of CuNi alloys rather than photocatalytic principle[30]. In addition, the apparent activation energy under solar irradiation fitted by the Johnson–Mehl–Avrami (JMA) equation joint with the Arrhenius equation is calculated to be 81.23 ± 3.35 kJ mol$^{-1}$, which is comparable to that driven by thermal heating (i.e., 81.56 ± 2.00 kJ mol$^{-1}$) (Supplementary Figs. 14f and 15h). These results validate that the solar-driven dehydrogenation of MgH$_2$ is mainly achieved based on the localized heat produced by photothermal effect of CuNi alloys with the combination of its catalytic effect.

To further highlight the advantages of solar-driven dehydrogenation over heat-driven dehydrogenation, the normalized H$_2$ desorption capacities at various light intensities (or corresponding surface temperatures) are calculated to intuitively demonstrate the differences in H$_2$ desorption ratio over time. MgH$_2$ catalyzed by Cu$_1$Ni$_1$ could release 60% of H$_2$ within 10 min under 2.5 W cm$^{-2}$ irradiation (Fig. 2g), while only 20% of H$_2$ is released from MgH$_2$ in 10 min under electrical heating. Particularly, the superiority of fast response and rate of H$_2$ desorption under solar irradiation over direct heating could be consistently observed for MgH$_2$ in the range of 2.5 W cm$^{-2}$ to 3.0 W cm$^{-2}$ (Fig. 2g and Supplementary Fig. 15g). In strong contrast, H$_2$ desorption kinetics of MgH$_2$ using direct heating strongly depends on adopted temperatures induced by the low thermal conductivity of MgH$_2$ (<1 W m$^{-1}$ K$^{-1}$)[31]. The advantages of solar-driven dehydrogenation over heat-driven H$_2$ desorption in realizing fast dehydrogenation are also measured and evaluated using the glass tube with contact-type thermocouple method (Supplementary Figs. 11c, d and 15a–f), and demonstrated in MgH$_2$ catalyzed by Cu$_1$Ni$_3$, Ni, and Cu, respectively (Supplementary Fig. 16a–f).

Specifically, when the light intensity is increased to 3.0 W cm$^{-2}$, the surface temperature of MgH$_2$ catalyzed by Cu$_1$Ni$_1$ and Cu$_1$Ni$_3$ could reach 251.6 °C and 248.9 °C, respectively, delivering a H$_2$ desorption capacity of 6.3 wt.% and 6.1 wt.% H$_2$, respectively, corresponding to almost complete dehydrogenation of MgH$_2$ (Supplementary Fig. 13a, b). In contrast, no H$_2$ desorption could be observed from the ball-milled MgH$_2$ without CuNi alloys under the identical condition, which directly demonstrates the vital role of CuNi alloys that act simultaneously as light adsorber and thermochemical catalysts in enabling solar-driven H$_2$ desorption from MgH$_2$. By comparison, although the addition of Cu to MgH$_2$ could obtain a higher surface temperature of up to 260 °C under 3.0 W cm$^{-2}$ light intensity induced by the strong photothermal effect of Cu nanoparticles, only a capacity of 4.6 wt.% could be achieved at 3.0 W cm$^{-2}$ due to the negligible catalytic effect of Cu in enhancing H$_2$ desorption performance of MgH$_2$ (Fig. 2c and Supplementary Fig. 7). Nonetheless, it is noteworthy that

the H$_2$ desorption capacity of MgH$_2$ catalyzed by Cu at 3.0 W cm$^{-2}$ is significantly higher than the 0.7 wt.% H$_2$ capacity under the heat-driven temperature of 260 °C (Supplementary Fig. 16e). This result further highlights the significant role of LSPR effect of metal nanoparticles in producing intense localized heat for driving H$_2$ desorption of MgH$_2$ under solar illumination. CuNi alloys, evenly distributed on the MgH$_2$ matrix, could generate localized heat under solar irradiation that could directly act on the catalytic center that is itself while the poor thermal conductivity of "MgH$_2$" could reduce the temperature gradient near CuNi alloys, which is capable of promoting rapid H$_2$ desorption from MgH$_2$[18]. Therefore, MgH$_2$ catalyzed by Cu$_1$Ni$_1$ releases 5.3 wt.% H$_2$ within 10 min under 2.8 W cm$^{-2}$ irradiation, whereas the time to achieve identical H$_2$ desorption capacity using external heating is doubled (Fig. 2h). Furthermore, upon increasing the loading ratio of Cu$_1$Ni$_1$ to 20%, the temperature of MgH$_2$ could be elevated to 238.1 °C, 7.7 °C higher than that with a loading ratio of Cu$_1$Ni$_1$ of 10% (Supplementary Fig. 17a). The extent of dehydrogenation increases to 93.05% under 2.5 W cm$^{-2}$, compared with 83.28% under the addition of 10% Cu$_1$Ni$_1$, together with a faster dehydrogenation rate (Supplementary Fig. 17b, c), which could be attributed to the enhanced photo-to-thermal effect combined with the catalytic effect in further reducing the dehydrogenation temperature (Supplementary Fig. 17d). This further validates that, with the increase ratio of CuNi alloys, the simultaneous enhancement of photothermal and catalytic effects of CuNi alloys would mutually contribute to the better solar-driven performance.

Subsequently, the reversible H$_2$ storage performance of MgH$_2$ under the catalysis of CuNi is investigated, which is a critical challenge in practical applications of solid-state hydrogen storage materials. Interestingly, after the initial hydrogenation process, the color of MgH$_2$ catalyzed with Cu$_1$Ni$_1$, Cu$_1$Ni$_2$, and Cu$_1$Ni$_3$ turns deep black, similar to that with Cu, while MgH$_2$ catalyzed by Ni remains yellowish-brown (Fig. 3a, inset). Surprisingly, MgH$_2$ catalyzed by CuNi after the initial hydrogenation process not only maintains strong absorption in the ultraviolet–visible range, like that after ball-milling, but also exhibits the significant increase of overall absorbance in the near-infrared region, with a relative absorbance from ~0.5 to ~0.9 (absorption ratio from ~70 to ~86%) at a wavelength of 2000 nm (Fig. 3b). As a result, the overall absorbance of MgH$_2$ catalyzed by CuNi after cycling even exceeds that of MgH$_2$ catalyzed by Cu (Fig. 3a). In comparison, the overall increase in absorbance of MgH$_2$ catalyzed by Cu and Ni after cycling is limited compared to that of CuNi (Supplementary Fig. 18). This enables MgH$_2$ catalyzed by CuNi to utilize the near-infrared irradiation (from 800 to 2000 nm) that accounts for 52% of the solar spectrum and the enhanced full-spectrum (from 250 to 2000 nm) absorption capability after cycling is achieved[32], which is critical for enhancing solar energy utilization efficiency and hence light-to-heat performance.

Owing to the increased overall absorption in the near-infrared range, the surface temperature of cycled MgH$_2$ catalyzed by Cu$_1$Ni$_1$ is higher than that of the ball-milled state under light irradiation (Fig. 3c). Significantly, under the irradiation intensity of 2.6 W cm$^{-2}$, the surface temperature of MgH$_2$ catalyzed by Cu$_1$Ni$_1$ after cycling is increased to 261.8 °C, which is 6.1% higher than that after ball-milling process, correspondingly leading to the enhancement of H$_2$ desorption rate of the cycled MgH$_2$. The ball-milled MgH$_2$ catalyzed by Cu$_1$Ni$_1$ releases 5.39 wt.% H$_2$ within 30 min, while the time required to release the same H$_2$ capacity could be shortened to 25 min with more H$_2$ capacity for MgH$_2$ catalyzed by Cu$_1$Ni$_1$ after cycling (Fig. 3d and Supplementary Fig. 19f). Similar phenomena of the increase of the surface temperature (with a 3.7% increase) and the acceleration of H$_2$ desorption could also be observed for MgH$_2$ under the catalysis of Cu$_1$Ni$_2$ and Cu$_1$Ni$_3$ after cycling under the identical light intensity (Supplementary Fig. 19a, b). In contrast, the elevation of the surface temperature of MgH$_2$ catalyzed by Cu or Ni after cycling is only around 2% (Supplementary Fig. 19c, d).

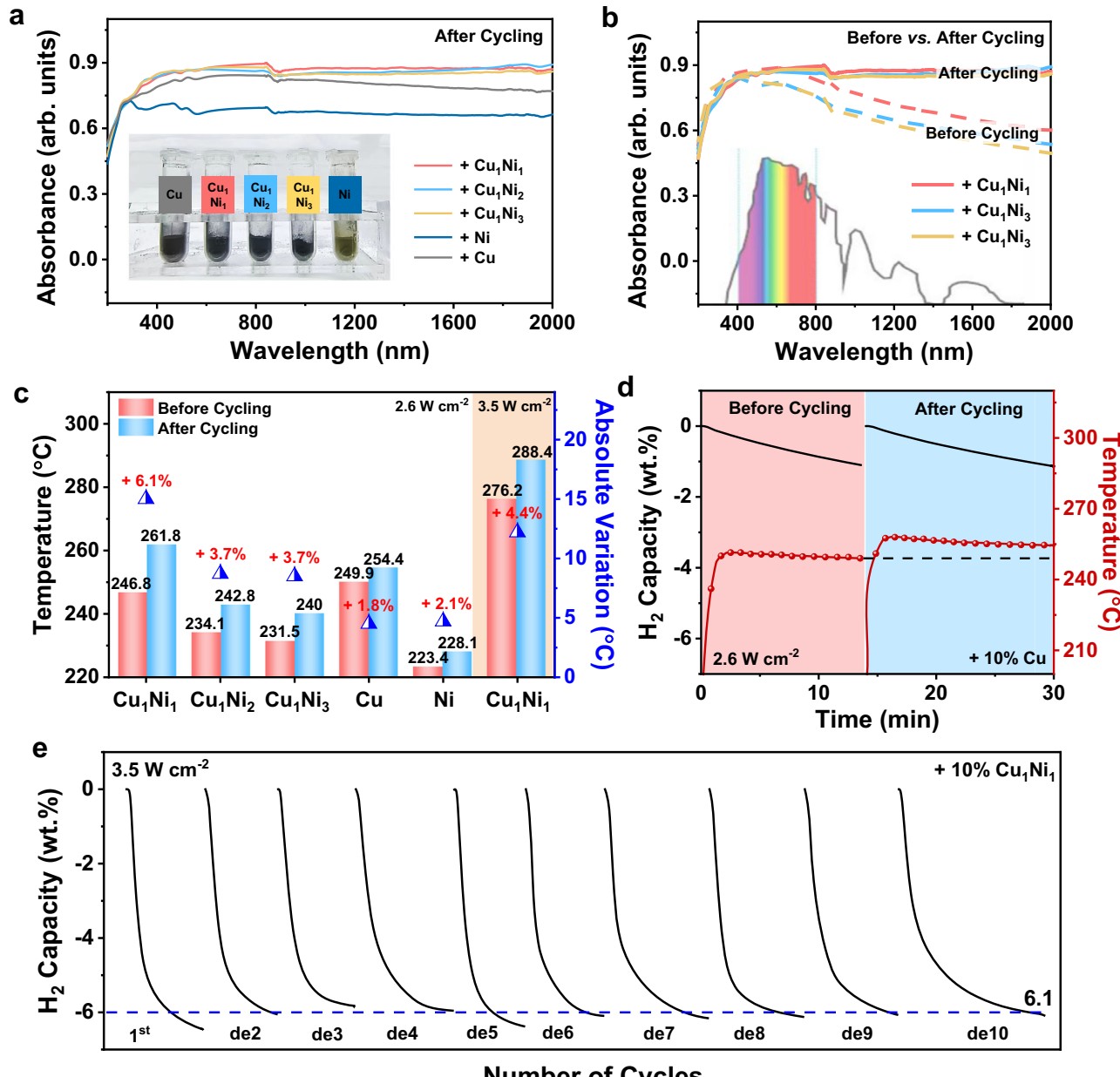

**Fig. 3 | The reversible solar-driven H₂ storage performance of MgH₂ catalyzed by CuNi alloys. a** UV–vis–NIR absorption spectra of MgH₂ under the catalysis of Cu₁Ni₁, Cu₁Ni₂, Cu₁Ni₃, Ni, and Cu after rehydrogenation. The inset shows the color of their cycled states. **b** UV–vis–NIR absorption spectra of MgH₂ under the catalysis of Cu₁Ni₁, Cu₁Ni₂, and Cu₁Ni₃ after cycling, in comparison with their pristine ball-milled states (before cycling). The inset exhibits solar irradiation distribution of natural sunlight. **c** The signal data of temperature to light irradiation of MgH₂ after cycling under the catalysis of Cu₁Ni₁, Cu₁Ni₂, Cu₁Ni₃, Ni, and Cu, respectively, with their pristine ball-milled states (before cycling) included for comparison. The shaded tan area corresponds to the values under 3.5 W cm⁻² and the half-filled triangles represent to the absolute temperature variation. **d** The response of temperature with corresponding H₂ desorption curves of MgH₂ under the catalysis of Cu₁Ni₁ under 2.6 W cm⁻² before and after cycling. **e** Cycling H₂ desorption curves of MgH₂ under the catalysis of Cu₁Ni₁ using a light intensity of 3.5 W cm⁻². The de1 to de10 represent the 1st to the 10nd dehydrogenation of cycles. Source data are provided as a source data file.

These results validate the significant enhancement of photothermal performance for MgH₂ with the catalysis of CuNi after cycling, which could further promote the hydrogen storage kinetics of MgH₂.

Under the light density of 3.5 W cm⁻², the operating temperatures for H₂ desorption and absorption of MgH₂ are determined to be 277 °C and 265 °C, respectively, in the first cycle, and the corresponding temperatures reach 288 °C and 270 °C in the second cycle (Supplementary Fig. 20a, b). Impressively, a reversible capacity of 6.1 wt.% could be achieved for MgH₂ catalyzed by Cu₁Ni₁ after ten cycles of H₂ adsorption and desorption, corresponding to a capacity retention of 95% (Fig. 3e). This result fits well with the stable cycling performance of

MgH₂ under the catalysis of Cu₁Ni₁ upon direct heating (Supplementary Fig. 10a, b), which indirectly demonstrates the excellent stability of the catalytic effect under solar irradiation. On the other hand, it could be clearly observed that nearly no change of the absorbance performance could be observed for MgH₂ under the catalysis of Cu₁Ni₁ after ten cycles compared with that after the initial hydrogenation process (Supplementary Fig. 21a), corresponding to the stable enhancement of surface temperature after cycling (Supplementary Fig. 21b), indicating the well-preserved photothermal effect, which is indispensable to realize stable solar-driven reversible hydrogen storage of MgH₂. These results indicate that, the stable full-spectrum light

absorption capability after cycling, combined with the excellent stability of the catalytic effect, enables MgH$_2$ catalyzed by CuNi to achieve stable hydrogen storage reversibility driven by solar energy.

Although the relatively high light intensity is required to drive the reversible hydrogen storage of MgH$_2$ even under the catalysis of Cu$_1$Ni$_1$ alloys, the low-intensity light ( ~ 0.5 W cm$^{-2}$) could be concentrated to over 3.5 W cm$^{-2}$ via Fresnel lens (Supplementary Fig. 22a), which makes it possible to drive reversible hydrogen storage reactions by natural light. To further demonstrate the feasibility of H$_2$ desorption using natural solar energy, an outdoor experiment of solar-driven H$_2$ desorption of MgH$_2$ under the catalysis of Cu$_1$Ni$_1$ alloys are performed via a low-cost PMMA Fresnel lens. A striking emergence of hydrogen bubbles from the pipe could be observed (Supplementary Fig. 22b and Supplementary Movie 1), and the predominant phase in the XRD pattern transforms into pure Mg following H$_2$ desorption process (Supplementary Fig. 22c), which offers direct evidence to the potential for driving solid-state hydrogen storage of MgH$_2$ by natural solar energy.

## Mechanism of photothermal and catalytic effects

To disclose the evolution of the photothermal and catalytic effects of CuNi, the phase and morphology change of MgH$_2$ catalyzed by CuNi before and after cycling is investigated. Direct observation of the characteristic peaks other than MgH$_2$ in XRD is challenging due to the amorphization of the samples and/or the ultralow content of CuNi (Supplementary Fig. 23b)[33]. Cu$_1$Ni$_1$ alloys with the characteristic lattice spacing of Cu$_1$Ni$_1$, however, could be found in HRTEM (Supplementary Fig. 24) with uniform distribution of Mg, Ni, Cu, and C elements, indicating that Cu$_1$Ni$_1$ alloys still existsin the original state and are evenly embedded inside of MgH$_2$ after ball-milling. The initial dehydrogenation of MgH$_2$ under the catalysis of Ni and Cu leads to the formation of Mg$_2$Ni (PDF#00-035-1225) and Mg$_2$Cu (PDF#04-008-6025), respectively. By comparison, the characteristic peaks of Mg$_2$Ni(Cu) between Mg$_2$Ni and Mg$_2$Cu could be clearly observed with the presence of a weak peak at around 38-39°, which provides additional evidence for the formation of Mg$_2$Ni(Cu) during the initial dehydrogenation of MgH$_2$ catalyzed by Cu$_1$Ni$_1$ (Supplementary Figs. 23d, e and 25a, b). The refined structure of powder neutron diffraction (PND) of MgH$_2$ catalyzed by CuNi after total dehydrogenation also confirms the presence of the solid solution phase Mg$_2$Ni$_{1-x}$(Cu)$_x$, where Cu enters Ni1(0.5,0,0) and Ni2(0,0,0) sites within Mg$_2$Ni (Supplementary Fig. 26 and Supplementary Tables 4 and 5). The (200) and (203) planes of Mg$_2$Ni(Cu) are detected in HRTEM with a slight increase compared to the corresponding planes of Mg$_2$Ni (Supplementary Fig. 27). These results indicate that, upon initial dehydrogenation process, CuNi alloys would react with Mg at an atomic ratio of 2:1, resulting in the in situ formation of Mg$_2$Ni with Cu solution, i.e., Mg$_2$Ni(Cu) ternary alloy. After rehydrogenation, the characteristic XRD peaks of Mg$_2$NiH$_4$ and MgCu$_2$ are detected (Fig. 4a and Supplementary Fig. 23c), which agrees with the lattice fringes detected in TEM with the observation of uniform distribution of MgCu$_2$ and Mg$_2$NiH$_4$ within the catalyst region (Supplementary Fig. 28). In addition, during the re-dehydrogenation process, MgCu$_2$ and Mg$_2$NiH$_4$ undergo solid solution at the initial stage (less than 30% reaction progress), and a gradual low-angle shift of Mg$_2$NiH$_4$ lattice towards the characteristic peak of Mg$_2$Ni(Cu) alloy could be observed due to the solid solution of Cu, which demonstrates the presence of an intermediate state of Mg$_2$Ni(Cu)(H) during the hydrogenation process (Fig. 4b and Supplementary Fig. 25c, d). More importantly, the characteristic peaks of Mg$_2$Ni(Cu) could still be clearly detected and remain unchanged in XRD patterns even after ten cycles of H$_2$ absorption and desorption. No phase separation between Cu and Ni is observed in the EDS elemental mapping results with uniform distribution of the characteristic (200) lattice fringes of Mg$_2$Ni(Cu) within Mg matrix. These results provide direct evidence for the reversible formation of Mg$_2$Ni(Cu) upon cycling dehydrogenation, wherein structurally stable phases of

Mg$_2$NiH$_4$ and MgCu$_2$ would form after hydrogenation, and they would then rapidly transform into an intermediate state of Mg$_2$Ni(Cu)(H) during the subsequent dehydrogenation process, accompanied by gradual alloying into single phase of Mg$_2$Ni(Cu). Therefore, the reversible formation of Mg$_2$Ni(Cu) with a Cu atomic ratio of $x$ in CuNi during H$_2$ desorption/absorption could be summarized:

$$2Mg_2Ni_{1-x}Cu_x + (4-x)H_2 \leftrightarrow 2Mg_2Ni_{1-x}(Cu)_x(H)$$
$$\leftrightarrow (2-2x)Mg_2NiH_4 + 3xMgH_2 + xMgCu_2$$

A similar reversible transformation between Mg$_2$Ni(Cu), Mg$_2$NiH$_4$, and MgCu$_2$ during H$_2$ desorption/absorption is also observed in MgH$_2$ catalyzed by equimolar ratios of Cu and Ni, further validating the reversible formation of Mg$_2$Ni(Cu) (Supplementary Fig. 23f, g). In addition, a loss in H$_2$ capacity of the above system could nearly be neglected under the catalysts of CuNi alloys with a loading ratio less than 10% (Supplementary Table 6). After the initial hydrogenation process, numerous interfaces and lattice distortions are observed at the area between MgCu$_2$, Mg$_2$NiH$_4$, and MgH$_2$. In addition, the presence of embedded MgCu$_2$ and atomic disorder are even found within MgH$_2$ lattice (Supplementary Fig. 28). After ten cycles of H$_2$ desorption, more lattice distortions and dislocations exist both at the interfaces and within the alloys, and specifically, dislocation analysis at region (4) reveals that these regions exhibit a higher degree of disorder than Mg matrix (Supplementary Fig. 29). This is attributed to the solid-state diffusion of Mg required for the transition between Mg$_2$Ni(Cu) and MgCu$_2$[34], and the lattice distortions and disorders generated during H$_2$ storage reactions could reduce local thermal conductivity[35,36], which also helps to enhance the local thermal field.

To reveal the mechanism behind the enhancement of solar absorption of MgH$_2$ under the catalysis of CuNi alloys before and after cycling, the electronic density of states (DOS) is investigated. In terms of pure MgH$_2$, the valence band is mainly contributed by H($s$) (79.3%) with strong hybridization with Mg($sp$) (20.7%) and the conduction band is composed of 63.1% Mg($sp$) and 36.9% H($s$) (Supplementary Fig. 30a). Strong ionic bonding occurs between Mg$^{2+}$ and H$^-$, resulting in electrons primarily distributed around H atoms (Supplementary Fig. 31a), while the electron distribution near the Fermi level approaches zero, indicating distinct insulating properties (Supplementary Figs. 30b and 32a, b). After adding CuNi alloys into MgH$_2$, strong absorption peaks in the visible light range could be ascribed to the LSPR effect due to intra-band absorption (Supplementary Fig. 12a), which occurs by the excitation of free electrons of conduction electrons near the Fermi surface from $sp$-$d$ hybridized atomic orbitals of Cu (Supplementary Fig. 33), implying that the initial dehydrogenation process is driven by the LSPR effect of CuNi. As mentioned above, CuNi alloys would react with Mg during the dehydrogenation process, leading to the formation of Mg$_2$Ni(Cu) alloy and then the transformation of Mg$_2$NiH$_4$, MgCu$_2$, and Mg$_2$Ni(Cu)H$_4$ upon reversible hydrogenation. In comparison to the ionic bonding of MgH$_2$, covalent bonds between Ni(Cu) metal atoms and their neighbor H atoms exist in Mg$_2$NiH$_4$ and Mg$_2$Ni(Cu)H$_4$ with clear electrons sharing (Supplementary Fig. 31b, c), leading to the weakening of Mg–H bonds and uniform distribution of $s$-$p$ band. Near the Fermi level, a relatively weak distribution of electrons could be observed in Mg$_2$NiH$_4$ with semiconductor properties (Fig. 4c, d and Supplementary Fig. 32c–e), while, due to the solubility of Cu, a more flattened hybridized orbital, resembling a "flat band", appears, which is mainly Ni($d$) orbitals hybridized with weaker Mg($sp$) and H($s$) (Fig. 4e, f and Supplementary Fig. 32f–h). As a result, a continuous band structure fills the vacant Fermi level in Mg$_2$Ni(Cu)H$_4$ with metallic properties, together with MgCu$_2$ (Supplementary Fig. 30c), which induces the facile excitation of electrons under light irradiation. Attributed to the robust hybridization of Cu($d$) and Ni($d$) orbitals within the valence band, the $d$-band is

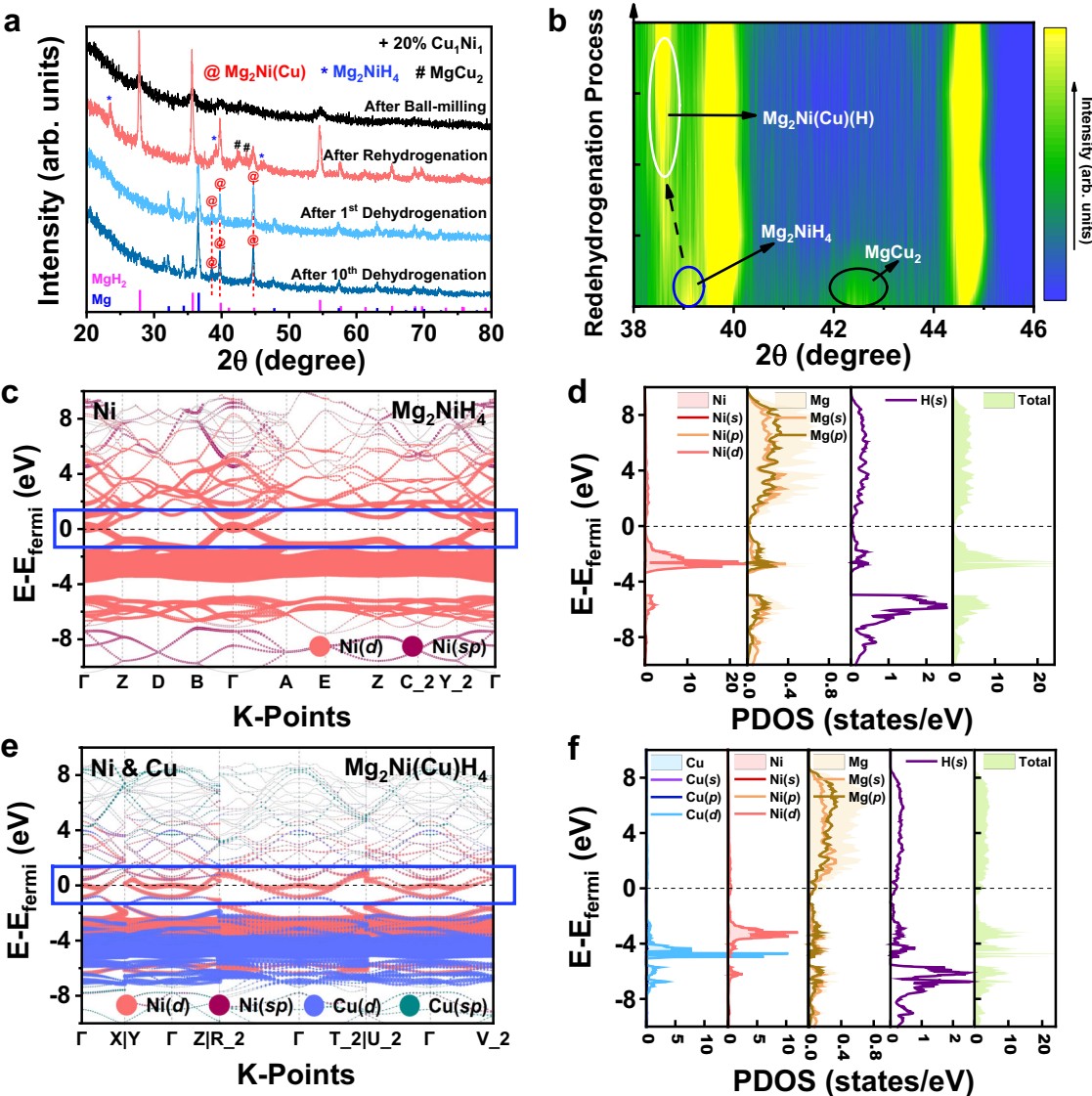

**Fig. 4 | Phase changes at different states and the electronic structures of catalytic phases. a** XRD patterns of MgH$_2$ under the catalysis of Cu$_1$Ni$_1$ at different states. **b** Contour plots of XRD patterns during re-dehydrogenation process of MgH$_2$ under the catalysis of Cu$_1$Ni$_1$. **c** Band structures of Ni-orbitals and **d** partial densities of states (PDOS) of corresponding atoms for Mg$_2$NiH$_4$. **e** Band structures of Ni(Cu)-orbitals and **f** partial densities of states (PDOS) of corresponding atoms for Mg$_2$Ni(Cu)H$_4$. Source data are provided as a source data file.

extended, which facilitates hybridization of them with Mg($sp$) and H($s$) orbitals, thereby providing facile pathway for inter-/intra-band transitions of electrons (including $s$-$p$ and $d$ orbitals) and leading to a broad absorption spectrum within the visible to near-infrared range[37–39]. In addition, the $d$ electrons may undergo transitions to a higher energy state when absorbing photons with more probability due to the overlap of the localization of $d$ orbitals, thereby inducing stronger inter-band absorption[40,41]. More importantly, the localization of $d$ electrons introduced by the presence of Ni and Cu could be well-preserved even after the formation of Mg$_2$Ni(Cu) alloy after dehydrogenation. These results indicate that the observed broad absorption spectrum of the hydrogenated state after cycling results from the synergistic effect of intra-band and inter-band transitions, and the in situ formation of Mg$_2$Ni(Cu) alloy expands the range of photothermal effects.

On the other hand, although it is well known that Mg$_2$Ni/Mg$_2$NiH$_4$ could function as the "hydrogen pump" to enhance hydrogen storage performance of MgH$_2$[42–44] as evidenced by the significant decrease of H$_2$ desorption temperature of MgH$_2$ induced by the catalysis of Ni, the

catalytic effect of CuNi in enhancing hydrogen storage performance of MgH$_2$ is even superior (Supplementary Fig. 7). To unveil the catalytic mechanism of Mg$_2$Ni(Cu)/Mg$_2$Ni(Cu)H$_4$ on MgH$_2$, theoretical calculations are conducted based on density functional theory (DFT). The average length of Mg–H bonds in Mg$_2$Ni(Cu)H$_4$ is calculated to be 2.03 Å, much longer than that in both MgH$_2$ (1.93 Å) and Mg$_2$NiH$_4$ (1.95 Å) (Supplementary Fig. 34). This result demonstrates that, in comparison to MgH$_2$ and Mg$_2$NiH$_4$, the dehydrogenation of Mg$_2$Ni(Cu)H$_4$ to form Mg$_2$Ni(Cu) is thermodynamically favored, which corresponds to the preferred formation of Mg$_2$Ni(Cu) alloy at the initial stage of re-dehydrogenation process (Fig. 4b). After the initial dehydrogenation of Mg$_2$Ni(Cu)H$_4$, under the presence of thus-formed Mg$_2$Ni(Cu) alloy, the barrier of H$_2$ desorption from MgH$_2$ is significantly reduced down to 1.13 eV, while this value reaches 3.12 eV for pristine MgH$_2$, validating the catalytic role of Mg$_2$Ni(Cu) alloy in improving the dehydrogenation performance of MgH$_2$ (Fig. 5a, b and Supplementary Fig. 35). More importantly, H atoms released from MgH$_2$ are spontaneously transferred into Mg$_2$Ni(Cu) alloy, which provides direct evidence for the "hydrogen pump" role of Mg$_2$Ni(Cu) in

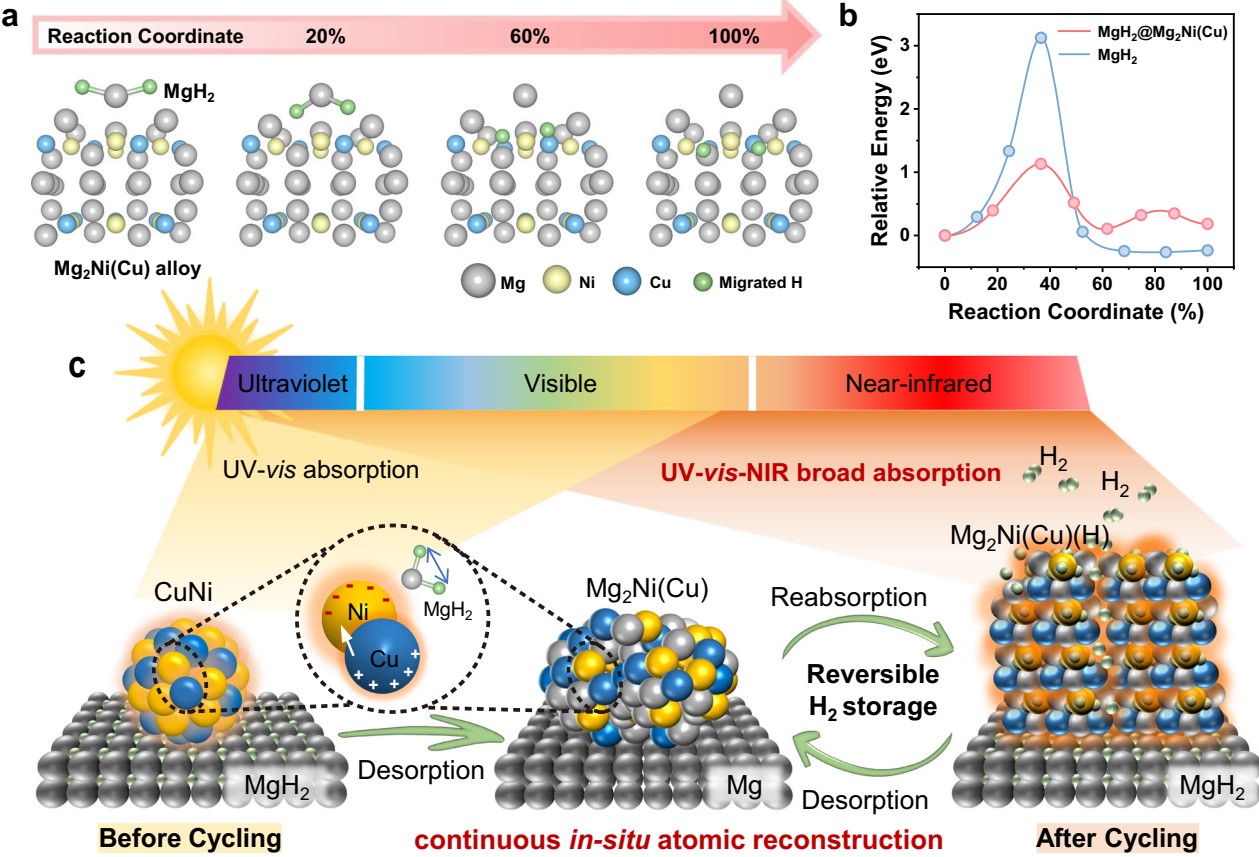

**Fig. 5 | The mechanism of the integration of photothermal and catalytic effects. a** Schematic illustration of $H_2$ desorption and H diffusion pathway of $MgH_2$ on the surface of $Mg_2Ni(Cu)$ alloy and **b** the corresponding energy profiles. **c** Schematic diagram of the ideal integration of photothermal and catalytic effects via continuous in situ atomic reconstruction upon repeated dehydrogenation process. Source data are provided as a source data file.

improving the dehydrogenation of $MgH_2$. In addition, the chemical disorders introduced by the interfaces and lattice distortions in the Mg–Cu lattice during cycling could enhance the reactivity of Mg and $Mg_2Ni(Cu)$ due to the formation of the diffusion pathways of H atoms, and serve as preferential catalytic sites for $H_2$ dissociation/ recombination[45,46], which corresponds well with the theoretical calculation results that the diffusion energy barriers of H atoms inside of $Mg_2Ni(Cu)$ is slightly lower than that in the bulk phases of $Mg_2Ni$. After hydrogenation, the diffusion energy barrier of H atoms inside of $Mg_2Ni(Cu)H_4$ (i.e., 0.62 eV) is significantly lower than that of $Mg_2NiH_4$ (i.e., 1.48 eV) (Supplementary Fig. 36), which endows $Mg_2Ni(Cu)$/ $Mg_2Ni(Cu)H_4$ superior "hydrogen pump" effect in improving hydrogen storage performance of $MgH_2$ than traditional $Mg_2Ni/Mg_2NiH_4$. Hence, the in situ stable formation of $Mg_2Ni(Cu)$ offers a kinetically favored pathway for repeated dehydrogenation an hydrogenation of $MgH_2$.

Furthermore, upon light irradiation, the presence of hot electrons induces internal disparities in electron distribution within CuNi and $Mg_2Ni(Cu)$ alloys as observed via in situ irradiated XPS (ISI-XPS) (Supplementary Fig. 37a–e). Attributed to the work function difference, the hot electrons generated by the LSPR effect of Cu transferred to Ni, leading to an increase in electron density around Ni (Supplementary Fig. 38a, b), which enables Ni sites to contribute more electrons to H bonded to Mg atoms of $MgH_2$ (Supplementary Fig. 39a–d), and hence leads to the weakening Mg–H bonds (Supplementary Fig. 39e–h). As a result, compared to pure Ni with uniform electron distribution, the electron enrichment at the Ni sites in CuNi and $Mg_2Ni(Cu)$ alloys facilitates the elongation of Mg–H bonds from 1.87 Å to 1.91 Å and 2.00 Å, respectively, while the impact of Cu sites remains unchanged. The internal uneven electron distribution within CuNi and

$Mg_2Ni(Cu)$ alloys is conducive to promoting the dissociation of Mg–H bonds and hence improving the "hydrogen pump" effect of $Mg_2Ni(Cu)/Mg_2Ni(Cu)H_4$. In addition, Mg does not affect the electron transfer between CuNi under light irradiation due to its positive charge, ensuring the promotion of Mg–H bonds dissociation by the in situ well-preserved $Mg_2Ni(Cu)$ during cycling. These hot electrons could be generated continuously under light irradiation and, however, cannot be directly participate in the $MgH_2$ dehydrogenation process, which subsequently would decay into localized thermal energy[47], acting on catalytic sites in the form of heat, thereby elevating the local temperature of catalytic sites and enhancing their catalytic effects.

Therefore, the in situ formation of $Mg_2Ni(Cu)$ alloys induces not only photothermal effect via broadening the light absorption range to cover the entire solar spectrum but also a light-enhanced "hydrogen pump" role in improving reversible hydrogen storage performance of $MgH_2$ (Fig. 5c). More importantly, the in situ atomic reconstruction of $Mg_2Ni(Cu)$ single phase upon repeated dehydrogenation process realizes the continuous integration of photothermal and catalytic roles to ensure the direct action of localized heat on the catalytic sites without any heat loss, thereby achieving the ideal coupling of photothermal and catalytic effects for driving reversible hydrogen storage of $MgH_2$ using solar energy.

## Discussion

In summary, a single-component phase of $Mg_2Ni(Cu)$ ternary alloy via atomic reconstruction is designed to achieve the ideal integration of photothermal and catalytic effects to enhance the solar-driven hydrogen storage performance of $MgH_2$. Taking advantage of the strong LSPR effect of CuNi alloys, the formation of $Mg_2Ni(Cu)$ ternary

alloy is realized through in situ alloying reaction between $MgH_2$ and CuNi alloys during the initial dehydrogenation process of $MgH_2$ under solar irradiation. Interestingly, $Mg_2Ni(Cu)$ alloy and its hydrogenated state (i.e., $Mg_2Ni(Cu)H_4$) exhibit metallic properties with strengthened intra/inter-band transitions, resulting in over 85% absorption in the full-spectrum range, which effectively elevate the surface temperature of $MgH_2$ to 261.8 °C under 2.6 W cm$^{-2}$ due to the promoted light-to-heat conversion efficiency. More importantly, in comparison with $MgH_2$, the hydrogen storage reaction of $Mg_2Ni(Cu)$ is thermodynamically and kinetically favored resulting from the weakening of Mg–H bonds and the low migration barrier of hydrogen atoms in both $Mg_2Ni(Cu)$ and $Mg_2Ni(Cu)H_4$, which in turn provides a facile pathway for the spontaneous breaking of Mg–H bonds of $MgH_2$. The uneven distribution of the light-induced hot electrons within CuNi and $Mg_2Ni(Cu)$ alloys also contributes to the effective weakening of Mg–H bonds of $MgH_2$ and the accelerated formation of $Mg_2Ni(Cu)$/$Mg_2Ni(Cu)H_4$, hence resulting in a light-enhanced "hydrogen pump" effect in improving cycling hydrogenation and dehydrogenation of $MgH_2$ and the significant decrease of the apparent activation energy required for driving hydrogen storage of $MgH_2$. As a result, the reversible formation of $Mg_2Ni(Cu)$ ideally integrates the photothermal and catalytic effects to ensure that the direct action of localized heat on the catalytic sites without any heat loss, which leads to the complete dehydrogenation of $MgH_2$ within 15 min under the irradiation intensity of 3.5 W cm$^{-2}$. Moreover, the stable reversibility between $Mg_2Ni(Cu)$ could be well-preserved during cycling $H_2$ desorption and adsorption, which results in the excellent stability of photothermal effect with full-spectrum light absorption and catalytic effect, leading to a reversible hydrogen storage capacity of 6.1 wt.% with a capacity retention of 95%. Our work expands the functionality of the alloying strategy to modify metal hydrides under solar irradiation and may enlighten the design of efficient photothermal-catalytic solar-driven reversible hydrogen storage systems and solar-chemical energy reactions.

## Methods

### Materials

Copper nitrate hydrate ($Cu(NO_3)_2 \cdot 3H_2O$), nickel nitrate hexahydrate ($Ni(NO_3)_2 \cdot 6H_2O$), tartaric acid ($C_4H_6O_6$), glycerol ($C_3H_8O_3$), and polyethylene glycol were all purchased from Sinopharm chemical reagent co., Ltd. Magnesium hydride ($MgH_2$) was purchased from Alfa Aesar. All chemicals were used without further purification.

### Synthesis of CuNi alloys

CuNi alloys were synthesized by a one-step solvothermal calcination method. Firstly, 386.6 mg of $Cu(NO_3)_2 \cdot 3H_2O$, 465.5 mg of $Ni(NO_3)_2 \cdot 6H_2O$, and 1200 mg of tartaric acid were added to 8 mL of deionized water and stirred until completely dissolved, referred to as Solution 1. Subsequently, 2 g of polyethylene glycol, 20 ml of glycerol, and 5 ml of deionized water were mixed and sonicated for 10 min, referred to as Solution 2. The two obtained solutions were mixed and stirred for 1 h to achieve uniform dispersion and transferred to a Teflon stainless-steel autoclave and heated at 150 °C for 3 h. The resulting product was centrifuged at about 3500×g and washed three times with anhydrous ethanol to remove excess reactants and impurities. The centrifuged product was then dried completely in an 80 °C oven for 12 h. The dried product was placed in a tube furnace and heated at a rate of 3 °C min$^{-1}$ in an $H_2$/Ar atmosphere (with 5% $H_2$) up to 800 °C for 2 h. CuNi alloys with encapsulated carbon layer were obtained through the gas reduction method, with a Cu to Ni atomic ratio of 1:1, denoted as $Cu_1Ni_1$. By varying the relative ratios of Cu and Ni salts added to precursor Solution 1, the ratio of CuNi alloys could be adjusted. Under the same reaction conditions, adding 184.74 mg $Cu(NO_3)_2 \cdot 3H_2O$ with 667.1 mg $Ni(NO_3)_2 \cdot 6H_2O$ yielded $Cu_1Ni_3$ with a Cu-to-Ni atomic ratio of 1:3. Without adding $Cu(NO_3)_2 \cdot 3H_2O$ or $Ni(NO_3)_2 \cdot 6H_2O$, Ni and Cu were obtained.

### Synthesis of $MgH_2$ catalyzed by CuNi alloys

CuNi–$MgH_2$ composite was synthesized via mechanical milling. Commercially available $MgH_2$ with 10 wt.% catalysts were added in a 100-mL stainless-steel container under an argon atmosphere. The ball-to-powder weight ratio was controlled to be 100:1, and the rotation speed of the miller was maintained at 500 rpm (alternate clockwise and counterclockwise rotation) for 12 h (5-min-stop every 30 min). Ball-milled $MgH_2$, Cu–$MgH_2$, and Ni–$MgH_2$ composites were also prepared under identical processing conditions. All samples were handled in an argon-filled glovebox with $O_2$ and $H_2O$ <0.1 ppm.

### Materials characterization

X-ray diffraction (XRD, D8 advance, Bruker AXS, USA) with Cu K$\alpha$ radiation (1.542 Å) at 40 kV and 40 mA was used to determine the phase compositions. The materials were protected against oxidation from the air by adhesive tape (amorphous with a broad peak around $2\theta \approx 20°$). X-ray photoelectron spectroscopy (XPS) results were obtained on a Thermo Scientific K-Alpha system equipped with a dual X-ray source, adopting an Al K$\alpha$ (1486.6 eV) anode with a hemispherical energy analyzer. In situ irradiated XPS (ISI-XPS) were performed on a Thermofisher ESCALAB 250Xi under Xe lamp irradiation. Time-of-flight (TOF) powder neutron diffraction (NPD) data were collected from a general-purpose powder diffractometer (GPPD) (90° bank) at the China Spallation Neutron Source (CSNS), Dongguan, China. Neutron diffraction patterns were recorded at room temperature. Rietveld refinements of the NPD patterns were performed using the GSAS software. The morphologies of the synthesized catalysts and $MgH_2$ catalyzed by CuNi with different states were observed by scanning electron microscopy (SEM, JEOL 7500FA, Japan) and transmission electron microscopy (TEM, JEOL JEM-2100F, Japan). The Cu and Ni contents of samples were measured by an inductively coupled plasma source mass spectrometer (ICP-MS, Agilent ICPOES730, USA). The absorption spectra of samples in the wavelength range of 200-2200 nm were determined using a UV–vis–NIR Spectrophotometer (Lambda 750, Perkin-Elmer, USA). The content of carbon and alloy were analyzed by thermogravimetric analysis (TGA, Discovery TGA 550, USA) from room temperature to 800 °C at a rate of 10 °C min$^{-1}$ in the air.

The light source was a 300 W Xenon lamp (Perfect Light Co., Ltd., China), and the light intensity was adjusted by tuning the current of the light source and the distance between the light source and the sample. The light intensity of the 300 W Xenon lamp was measured by an optical power meter (PL-MW2000, Perfect Light Co., Ltd., China). A short-wave infrared thermometer (DGE44N, range: 75-50 °C, DIAS, Germany) was used to measure the surface temperature of the composites during the reaction process under vacuum conditions through a sapphire window.

### Hydrogen storage measurements

Temperature-programmed desorption (TPD), isothermal dehydrogenation tests, and corresponding hydrogenation process of the as-prepared samples were conducted on a home-built high-pressure gas sorption apparatus (HPSA-auto), which was carefully calibrated by adopting $LaNi_5$ as a reference sample in terms of hydrogen storage capacity and guaranteed accuracy of ± 1%. All samples' hydrogen capacity (wt.%) is calculated based on the total mass of $MgH_2$ and the additives.

As for electric heating, for each volumetric temperature-programmed desorption measurement, approximately 30 mg of ball-milled sample were conducted under vacuum from room temperature to preset temperature (400 °C) at a heating rate of 5 °C min$^{-1}$ under an initial pressure lower than 0.0001 bar, with an electric heater to determine the hydrogen release properties of the $MgH_2$ materials briefly. The isothermal $H_2$ desorption and absorption test were handled by rapidly heating up to the target temperature followed by

keeping at the preset temperature. In the isothermal $H_2$ absorption experiment, ~80 mg of samples were used for hydrogenation under the hydrogen pressure of 10 atm.

As for light irradiation, customized solar-driven $H_2$ absorption and desorption equipment (Supplementary Fig. 11) was used to perform experiments under solar energy. The adjustment of the light intensity was realized by changing the current intensity. The light was incident through the above sapphire window, and the outlet of the reactor was connected to the PCT equipment. The temperature of the material during the $H_2$ absorption and desorption was measured by a short-wave infrared thermometer. About 30 mg samples were taken each time and pressed into pieces with a diameter of 15 mm under a pressure of 5 tons to ensure the stability of the test and the comparability of the results. Glass fiber sheets made up of mainly $SiO_2$ with low thermal conductivity, which have limited absorption of solar energy, were placed at the bottom of the reactor to reduce the heat dissipation of the material to the outside. The sample was placed on an $Al_2O_3$ ceramic sheet, which could be isolated from the glass fiber to prevent the sample from oxidizing by reacting with the insulation layer due to the high temperature during the reaction. The expansion of internal gases within the reactor due to solar irradiation has a negligible impact on the $H_2$ capacity results, as a large chamber, 25 times the volume of the reactor, was opened during the dehydrogenation tests to minimize testing errors. The hydrogen pressure used for $H_2$ absorption under solar irradiation was consistent with that using electric heating.

## Theoretical calculations

Density functional theory (DFT) calculations were carried out using the Vienna ab initio simulation package (VASP)[48,49]. Core-valence electron interaction was described by the projector-augmented wave (PAW) method[50], and electron exchange-correlation was processed by generalized gradient approximation (GGA) of Perdew-Burke-Ernzerhof (PBE) functional[51–53]. The van der Waals (vdW) correction DFT-D3 proposed by Grimme was employed to include the dispersion interaction[54]. A plane wave energy cutoff of 520 eV and Gamma-centered k-point meshes ($3 \times 3 \times 1$) with same density were applied to surface calculations. A $7 \times 7 \times 7$ Gamma-centered k-point meshes is used for the dos calculation of Cu, $MgH_2$, $Mg_2NiH_4$, $Mg_2Ni(Cu)H_4$, and $10 \times 10 \times 10$ is used for the dos calculation of CuNi. For all slab structures, a 20 Å vacuum layer in the $z$-direction is added to prevent surface interactions between the upper and lower surfaces. In energy band calculation, the number of k-points is uniformly 20, and the k-path and energy band information are exported by the VASP post-processing tool VASPKIT. The Gaussian smearing method was used with a width of 0.05 eV, and spin polarization was considered in all calculations. The structures were relaxed until the forces and total energy on all atoms were converged to less than 0.05 eV Å$^{-1}$ and $1 \times 10^{-5}$ eV. The $MgH_2$ dissociation and diffusions of H in various crystal lattices are simulated using the climbing-image nudged elastic band (CI-NEB) method[55]. The climbing-image nudged elastic band (CI-NEB) method developed by Henkelman et al. was employed to locate the transition states of the $MgH_2$ dissociation and diffusions of H in various crystal lattice[56,57].

## Reporting summary

Further information on research design is available in the Nature Portfolio Reporting Summary linked to this article.

# Data availability

The data that support the findings of the study are included in the main text and supplementary information files. All the raw data are provided in the Source Data file. Source data are provided with this paper.

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

## Acknowledgements

This work was financially supported by the National Key R&D Program of China (no. 2020YFA0406204 to X. Yu), the National Natural Science Foundation of China (no. U2130208 and 22279020 to G. Xia, 22109026 to Y. Sun), the Science and Technology Commission of Shanghai Municipality (no. 21ZR1407500 and 23ZR1406500 to G. Xia).

## Author contributions

G. Xia and X. Yu conceived the study and supervised this work. X. Zhang performed the material synthesis, characterization, hydrogen storage measurements and analyzed the data. S. Ju and C. Li performed the theoretical calculations. J. Hao, J. Chen, and L. He, carried out NPD measurements and analyzed the data. S. Ju, C. Li, and X. Hu helped to analyze the data. Y. Sun and W. Chen contributed to the design of the reactor. G. Xia, D. Sun, F. Fang, and X. Yu revised the manuscript. X. Yu, G. Xia, and X. Zhang co-wrote the paper with input from all the authors. All authors contributed to the discussion.

## Competing interests

The authors declare no competing interests.
