## [Peer Review File · Nature Communications]

REVIEWER COMMENTS

Reviewer #1 (Remarks to the Author):

The paper presents an interesting study on the integration of photothermal and catalytic properties using the single component phase of the Mg₂Ni(Cu) ternary alloy for solar-driven hydrogen storage. This work undoubtedly innovatively extends the functionality of alloying strategies in realizing reversible hydrogen storage of metal hydrides. The experimental data provided in this paper is sound, and the conclusions drawn are well motivated. Thus, this manuscript could be published after careful improvement.

1. The in-situ generated Mg₂Ni(Cu) exhibits excellent photothermal effect. Would the increased proportion of the catalyst further enhance the photothermal effect brought about by the integrated construction? Do they have advantages in terms of hydrogen release rate and temperature? It would further support the conclusion proposed by the authors regarding the photothermal effect of in-situ generated Mg₂Ni(Cu).
2. During dehydrogenation, a portion of Mg in MgH₂ forms Mg₂Ni(Cu)H. Does this lead to a loss in the hydrogen capacity of the system?
3. Is there a difference in catalytic performance between CuNi alloy and Cu and Ni nanoparticles with a molar ratio of 1:1? A comparison of the catalytic performance between CuNi alloy and the addition of Cu and Ni nanoparticles should be conducted. Moreover, it would be valuable to determine if the catalytic phase still forms Mg₂Ni(Cu) alloy.
4. How does the catalytic effect of MgH₂ with CuNi alloy compared to the reported catalysts? Much more data related to alloy-catalyst-modified MgH₂ are needed for comparison, such as the enthalpy and the temperature of dehydrogenation.

Reviewer #2 (Remarks to the Author):

Hydrogen is a potential clean energy carrier for developing a carbon-neutral system and of great significance for realizing a global transition to a sustainable energy economy. In the manuscript, the authors reported that the Mg₂Ni(Cu) ternary alloy via atomic reconstruction is designed to achieve the ideal integration of excellent photothermal and catalytic effect for realizing stable solar-driven hydrogen storage of MgH₂. Although the work contains a number of a characterization techniques and testing of samples, however the article lacks innovation in both mechanism explanation and material design. Magnesium hydride (MgH₂) materials have been widely used in the fields of the reversible storage of hydrogen. The alloys are currently widely used in various material designs, and the author has not

proposed any new highlights. About the mechanism explanation, the author has not proposed a more reasonable explanation or new point. Light only provides heat during the reaction process, and its significance is not very significant (LSPR). Based on the current requirements of Nat. Commun., at present stage, I do not suggest its publication in this journal.

1. The H₂ capacity comparison under different light intensities irradiation should be given (under the same temperature). In addition, also need to study the effect of temperature on its performance. (under the same light intensity)
2. In Figure 2d and 2e, the author compared the performance differences between photothermal and thermal, and did not provide a reasonable explanation for why photothermal performance was higher than thermal. What is the role of light in this work.
3. How did the author control the temperature during the light irradiation process? A detailed explanation was needed.
4. In Figure 1, the current TEM data cannot prove that CuNi is an alloy structure, and the author needs to provide clearer evidence.
5. The author synthesized CuNi alloy materials with different components (Cu₁Ni₁ and Cu₁Ni₃). How about Cu₁Ni₂.
6. The author claims that it is the LSPR effect of CuNi alloy, but does not provide sufficient evidence. In Figure 2b, there is no characteristic absorption peak of Cu or CuNi.

Reviewer #3 (Remarks to the Author):

This paper describes reversible hydrogen storage on the Mg₂Ni(Cu)H₄ under solar light irradiation. By addition of Ni and Cu into the MgH₂, the samples showed better solar light absorption property, leading to higher surface temperature and better properties related to hydrogen storage. This is very interesting; it seems that relation between light absorption and increase of temperatures might be very important in this paper. Thus, the authors should discuss the relation and the mechanism of the light absorption more deeply. Especially, following points as listed below should be discussed carefully.

1) Figure 4f shows DOS of the samples. To compare the density (peak intensity) among the orbitals, the authors should add the scale. The unit of a.u. is not suitable. More accurate unit should be used.

2) L291

The authors described that in terms of pure MgH₂, the valence band is mainly contributed by H(s) orbitals and the conduction band is composed of Mg(s) and Mg(p) orbitals. However, Figure 4d also suggested that Mg orbitals and H orbitals contribute to the conduction band and valence band, respectively. So, the authors should calculate the contribution percentages of H and Mg orbitals at the valence band and the conduction band. (Qualitative discussion is needed.)

3)

L296 The authors showed that strong absorption peaks in the visible light range could be observed after the addition of CuNi alloy into MgH₂, mainly attributed to the LSPR effect of the metal catalyst.

However, from the UV-vis absorption spectra it is hard to conclude that strong absorption peaks in the visible light range was mainly attributed to the LSPR effect. Quantitative discussion is needed. The authors should do peak separation for good understanding of the absorption spectra; the absorption may be sum of d-d transition, LSPR, and other effects.

4) L304

The authors insisted that the electronic distribution of Mg(s) and Mg(p) in both Mg₂NiH₄, MgCu₂, and Mg₂Ni(Cu)H₄ tends to fill the originally vacant Fermi level, resulting in the uniform distribution of s-p band.

The authors should discuss why Mg s and p orbitals in Mg₂Ni(Cu)H₄ fill the vacant Fermi level although the Mg orbitals in MgH₂ do not fill.

5) L306

It was described that this provides facile pathway for intra-band transitions of electrons (including s-p and d orbitals), leading to a broad absorption spectrum within the visible to near-infrared range and the localization of d electrons is capable of inducing stronger inter-band absorption than electrons in the s and p orbitals.

The authors should more carefully discuss which orbitals contribute to the intra band and inter band transitions. Furthermore, the authors should explain why the localization of d electrons induces stronger inter-band absorption.

Response to Reviewers

Dear Reviewers,

Thank you very much for providing such constructive and insightful comments on our manuscript. We have meticulously incorporated these suggestions while preparing the revised version of our paper. We believe that these modifications have significantly enhanced the overall quality of this paper.

Below, our responses to your valuable comments are highlighted in blue. Additionally, all the changes made in the revised manuscript have been marked in red for your convenience.

Response to Reviewer #1

Reviewer #1: The paper presents an interesting study on the integration of photothermal and catalytic properties using the single component phase of the $Mg_2Ni(Cu)$ ternary alloy for solar-driven hydrogen storage. This work undoubtedly innovatively extends the functionality of alloying strategies in realizing reversible hydrogen storage of metal hydrides. The experimental data provided in this paper is sound, and the conclusions drawn are well motivated. Thus, this manuscript could be published after careful improvement.

Response: We sincerely thank the reviewer for these valuable comments for strengthening our manuscript. We have carefully addressed the reviewer's concerns point by point as follows.

1. The in-situ generated $Mg_2Ni(Cu)$ exhibits excellent photothermal effect. Would the increased proportion of the catalyst further enhance the photothermal effect brought about by the integrated construction? Do they have advantages in terms of hydrogen release rate and temperature? It would further support the conclusion proposed by the authors regarding the photothermal effect of in-situ generated $Mg_2Ni(Cu)$.

Response: We thank the reviewer for this constructive comment. In the manuscript, we have compared the catalytic performance of MgH_2 catalyzed by bimetallic CuNi alloys with that of Ni and Cu catalysts, demonstrating the superior photothermal effect of CuNi-catalyzed MgH_2 due to the *in-situ* formation of $Mg_2Ni(Cu)$. We fully concur with the reviewer's suggestion to further investigate the photothermal effect of *in-situ* alloying by varying the content of catalysts. As proved in the part of "Mechanism of Photothermal and Catalytic Effect", together with the initial dehydrogenation process,

the *in-situ* alloying reaction between Mg and CuNi results in simultaneous atomic reconstruction, leading to the formation of the Mg₂Ni(Cu) ternary alloy. Therefore, in addition to 10% catalyst loading, we have extended our investigation by adding 5% and 20% Cu₁Ni₁ to explore the initial dehydrogenation performance under 2.5 W/cm² irradiation.

As expected, the increased loading ratio of Cu₁Ni₁ leads to the elevation of the stable temperature under the same illumination intensity (Fig. S16a). As a result, upon increasing the loading ratio of Cu₁Ni₁ to 20%, the temperature of MgH₂ could be elevated to 238.1 °C, 7.7 °C higher than that with a loading ratio of Cu₁Ni₁ of 10%, which indicates that increasing the catalyst loading indeed confers a photothermal advantage under the same illumination intensity.

Accordingly, the higher temperature results in more thorough dehydrogenation of MgH₂ with 20% Cu₁Ni₁ loading, as evidenced by the increase of the extent of dehydrogenation from 83.28% under 10% Cu₁Ni₁ catalysis to 93.05% in the light intensity of 2.5 W/cm² (Fig. S16b). The normalized H₂ desorption curves further demonstrate faster dehydrogenation rate of MgH₂ under the catalysis of 20% Cu₁Ni₁ (Fig. S16c). Combining the heat-driven TPD curves under different Cu₁Ni₁ loading ratios (Fig. S16d), it is evident that a higher catalyst content also contributes to a partial reduction in the peak dehydrogenation temperature of MgH₂. This further validates that the synergic enhancement of photothermal and catalytic effects through the integrated construction of Mg₂Ni(Cu) is responsible for the solar-driven dehydrogenation.

Nevertheless, the increase of the loading ratio of Cu₁Ni₁ would decrease the systematic H₂ storage capacity down to 4.82 wt.%, 0.11 wt.% lower than that of 10% Cu₁Ni₁. Therefore, it is of great significance to optimize the balance between the photothermal effect of CuNi alloys and the systematic H₂ storage capacity for practical applications, which also provides a basis for the selection of a 10% loading in this work.

Furthermore, MgH₂ catalyzed by 5% Cu₁Ni₁ reaches only a temperature of 221.7 °C with a dehydrogenation capacity of 3.71 wt.%, corresponding to a completion rate of merely 55.71%, similarly validating the indispensable role of *in-situ* generated Mg₂Ni(Cu) in both photothermal and catalytic effects.

We appreciate the reviewer's comment and all the above-mentioned results and the related discussion (highlighted in red color) have been provided in the revised manuscript as follows:

[Manuscript - Page 9]

“... Therefore, MgH₂ catalyzed by Cu₁Ni₁ releases 5.3 wt.% H₂ within 10 minutes under 2.8 W/cm² irradiation, whereas the time to achieve identical H₂ desorption capacity using external heating is doubled (Fig. 2h). Furthermore, upon increasing the loading ratio of Cu₁Ni₁ to 20%,

the temperature of MgH_2 could be elevated to $238.1\text{ }^\circ\text{C}$, $7.7\text{ }^\circ\text{C}$ higher than that with a loading ratio of Cu_1Ni_1 of 10% (Fig. S16a). The extent of dehydrogenation increases to 93.05% under 2.5 W/cm^2 , compared with 83.28% under the addition of 10% Cu_1Ni_1 , together with a faster dehydrogenation rate (Fig. S16b-c), which could be attributed to the enhanced photo-thermal effect combined with the catalytic effect in further reducing the dehydrogenation temperature (Fig. S16d). This further validates that, with the increase ratio of CuNi alloys, the simultaneous enhancement of photothermal and catalytic effects of CuNi alloys would mutually contribute to the better solar-driven performance.”

[Supporting Information - Page 17]

Fig. S16. a) Surface temperatures, b) the comparison of H_2 capacity with theoretical capacity and completion ratio, and c) the normalized H_2 desorption curves of MgH_2 under the catalysis of Cu_1Ni_1 with various ratios using a light intensity of 2.5 W/cm^2 . d) TPD curves of MgH_2 under the catalysis of Cu_1Ni_1 with various ratios.

Nevertheless, the increase of the loading ratio of Cu_1Ni_1 would decrease the systematic H_2 storage capacity down to 4.82 wt.%, 0.11 wt.% lower than that of 10% Cu_1Ni_1 . Therefore, the balance between the photothermal effect of CuNi alloys and the systematic H_2 storage capacity is of great significance to be optimized for practical applications, which also provides a basis for the selection of a loading ratio of 10% in this work.

Furthermore, MgH_2 catalyzed by 5% Cu_1Ni_1 reaches only a temperature of $221.7\text{ }^\circ\text{C}$ with a

dehydrogenation capacity of 3.71 wt.%, corresponding to a completion rate of merely 55.71%, similarly validating the indispensable role of *in-situ* generated Mg₂Ni(Cu) in both photothermal and catalytic effects.

2. During dehydrogenation, a portion of Mg in MgH₂ forms Mg₂Ni(Cu)H. Does this lead to a loss in the hydrogen capacity of the system?

Response: Thanks for the invaluable comment. Taking 10% catalyst addition employed in this work as an example, the theoretical rehydrogenation with H₂ capacity maintenance rate of Mg under the catalysis of CuNi alloys, Cu, and Ni are calculated (Tab. S6). It was mentioned in the original manuscript that the reversible formation of Mg₂Ni(Cu) with a Cu atomic ratio of x in CuNi during H₂ desorption/absorption could be summarized as: $2\text{Mg}_2\text{Ni}_{1-x}\text{Cu}_x + (4 - x)\text{H}_2 \leftrightarrow 2\text{Mg}_2\text{Ni}_{1-x}(\text{Cu})_x(\text{H}) \leftrightarrow (2 - 2x)\text{Mg}_2\text{NiH}_4 + 3x\text{MgH}_2 + x\text{MgCu}_2$.

Specifically, the addition of Cu is detrimental to the retention of the system's theoretical H₂ storage capacity of MgH₂, with 97.5% H₂ maintenance rate under 10% Cu catalyst. While Mg₂NiH₄ is formed for MgH₂ under the catalysts of Ni, which possesses the same theoretical H₂ absorption capacity per Mg unit as pure MgH₂, and thus it does not result in a significant loss of the system's H₂ capacity. As for CuNi alloys, the theoretical H₂ capacity maintenance rate of MgH₂ under the catalysts of Cu₁Ni₁, Cu₁Ni₂, and Cu₁Ni₃ is 98.8%, 99.2%, and 99.4%, respectively. Therefore, a loss in H₂ capacity of the system could nearly be neglected under the catalysts of CuNi alloys with low addition ratios ($\leq 10\%$).

In response to this comment, we have supplemented the calculation results of the theoretical H₂ capacity maintenance rate of MgH₂ with the catalysts of CuNi alloys, Cu, and Ni, including pure MgH₂ for comparison in the revised manuscript and supporting information as follows:

[Manuscript - Page 12]

“A similar reversible transformation between Mg₂Ni(Cu), Mg₂NiH₄, and MgCu₂ during H₂ desorption/absorption is also observed in MgH₂ catalyzed by equimolar ratios of Cu and Ni, further validating the reversible formation of Mg₂Ni(Cu) (Fig. S22f-g). In addition, a loss in H₂ capacity of the above system could nearly be neglected under the catalysts of CuNi alloys with a loading ratio less than 10% (Tab. S6).”

[Supporting Information - Page 26]

Tab. S6. The theoretical rehydrogenation with H₂ capacity maintenance rate of Mg under the catalysis of CuNi alloys, Cu, and Ni, including pure MgH₂ for comparison.

Samples	Theoretical H ₂ Absorption of Mg in System	Theoretical H ₂ Capacity Maintenance Rate
MgH ₂	1 Mg: 1 H ₂	100.0%
+10% Ni	1 Mg: 1 H ₂	100.0%
+10% Cu ₁ Ni ₁	90% (1 Mg: 1 H ₂) + 10% (1 Mg: 0.875 H ₂)	98.8%
+10% Cu ₁ Ni ₂	90% (1 Mg: 1 H ₂) + 10% (1 Mg: 0.917 H ₂)	99.2%
+10% Cu ₁ Ni ₃	90% (1 Mg: 1 H ₂) + 10% (1 Mg: 0.938 H ₂)	99.4%
+10% Cu	90% (1 Mg: 1 H ₂) + 10% (1 Mg: 0.750 H ₂)	97.5%

The addition of Cu is detrimental to the retention of the system's theoretical H₂ storage capacity of MgH₂, with 97.5% H₂ maintenance rate under 10% Cu catalyst. While Mg₂NiH₄ is formed for MgH₂ under the catalysts of Ni, which possesses the same theoretical H₂ absorption capacity per Mg unit as pure MgH₂, and thus it does not result in a significant loss of the system's H₂ capacity. As for CuNi alloys, the theoretical H₂ capacity maintenance rate of MgH₂ under the catalysts of Cu₁Ni₁, Cu₁Ni₂, and Cu₁Ni₃ is 98.8%, 99.2%, and 99.4%, respectively. Therefore, a loss in H₂ capacity of the system could nearly be neglected under the catalysts of CuNi alloys with low loading ratios ($\leq 10\%$).

3. Is there a difference in catalytic performance between CuNi alloy and Cu and Ni nanoparticles with a molar ratio of 1:1? A comparison of the catalytic performance between CuNi alloy and the addition of Cu and Ni nanoparticles should be conducted. Moreover, it would be valuable to determine if the catalytic phase still forms Mg₂Ni(Cu) alloy.

Response: We appreciate the reviewer's careful review of our manuscript. To further discuss the difference in catalytic performance between Cu₁Ni₁ alloys with equimolar Cu and Ni, Cu and Ni nanoparticles prepared by the same method are added into MgH₂ with Cu at 5.2% and Ni at 4.8% and the total catalyst loading remains at 10 wt.%, denoted as "+ Equimolar Cu and Ni".

As for the dehydrogenation performance of TPD curves at a heating rate of 2 °C/min, the dehydrogenation peak temperature for equimolar Cu and Ni is 253.9 °C, which is 7.5 °C higher than that of Cu₁Ni₁ (246.6 °C). Additionally, the dehydrogenation onset temperature for equimolar Cu and Ni is 12.9 °C higher than that of Cu₁Ni₁ (196.9 °C), and the temperature range required for dehydrogenation is also broader (Fig. S7c). These results indicate that, compared to Cu₁Ni₁ alloys, equimolar ratios of Cu and Ni exhibit a weakened catalytic effect on MgH₂, which may be attributed to the lack of close contact between Cu and Ni, hindering the rapid formation of the catalytic phase Mg₂Ni(Cu) ternary alloy during the dehydrogenation process. Moreover, the addition of equimolar Cu and Ni decreases the dehydrogenation activation energy of MgH₂ to

88.79 ± 5.37 kJ/mol, which is also much lower than that of pure MgH_2 without catalysts (125.04 ± 5.72 kJ/mol) but a little higher than that of the Cu_1Ni_1 catalyzed sample (81.39 ± 5.18 kJ/mol) (Fig. S8e-f, Fig. S9a-b). This provides a reasonable explanation for the superior catalytic effect of CuNi alloys compared to equimolar ratios of Cu and Ni, further demonstrating the synergistic effect of Cu and Ni alloying on the improvement of MgH_2 dehydrogenation performance.

In terms of the differences in catalytic mechanisms between them, the phase transformation of MgH_2 catalyzed by equimolar Cu and Ni catalysis is investigated at different states during cycling (Fig. S22f-g). Interestingly, MgH_2 catalyzed by equimolar ratios of Cu and Ni exhibit the same reversible transformation between $\text{Mg}_2\text{Ni}_{0.5}\text{Cu}_{0.5}$ alloy, MgCu_2 , and Mg_2NiH_4 during H_2 desorption/absorption, which is in good agreement with the catalytic mechanism proposed for Cu_1Ni_1 on MgH_2 in our original manuscript. This demonstrates that the catalytic mechanism based on CuNi alloying could also be achieved by separately adding monometallic Cu and Ni, validating the reversible formation of $\text{Mg}_2\text{Ni}(\text{Cu})$ and the universality of the *in-situ* atomic reconstruction strategy for solar-driven hydrogen storage *via* alloying.

We appreciate the reviewer's comment and the above-mentioned results and the related discussion (highlighted in red color) have been provided in the revised manuscript and supporting information as follows:

[Manuscript - Page 12]

“A similar reversible transformation between $\text{Mg}_2\text{Ni}(\text{Cu})$, Mg_2NiH_4 , and MgCu_2 during H_2 desorption/absorption is also observed in MgH_2 catalyzed by equimolar ratios of Cu and Ni, further validating the reversible formation of $\text{Mg}_2\text{Ni}(\text{Cu})$ (Fig. S22f-g). In addition, a loss in H_2 capacity of the above system could nearly be neglected under the catalysts of CuNi alloys with addition less than 10% (Tab. S6).”

[Supporting Information - Page 7]

Fig. S7. a) TPD curves and b) TPD derivative curves of MgH₂ under the catalysis of Cu₁Ni₁, Cu₁Ni₂, Cu₁Ni₃, Cu₃Ni₁, Ni, and Cu, including pure MgH₂ for comparison. c) TPD curves of MgH₂ under the catalysis of Ni with various ratios, including MgH₂ catalyzed by 10% CuNi and equimolar Cu and Ni for comparison. d) Isothermal dehydrogenation profiles of MgH₂ under the catalysis of Cu₁Ni₁ at various temperatures. e) Isothermal hydrogenation curves at 250 °C under 1 MPa of MgH₂ under the catalysis of Cu₁Ni₁, Cu₁Ni₂, Cu₁Ni₃, Cu₃Ni₁, Ni, and Cu, respectively, including ball-milled MgH₂ for comparison. f) Isothermal hydrogenation curves of MgH₂ under the catalysis of Cu₁Ni₁ at various temperatures.

To further demonstrate the synergistic effect of CuNi alloys, equimolar Cu and Ni nanoparticles (Cu at 5.2% and Ni at 4.8%) are added separately into MgH₂ with a total catalyst loading remaining at 10 wt.%. The dehydrogenation peak temperature for equimolar Cu and Ni is 253.9 °C, which is 7.5 °C higher than that of Cu₁Ni₁ (246.6 °C). Additionally, the dehydrogenation onset temperature for equimolar Cu and Ni is 12.9 °C higher than that of Cu₁Ni₁ (196.9 °C), and the temperature range required for dehydrogenation is also broader. These results indicate that, compared to Cu₁Ni₁ alloys, equimolar ratios of Cu and Ni exhibit

a weakened catalytic effect on MgH_2 , which may be attributed to the lack of close contact between Cu and Ni, hindering the rapid formation of the catalytic phase $\text{Mg}_2\text{Ni}(\text{Cu})$ ternary alloy during the dehydrogenation process.

Fig. S8. TPD curves and their corresponding derivative curves of MgH_2 under the catalysis of a-b) Cu_1Ni_1 , c-d) Cu_1Ni_3 , e-f) equimolar Cu and Ni, g-h) Ni, i-j) Cu, including k-l) ball-milled MgH_2 for comparison.

The activation energy (E_a) for dehydrogenation of MgH_2 with different catalysts (selecting Cu_1Ni_1 and Cu_1Ni_3 as representatives of CuNi alloys due to the similar dehydrogenation performance) are quantitatively calculated (Fig. S9a) using the Kissinger equation by TPD curves under different heat rates (Fig. S8):

$$\ln(\beta/T_p^2) = A - E_a/RT_p \quad (\text{Equ. S1})$$

where β and T_p are the heating rate and dehydrogenation peak temperature, respectively, A is a temperature-independent coefficient, and R is the universal gas constant.

The addition of Cu_1Ni_1 and Cu_1Ni_3 decreases the dehydrogenation activation energy of MgH_2 to 81.39 ± 5.18 kJ/mol and 80.81 ± 4.99 kJ/mol, respectively, which is not only much lower than that of pure MgH_2 without catalysts (125.04 ± 5.72 kJ/mol) but also lower than that of the Ni catalyzed sample (89.79 ± 6.38 kJ/mol). Moreover, the addition of equimolar Cu and Ni decreases the dehydrogenation activation energy of MgH_2 to 88.79 ± 5.37 kJ/mol, which is a little higher than that of the Cu_1Ni_1 catalyzed sample. Interestingly, the activation energy of Cu-catalyzed MgH_2 is even higher than that of pure MgH_2 without catalysts, indicating that pure Cu has no catalytic effect on MgH_2 .

Fig. S9. a) Kissinger's plots and b) activation energies for dehydrogenation of MgH_2 under the catalysis of Cu_1Ni_1 , Cu_1Ni_3 , equimolar Cu and Ni, Ni, and Cu, including ball-milled MgH_2 for comparison.

[Supporting Information - Page 21]

Fig. S22. a) XRD patterns of MgH_2 under the catalysis of 10% Cu_1Ni_1 at different states. XRD patterns of MgH_2 b) after ball-milling, c) after rehydrogenation, and after dehydrogenation under the catalysis of Cu_1Ni_1 , Cu_1Ni_2 , Cu_1Ni_3 , Ni, and Cu, in the 2θ range of d) 20~80 degrees and e) an enlarged view of 38~46 degrees. XRD patterns of MgH_2 after dehydrogenation under the catalysis

of equimolar Cu and Ni, in the 2θ range of f) 20~80 degrees and g) an enlarged view of 38~46 degrees, including Cu_1Ni_1 and Ni for comparison.

4. How does the catalytic effect of MgH_2 with CuNi alloy compared to the reported catalysts? Much more data related to alloy-catalyst-modified MgH_2 are needed for comparison, such as the enthalpy and the temperature of dehydrogenation.

Response: We sincerely appreciate the reviewer for providing this valuable comment. In response to this comment, more recently reported results related to Ni-based or bimetallic-catalysts for MgH_2 have been added in the revised manuscript (as shown in Tab. S3 below), including the comparison of the temperature (onset and peak H_2 desorption temperature) and the activation energies of dehydrogenation with the H_2 capacity of several MgH_2 -catalyst hydrogen storage systems.

By comparison, MgH_2 catalyzed by CuNi alloys exhibits excellent performance in terms of both the temperature and activation energy of dehydrogenation, which are among the leading reported Ni-based or bimetallic-catalyst for MgH_2 hydrogen storage materials so far.

We appreciate the reviewer's comment and the comparison of the temperature of dehydrogenation, activation energies for H_2 desorption and H_2 capacity of several MgH_2 -catalyst hydrogen storage systems (Tab. S3) have been added in the revised supporting information as follows:

[Supporting Information - Page 10]

And by comparison, MgH_2 catalyzed by CuNi alloys exhibits excellent performance in terms of both the temperature and activation energy of dehydrogenation, which are among the leading reported Ni-based or bimetallic-catalyst for MgH_2 hydrogen storage materials so far (Tab. S3).

Tab. S3. Comparison of the onset and peak H_2 desorption temperature with the H_2 capacity of several MgH_2 -catalyst hydrogen storage systems.

Dopants	Onset Temperature (°C)	Peak Temperature (°C)	E_a (kJ/mol H_2)	H_2 Capacity (wt.%)	Ref.
+ FeNi/rGO	230	273	93.6	6.9	S1
+ $\text{Ni}_3\text{Fe}/\text{BC}$	185	272	102	6.4	S2
+ $\text{Fe}_{0.64}\text{Ni}_{0.36}@C$	250	343	86.9	5.2	S3
+ Ni-Nb/rGO	198	230	57.9	6.4	S4
+ NiMoO_4	243	257	85.9	6.5	S5
+ NiV_2O_6	227	—	92.9	5.8	S6
+ $\text{Ni}_3(\text{VO}_4)_2$	210	280	72.6	5.5	S7

+ CoNi@C	173	254	82.3	6.2	S8
+ NiTiO ₃	235	261	74	6.9	S9
+ PdNi metallene	200	263	62.5	6.4	S10
+ MoNi alloy	180	268	89.7	6.7	S11
+ NbTiC	195	230	80	6.8	S12
+ TiNb ₂ O ₇	177	—	96	6.3	S13
+ Cu ₁ Ni ₁ alloys	192.5	246.4	81.4	6.6	This work
+ Cu ₁ Ni ₃ alloys	198.7	244.7	80.8	6.6	

Response to Reviewer #2

Reviewer #2: Hydrogen is a potential clean energy carrier for developing a carbon-neutral system and of great significance for realizing a global transition to a sustainable energy economy. In the manuscript, the authors reported that the Mg₂Ni(Cu) ternary alloy via atomic reconstruction is designed to achieve the ideal integration of excellent photothermal and catalytic effect for realizing stable solar-driven hydrogen storage of MgH₂. Although the work contains a number of a characterization techniques and testing of samples, however the article lacks innovation in both mechanism explanation and material design. Magnesium hydride (MgH₂) materials have been widely used in the fields of the reversible storage of hydrogen. The alloys are currently widely used in various material designs, and the author has not proposed any new highlights. About the mechanism explanation, the author has not proposed a more reasonable explanation or new point. Light only provides heat during the reaction process, and its significance is not very significant (LSPR). Based on the current requirements of Nat. Commun., at present stage, I do not suggest its publication in this journal.

Response: We sincerely appreciate the reviewer's thorough and professional evaluation of our article. Based on your invaluable suggestions, we have made extensive improvements to our previous draft.

As stated in the introduction section, although MgH₂ is widely regarded as a potential solid-state hydrogen storage material due to its high gravimetric and volumetric hydrogen density and excellent reversibility, a theoretical operating temperature of approximately 280 °C should be supplied to drive reversible hydrogen storage reaction of MgH₂ and the extra high kinetic barrier would lead to even higher operating temperature of 400 °C. As a result, the reversible hydrogen storage of MgH₂ driven by

external heating is inevitably constrained by massive energy input and low systematic energy density even after extensive research for a long time. To solve this critical issue, we have recently proposed the concept of solar-driven reversible hydrogen storage of metal hydrides by using solar energy as the sustainable energy source. The light adsorber and thermal catalysts are physically separated and the phase separation of thermochemical catalysts, light adsorber, and MgH_2 , however, inevitably leads to the delay of the transfer of thus-generated heat from photothermal materials towards catalytic sites and the loss of heat, and hence an extremely high light intensity of 4.0 W/cm^2 is required for driving reversible hydrogen storage reaction of MgH_2 . Therefore, the integration of excellent photothermal and catalytic effect is of great significance but still a big challenge to realize reversible hydrogen storage of MgH_2 using solar energy as the only energy input. However, the research in this field has yet to be explored up to now as far as we know.

In this work, a single component phase of $\text{Mg}_2\text{Ni}(\text{Cu})$ ternary alloy *via* atomic reconstruction is found and designed to achieve ***the ideal integration of excellent photothermal performance with full-spectrum light absorption and catalytic effect*** that significantly lowers the apparent activation energy of hydrogen storage reaction for realizing stable solar-driven hydrogen storage of MgH_2 . First, in terms of materials design, an *in-situ* alloying reaction between MgH_2 and CuNi is developed to achieve reversible continuous atomic construction and design a single component phase of $\text{Mg}_2\text{Ni}(\text{Cu})$, which integrates the photothermal performance with full-spectrum light absorption and catalytic effects, to ensure that the direct action of localized heat on the catalytic sites without any heat loss. This kind of material designs lead to the complete dehydrogenation of MgH_2 within 15 minutes under the irradiation intensity of 3.0 W/cm^2 is achieved (Fig. R1b), showing the superiority of fast response and rate of H_2 desorption under solar irradiation over direct heating (Fig. R1a). Moreover, the stable reversible generation of $\text{Mg}_2\text{Ni}(\text{Cu})$ alloy single phase *via* continuous *in-situ* atomic reconstruction upon repeated dehydrogenation process enables the well-preserved integration of photothermal and catalytic roles (Fig. R1c), thereby achieving the ideal coupling of photothermal and catalytic effect with excellent stability, delivering a reversible hydrogen storage capacity of 6.1 wt.% with a capacity retention of 95% within 15 minutes under 3.5 W/cm^2 (Fig. R1d). On the other hand, we further verify the solid solution phase $\text{Mg}_2\text{Ni}_{1-x}(\text{Cu})_x$ by refined powder neutron diffraction, and elucidate the mechanism of enhanced photothermal effect through intra/inter-band transitions with metal properties. Due to the solubility of Cu , a more flattened hybridized orbital, resembling a "flat band" appears, and thus a continuous band structure fills the vacant Fermi level in $\text{Mg}_2\text{Ni}(\text{Cu})(\text{H})$ with metallic properties, which induces the facile excitation of electrons under light irradiation. This mechanism explanation expands the applicability of the conventional alloying strategy to modify

other metal hydrides under solar irradiation, which may enlighten the design of efficient photothermal-catalytic solar-driven reversible hydrogen storage systems. These exciting properties point to a novel design strategy of solar-driven hydrogen storage technology using metal hydrides appropriate for practical applications.

Fig. R1. a) Completeness ratio over time using solar energy and direct heating with normalized H₂ release capacity of MgH₂ under the catalysis of Cu₁Ni₁ at varying light intensities (or corresponding temperatures). b) H₂ desorption curves of MgH₂ catalyzed by Cu₁Ni₁ using solar energy (light intensity: 3.0 W/cm², corresponding to a temperature of 260 °C) and electrical heating (260 °C), respectively. c) XRD patterns of MgH₂ under the catalysis of Cu₁Ni₁ at different states. d) Cycling H₂ desorption curves of MgH₂ under the catalysis of Cu₁Ni₁ using a light intensity of 3.5 W/cm².

In addition, the operating temperature required for reversible hydrogen storage of MgH₂ using traditional heating method is over 300 °C in general induced by its high thermodynamic stability and kinetic barrier. As a result, over 60% of H₂ stored in MgH₂ would be theoretically consumed in general to drive its reversible H₂ desorption and adsorption reaction. It would inevitably lower the hydrogen storage density and energy efficiency of the whole hydrogen storage system. Unfortunately, this major issue has been ignored for a long time. In order to highlight the important role of solar-driven reversible hydrogen storage of MgH₂, an outdoor experiment of solar-driven H₂ desorption of MgH₂ under the catalysis of Cu₁Ni₁ alloys using natural solar *via* a low-cost PMMA Fresnel lens (Vid. S1 and Fig. S21) are supplementally performed, which further demonstrates the feasibility of H₂ desorption using natural solar energy. A striking emergence of hydrogen bubbles from the pipe could be observed, and XRD

patterns revealed the predominant phase of MgH_2 catalyzed by Cu_1Ni_1 alloys transforms into pure Mg following H_2 desorption process, which offers tangible proof of the potential for utilizing natural solar energy to facilitate solid-state hydrogen storage in MgH_2 . This design could effectively avoid the consumption of hydrogen to provide energy for driving H_2 storage of MgH_2 and hence significantly improve the actual energy density and energy utilization efficiency of metal hydrides as energy source.

The related discussion (highlighted in red color) has been modified in the revised manuscript and supporting information as follows:

[Manuscript - Page 11]

“Although the relatively high light intensity is required to drive the reversible hydrogen storage of MgH_2 even under the catalysis of Cu_1Ni_1 alloys, the low-intensity light (*ca.* 0.5 W/cm^2) could be concentrated to over 3.5 W/cm^2 *via* Fresnel lens (Fig. S21a), which makes it possible to drive reversible hydrogen storage reactions by natural light. To further demonstrates the feasibility of H_2 desorption using natural solar energy, an outdoor experiment of solar-driven H_2 desorption of MgH_2 under the catalysis of Cu_1Ni_1 alloys are performed *via* a low-cost PMMA Fresnel lens. A striking emergence of hydrogen bubbles from the pipe could be observed (Fig. S21b, Vid. S1), and XRD patterns revealed the predominant phase transforms into pure Mg following H_2 desorption process (Fig. S21c), which offers direct evidence to the potential for driving solid-state hydrogen storage of MgH_2 by natural solar energy.”

[Supporting Information - Page 20]

Fig. S21. a) Light intensity of natural solar without and with a commercial PMMA Fresnel lens. b) H_2 desorption experiment using natural solar as the energy source. The inset image shows the enlarged hydrogen bubbles. c) XRD patterns of the sample pellet after H_2 desorption under natural light.

The PMMA Fresnel lens measures $34.5 \text{ cm} \times 34.5 \text{ cm}$, with a thickness of 5 mm and a focal length of 35 cm, which costs around 14 US dollars. The surface has ring-shaped patterns from cold pressing, with a 93% light transmittance.

We sincerely thank the reviewer for all the invaluable comments, which have greatly strengthened our manuscript. We have carefully addressed the concerns point by point as follows.

1. The H₂ capacity comparison under different light intensities irradiation should be given (under the same temperature). In addition, also need to study the effect of temperature on its performance (under the same light intensity).

Response: We thank the reviewer for this valuable comment. First, it is crucial to clarify and supplement that all H₂ absorption/desorption results provided in the manuscript are obtained under only solar irradiation or only external heating, with no simultaneous application of solar irradiation and external heating implemented. That is, there is no additional heat input during the process of solar irradiation, and solar energy serves as the only energy input source, which aligns with our goal of completely replacing the electric heating with solar energy. Therefore, in addition to the intrinsic heat from light itself, under the same light intensity, the differences in surface temperature and H₂ release performance of MgH₂ catalyzed by different catalysts are primarily attributed to the photothermal properties and catalytic performance of the composites.

The H₂ release capacity of the sample (MgH₂ catalyzed by Cu₁Ni₁ alloys) under different light intensities (without external heat applied) could be observed from Fig. 2f in the manuscript. According to the reviewer's suggestion, a bar chart has been added (Fig. S14e) to provide a more intuitive presentation of the influence of varying light intensities on H₂ capacity. For the same sample, the variation in H₂ capacity under different light intensities demonstrates its photothermal performance, excluding the influence of catalytic differences. With the increase in light intensity, the surface temperature of the sample is higher (Fig. S14b), resulting in a more H₂ release capacity with more complete extent (Fig. S14a), as well as a faster release rate (Fig. S14c).

The performance at different temperatures (without solar irradiation applied), including isothermal and TPD dehydrogenation curves, has been presented in Fig. S7 in supporting information, along with comprehensive explanations. The TPD curves demonstrate the catalytic performance of the composites themselves in the absence of photothermal differences (Fig. S7a-c). Based on this, the well-performing MgH₂ under the catalysis of CuNi alloys are selected as the subjects of subsequent solar-driven experiments, with the onset and peak dehydrogenation temperatures roughly determined.

For the comparison of the performance under solar energy and external heating (Fig. 2h, Fig. S15), the surface temperature of the material is initially measured and

determined under solar irradiation, followed by the assessment of its isothermal dehydrogenation performance driven by heat at the same temperature. Despite the different sources and methods of energy supply for both, the hydrogen storage materials reach the same temperature. Therefore, the exhibited hydrogen release rate and rapid response of the sample under solar irradiation reflect the advantages of solar energy as a replacement for external heating (Fig. 2g-h). Further discussion regarding the role of light is addressed in Question 2.

We thank the reviewer's comment and all the above-mentioned results and the related discussion (highlighted in red color) have been provided in the revised manuscript and supporting information. The revised contents are shown as follows:

[Manuscript - Page 6]

“... Therefore, attempting to use inexhaustible solar energy to drive H₂ absorption and desorption of CuNi-catalyzed MgH₂ is a highly promising and environmentally-friendly alternative. To investigate the solar-driven H₂ storage performance of MgH₂, solar energy serves as the only energy input source without external heating, which aligns with our goal of completely replacing the electric heating with solar energy, *via* the apparatus shown in Fig. S11.”

[Supporting Information - Page 7]

Fig. S7. a) TPD curves and b) TPD derivative curves of MgH₂ under the catalysis of Cu₁Ni₁, Cu₁Ni₂, Cu₁Ni₃, Cu₃Ni₁, Ni, and Cu, including pure MgH₂ for comparison. c) TPD curves of MgH₂ under the catalysis of Ni with various ratios, including MgH₂ catalyzed by 10% CuNi and equimolar Cu and Ni for comparison. d) Isothermal dehydrogenation profiles of MgH₂ under the catalysis of Cu₁Ni₁ at various temperatures. e) Isothermal hydrogenation curves at 250 °C under 1 MPa of MgH₂ under the catalysis of Cu₁Ni₁, Cu₁Ni₂, Cu₁Ni₃, Cu₃Ni₁, Ni, and Cu, respectively, including ball-milled MgH₂ for comparison. f) Isothermal hydrogenation curves of MgH₂ under the catalysis of Cu₁Ni₁ at various temperatures.

A series of catalytic performance of MgH₂ under the catalysis of CuNi alloys with different components is first presented and analyzed, which provides a basis on the selection of CuNi alloys with suitable proportions for the subsequent solar-driven experiments.

Fig. S14. a) XRD patterns after H_2 desorption, b) the surface temperature, c) the dehydrogenation instantaneous rate, and d) the fitting of H_2 release rate of MgH_2 under the catalysis of Cu_1Ni_1 using different light intensities. e) Comparison of the amount of H_2 desorption capacity and f) activation energies of MgH_2 under the catalysis of Cu_1Ni_1 using solar energy under different light intensities and thermal heating at different temperatures.

2. In Figure 2d and 2e, the author compared the performance differences between photothermal and thermal, and did not provide a reasonable explanation for why photothermal performance was higher than thermal. What is the role of light in this work.

Response: We would appreciate the reviewer's constructive comment. As mentioned in the original manuscript, CuNi alloys, evenly distributed on the MgH₂ matrix, could generate localized heat under solar irradiation that could directly act on the catalytic center that is itself while the poor thermal conductivity of "MgH₂" could reduce the temperature gradient near CuNi alloys, which is capable of promoting rapid H₂ desorption from MgH₂. Therefore, we believe that light is primarily converted from photon into heat through CuNi alloys, followed by non-radiative relaxation into localized heat. Attributed to the direct action of localized heat generated by the dispersed CuNi alloy on the catalytic sites, superior initial H₂ desorption performance is observed than that driven by traditional heating.

To further investigate the role of light and the photothermal-activated mechanism for H₂ release from MgH₂, the fitting between light intensity and dehydrogenation rate is supplemented. The dehydrogenation instantaneous rate peaks within the initial 100 seconds under solar irradiation, and the relationship between initial H₂ release rates and light intensities is exponential (Fig. S14c-d), consistent with the Arrhenius empirical equation, which suggests that the photothermal effect primarily operates as a plasma heating. Comparing the amount of H₂ desorption at various temperatures and light intensities (Fig. S14e), the H₂ desorption performance under solar energy is similar to that driven by thermal heating, that is, H₂ release from MgH₂ through a photothermal pathway induced by the light-to-heat conversion ability of CuNi alloys rather than photocatalytic principle. In addition, the apparent activation energy under solar irradiation fitted by the Johnson–Mehl–Avrami (JMA) equation joint with the Arrhenius equation is calculated to be 81.23 ± 3.35 kJ/mol, which is comparable to that driven by thermal heating (i.e., 81.56 ± 2.00 kJ/mol) (Fig. S14f).

These results validate that the solar-driven dehydrogenation of MgH₂ is mainly achieved based on the localized heat produced by photothermal effect of CuNi alloys with the combination of its catalytic effect. Consequently, light is converted by CuNi alloys and the regenerated Mg₂Ni(Cu), which concurrently serve as light absorbers and thermochemical catalysts, into localized heat that directly influences catalytic sites, which is the reason for why photothermal performance is higher than thermal one.

We appreciate the reviewer's comment and the related discussion (highlighted in red color) has been modified in the revised manuscript and supporting information as follows:

[Manuscript - Page 7]

“Upon the elevation of the irradiation intensity, the surface temperature of MgH₂ catalyzed by Cu₁Ni₁ would increase, which is capable of promoting the degree and rate of H₂ desorption from MgH₂ (Fig. 2f, Fig. S14a-b). **The dehydrogenation instantaneous rate peaks within the initial 100 seconds under solar irradiation, and the relationship between initial H₂ release rates**

and light intensities is exponential (Fig. S14c-d), consistent with the Arrhenius empirical equation^{14, 15}, which suggests that the photothermal effect primarily operates as a plasma heating. Comparing the amount of H₂ desorption at various temperatures and light intensities (Fig. S14e), the H₂ desorption performance under solar energy is similar to that driven by thermal heating, that is, H₂ release from MgH₂ through a photothermal pathway induced by the light-to-heat conversion ability of CuNi alloys rather than photocatalytic principle¹⁶. In addition, the apparent activation energy under solar irradiation fitted by the Johnson–Mehl–Avrami (JMA) equation joint with the Arrhenius equation is calculated to be 81.23 ± 3.35 kJ/mol, which is comparable to that driven by thermal heating (*i.e.*, 81.56 ± 2.00 kJ/mol) (Fig. S14f).”

[Supporting Information - Page 15]

Fig. S15. a) XRD patterns after H₂ desorption, b) the surface temperature, c) the dehydrogenation

instantaneous rate, and d) the fitting of H₂ release rate of MgH₂ under the catalysis of Cu₁Ni₁ using different light intensities. e) Comparison of the amount of H₂ desorption capacity and f) activation energies of MgH₂ under the catalysis of Cu₁Ni₁ using solar energy under different light intensities and thermal heating at different temperatures.

The activation energy of MgH₂ under the catalysis of Cu₁Ni₁ alloys under solar irradiation and heat is also fitted by the Johnson–Mehl–Avrami (JMA) equation joint with the Arrhenius equation. The JMA equation is expressed as:

$$\ln[-\ln(1 - \alpha)] = n \ln k + n \ln t \quad (\text{Equ. S3})$$

where α is the fraction of Mg transformed into MgH₂ at reaction time t , and n and k represent the Avrami exponent and effective kinetic parameter, respectively. The values of n and k can be deduced based on the isothermal kinetics data at different temperatures. And the E_a value could be calculated from the Arrhenius equation, written as:

$$\ln k = -E_a/RT + \ln A \quad (\text{Equ. S4})$$

where T is the temperature of isothermal driven by heat or solar-driven hydrogen desorption and A is the pre-exponential factor.

3. How did the author control the temperature during the light irradiation process? A detailed explanation was needed.

Response: We would appreciate the reviewer's careful review of our manuscript. We have added the photograph of the solar-driven H₂ absorption and desorption apparatus used for this work in the supporting information (Fig.S11), along with a detailed description provided in the "Hydrogen Storage Measurement" section of Method in the manuscript. The light is incident through the above sapphire window with a thickness of approximately 5 cm, and the outlet of the reactor is connected to the PCT equipment. For temperature testing, the surface temperature during the H₂ absorption and desorption processes of the sample is continuously monitored in real-time *via* a short-wave infrared thermometer, which could receive the signal of infrared radiation from the sample's surface through the sapphire, and subsequently converts it into temperature data.

In terms of temperature control, as addressed in Question 1, there is no additional external heat input during the process of solar irradiation, that is, solar energy serves as the only energy source. Therefore, in addition to the intrinsic heat from light itself, under the same light intensity, the differences in surface temperature of MgH₂ catalyzed by different catalysts are primarily attributed to the photothermal properties of the composites. The bottom of the reactor is set by white glass fibers with low thermal conductivity, which have limited absorption of solar energy, while the dark-colored sample pellet seems as an internal heat sources within the reactor under solar

irradiation.

For the comparison of the performance under solar energy and external heating, the surface temperature of the material is initially measured and determined under solar irradiation, followed by the assessment of its isothermal dehydrogenation performance driven by heat at the same temperature. Despite the different sources and methods of energy supply for both, the hydrogen storage materials reach the same temperature.

Furthermore, the expansion of internal gases within the reactor due to solar irradiation has a negligible impact on the H₂ capacity results, as a large chamber, 25 times the volume of the reactor, is opened during the dehydrogenation tests to minimize testing errors.

We appreciate the reviewer's comment and the related discussion (highlighted in red color), have been provided in the revised manuscript and supporting information as follows:

[Manuscript - Page 6]

"... Therefore, attempting to use inexhaustible solar energy to drive H₂ absorption and desorption of CuNi-catalyzed MgH₂ is a highly promising and environmentally-friendly alternative. To investigate the solar-driven H₂ storage performance of MgH₂, solar energy serves as the only energy input source without external heating, which aligns with our goal of completely replacing the electric heating with solar energy, via the apparatus shown in Fig. S11."

[Manuscript - Page 18]

As for light irradiation, customized solar-driven H₂ absorption and desorption equipment (Fig.S11) is used to perform experiments under solar energy. The adjustment of the light intensity is realized by changing the current intensity. The light is incident through the above sapphire window, and the outlet of the reactor is connected to the PCT equipment. The temperature of the material during the H₂ absorption and desorption is measured by a short-wave infrared thermometer. About 30 mg samples are taken each time and pressed into pieces with a diameter of 15 mm under a pressure of 5 tons, to ensure the stability of the test and the comparability of the results. Glass fiber sheets made up of mainly SiO₂ with low thermal conductivity, which have limited absorption of solar energy, are placed at the bottom of the reactor to reduce the heat dissipation of the material to the outside. The sample is placed on an Al₂O₃ ceramic sheet, which could be isolated from the glass fiber to prevent the sample from oxidizing by reacting with the insulation layer due to the high temperature during the reaction. The expansion of internal gases within the reactor due to solar irradiation has a negligible impact on the H₂ capacity results, as a large chamber, 25 times the volume of the reactor, is opened during the dehydrogenation tests to minimize testing errors. The hydrogen pressure used for H₂ absorption under solar irradiation is consistent with that using electric

heating.

[Supporting Information - Page 11]

Fig. S11. The photograph of solar-driven H₂ absorption and desorption apparatus.

For temperature testing, the surface temperature during the H₂ absorption and desorption processes of the sample is continuously monitored in real-time *via* a short-wave infrared thermometer, which could receive the signal of infrared radiation from the sample's surface through the sapphire, and subsequently converts it into temperature data.

Another detailed description for the hydrogen storage measurement and sample preparation is provided in the "Hydrogen Storage Measurement" section of Method.

4. In Figure 1, the current TEM data cannot prove that CuNi is an alloy structure, and the author needs to provide clearer evidence.

Response: We would appreciate the reviewer's careful review of our manuscript. In response to this comment, we have reconducted TEM measurements for Cu₁Ni₁ alloys and replaced Fig. 1e and f with clearer images. Additionally, we have included more TEM images of different regions in Fig. S3c-d for further clarity. The lattice spacing inside of Cu₁Ni₁ is calculated to be 0.206 nm, corresponding to the (111) crystal plane of Cu₁Ni₁ (Fig. 1e). The selected area electron diffraction (SAED) pattern confirms the polycrystal structure of Cu₁Ni₁ alloys and the diffraction rings could be well recorded to (111), (200), and (220) crystal planes (Fig. 1f). Additionally, the lattice spacing measured from HRTEM images presents a lattice spacing (0.206 nm) of Cu₁Ni₁ (111) that is between the lattice spacing (0.210 nm) of Cu (111) and the lattice spacing (0.203 nm) of Ni (111) with a periodic pattern (Fig. S4a-d), implying the successful synthesis

of Cu₁Ni₁ alloys. Furthermore, energy dispersive X-ray spectroscopy (EDS) elemental mapping (Fig. 1h-j) result confirms the homogeneous distribution of Cu and Ni, which coincides well with the morphology of particles, further validating the uniform synthesis of CuNi alloys. These TEM results corroborate with the XRD patterns (Fig. 1a), where the diffraction peaks of Cu₁Ni₁ are located at 43.86°, 51.09°, and 75.17°, respectively, matching the standard diffraction peaks of Cu_{0.5}Ni_{0.5} (PDF#04-002-1345) ideally. These results provide strong evidence to demonstrate that Cu₁Ni₁ exists as alloys.

Moreover, the distribution of electrons investigated *via* X-ray photoelectron spectroscopy (XPS) survey spectra provides further evidence to the formation of an alloy structure. XPS spectra of Cu and Ni alloys are supplementally performed for comparison to observe the transfer of electrons in CuNi alloys. High-resolution Cu 2p and Ni 2p XPS spectra (Fig. 1b-c, Fig. S2c-d) demonstrate that Cu and Ni are present mainly in their zero-valent state, consistent with the XRD results. Particularly, the binding energies of Cu⁰ and Ni⁰ in CuNi alloys exhibit negative (Δ_{\max} , Cu = -0.35 eV) and positive (Δ_{\max} , Ni = 0.30 eV) shifts compared to pure Cu and Ni, respectively, indicating the transfer of electrons from Ni atoms to Cu atoms as normally observed in CuNi alloys, which provides another evidence to the alloy structure of CuNi alloys.

We appreciate the reviewer's comment and all the above-mentioned results and the related discussion (highlighted in red color) have been provided in the revised manuscript and supporting information as follows:

[Manuscript - Page 5]

“..., thereby confirming the successful synthesis of Cu₁Ni₁ alloys. Moreover, the presence of Cu and Ni in CuNi alloys is confirmed in X-ray photoelectron spectroscopy (XPS) survey spectra (Fig. S2a). High-resolution Cu 2p and Ni 2p XPS spectra (Fig. 1b-c, Fig. S2c-d) demonstrate that Cu and Ni are present mainly in their zero-valent state, consistent with the XRD results. Particularly, the binding energies of Cu⁰ and Ni⁰ in CuNi alloys exhibit negative (Δ_{\max} , Cu = -0.35 eV) and positive (Δ_{\max} , Ni = 0.30 eV) shifts compared to pure Cu and Ni^{17, 18}, respectively, indicating the transfer of electrons from Ni atoms to Cu atoms in CuNi alloys^{19, 20}. The weak peaks of Cu²⁺ and Ni²⁺ could be attributed to the slight oxidation of CuNi alloys during measurement (Tab. S2) and no chemical interaction between the outer carbon layer and CuNi alloys could be observed in the high-resolution C 1s XPS spectrum (Fig. S2b).”

[Manuscript - Page 5]

“Scanning electron microscopy (SEM) image (Fig. 1d) illustrates that Cu₁Ni₁ is composed of densely packed spherical particles with an average particle size of approximately 20 nm (Fig. 1g). High-resolution transmission electron microscopy (HRTEM) images (Fig. 1e, Fig. S3) confirm that Cu₁Ni₁ alloys are wrapped by a carbon layer of 3 nm with a weight percent of approximately 9% as evidenced by thermogravimetric analysis (TGA) (Fig. S5a). The lattice

spacing inside of Cu_1Ni_1 is calculated to be 0.206 nm, corresponding to the (111) crystal plane of Cu_1Ni_1 . The selected area electron diffraction (SAED) pattern in Fig.1f confirms the polycrystal structure of Cu_1Ni_1 alloys and the diffraction rings could be well recorded to (111), (200) and (220) crystal planes. Additionally, the lattice spacing measured from HRTEM images (Fig. S4a-d) presents a lattice spacing (0.206 nm) of Cu_1Ni_1 (111) that is between the lattice spacing (0.210 nm) of Cu (111) and the lattice spacing (0.203 nm) of Ni (111) with a periodic pattern, implying the successful synthesis of Cu_1Ni_1 alloys wrapped with ultrathin carbon layers. Furthermore, energy dispersive X-ray spectroscopy (EDS) elemental mapping (Fig. 1h-j) result confirms the homogeneous distribution of Cu and Ni, which coincides well with the morphology of particles, further validating the uniform synthesis of CuNi alloys.”

[Manuscript - Page 25]

Fig. 1. a) XRD patterns of the as-synthesized of CuNi alloys, including Ni and Cu for comparison. High-resolution b) Cu 2p and c) Ni 2p XPS spectrum of Cu_1Ni_1 , with Cu and Ni for comparison. d) SEM and e) HRTEM image and f) SAED pattern of Cu_1Ni_1 . g) the corresponding particle size distribution. h-j) EDS elemental mapping images of Cu_1Ni_1 alloys.

[Supporting Information - Page 3]

Fig. S2. a) XPS survey spectra and b) high-resolution C 1s XPS spectrum of CuNi alloys, including Cu and Ni for comparison. High-resolution c) Cu 2p and d) Ni 2p XPS spectrum of Cu₁Ni₂ and Cu₁Ni₃, including Cu and Ni for comparison.

[Supporting Information - Page 4]

Fig. S3. TEM images of Cu_1Ni_1 alloys.

[Supporting Information - Page 4]

Fig. S4. TEM images and lattice spacing of a-b) Cu and c-d) Ni nanoparticles.

5. The author synthesized CuNi alloy materials with different components (Cu_1Ni_1 and Cu_1Ni_3). How about Cu_1Ni_2 .

Response: We thank the reviewer for this constructive comment. In the original manuscript, we only presented the characterization results for Cu_1Ni_1 (Cu: Ni atomic ratio of 1:1) and Cu_1Ni_3 alloys. Actually, CuNi alloys with controllable atomic ratios could be successfully synthesized by easily adjusting the precursor ratios. As a result, more CuNi alloys with different Cu/Ni components are obtained based on the same synthesis method, such as Cu_1Ni_2 and Cu_3Ni_1 , as XRD patterns shown (Fig. 1a). Attributed to the slight change of crystal plane spacing caused by various Cu/Ni molar ratio, the diffraction peaks of CuNi alloy site between the diffraction peaks of Cu and Ni, and show shifts from Ni to Cu with the increase of Cu/Ni ratio. Especially, as for Cu_1Ni_2 alloys you mentioned, ICP-OES analysis has indicated that the atomic ratio of Cu to Ni in Cu_1Ni_2 is 1:2.01 (Tab. S1). The morphology (Fig. S6a), particle size (Fig. S6b), and carbon content (8.4%) (Fig. S5b) of Cu_1Ni_2 are also similar to those of Cu_1Ni_1 and Cu_1Ni_3 . Therefore, from the perspective of the synthesis of alloys, Cu_1Ni_2 could be successfully prepared.

It has been demonstrated that both the photothermal and catalytic effect of MgH_2 composites are crucial for achieving solar-driven hydrogen storage. A series of catalytic performance of MgH_2 under the catalysis of CuNi alloys with different components is first presented and analyzed in the original supporting information, which provides a basis on the selection of CuNi alloys with suitable proportions for the subsequent solar-driven experiments. In accordance with the reviewer's suggestion, we further supplemented the TPD dehydrogenation and isothermal hydrogenation data for Cu_1Ni_2 and Cu_3Ni_1 . Among them, Cu_1Ni_2 exhibits quite similar heat-driven performance in both dehydrogenation and hydrogenation compared to Cu_1Ni_1 and Cu_1Ni_3 . TPD curves and their differential curves (Fig. S7a-b) show that the introduction of 10% Cu_1Ni_1 , Cu_1Ni_2 , and Cu_1Ni_3 alloys lowers the onset temperature of H_2 desorption from MgH_2 to below $210\text{ }^\circ\text{C}$, and the H_2 desorption peak temperature is reduced to about $245\text{ }^\circ\text{C}$, which is more than $80\text{ }^\circ\text{C}$ lower than the H_2 release temperature and peak temperature of ball-milled MgH_2 , respectively. Mg catalyzed by Cu_1Ni_1 , Cu_1Ni_2 , and Cu_1Ni_3 absorb 6.1 wt.% H_2 within only 10 minutes after dehydrogenation, which is more than 95% of their maximum absorption capacity (Fig. S7e). However, in the case of MgH_2 catalyzed by Cu_3Ni_1 , the dehydrogenation peak temperature rises to $266\text{ }^\circ\text{C}$, and the hydrogenation rate is also relatively slower. This indicates that if the Cu content in CuNi alloys exceeds 50%, it may lead to a deterioration in the catalytic effect, which is not conducive to achieving highly efficient solar-driven hydrogen storage. Therefore, Cu_3Ni_1 alloy is excluded and not further discussed in subsequent experiments.

Furthermore, with regard to the photothermal performance, according to the reviewer's suggestions, we added Cu_1Ni_2 into the comparison of the color and UV-vis-NIR absorption spectra, and their performance still adheres to the pattern (Fig.

2a-b). Specifically, an obvious color difference could be observed under the catalysis of CuNi alloys with different components, and it becomes darker and finally dark-black after the addition of CuNi alloys with the increasing Cu content. Additionally, the UV-vis-NIR absorption spectra are employed to provide the overall absorbance intensity of MgH₂ catalyzed by Cu₁Ni₁, Cu₁Ni₂, Cu₁Ni₃, Cu, and Ni, indicating that the absorbance of MgH₂ increases with the increasing Cu content in CuNi alloys. Particularly, the color and the overall absorbance capability of Cu₁Ni₂ fall between those of Cu₁Ni₃ and Cu₁Ni₁. Therefore, with the consideration of maximizing the differences brought by photothermal effects under similar catalytic performance, Cu₁Ni₁ and Cu₁Ni₃ are selected as representatives in the original manuscript to provide direct evidence to the crucial role of photothermal effect in enhancing solar-driven hydrogen storage of MgH₂ (Fig. 2d-e).

We fully agree with the reviewer's suggestion and believe that Cu₁Ni₂ further validate our proposed design strategy of *in-situ* alloying to form a single-component phase of Mg₂Ni(Cu) ternary alloy *via* atomic reconstruction to achieve the ideal integration of excellent photothermal and catalytic effects for solar-driven hydrogen storage of MgH₂. We have supplemented the response of temperature with corresponding H₂ desorption curves for MgH₂ under the catalysis of Cu₁Ni₂ before and after cycling under 2.6 W/cm² (Fig. S18a). Similar phenomena of the increase of surface temperature (corresponding to the increase of 10.1 °C in absolute temperature from 234.1 °C to 242.8 °C) (Fig. 3c) and the acceleration of H₂ desorption with more H₂ capacity could also be observed for MgH₂ under the catalysis of Cu₁Ni₂ after cycling , which could be also attributed to the increased overall absorption in the near-infrared range (Fig. S18f). As for the mechanism, the characteristic peaks of Mg₂Ni(Cu) between Mg₂Ni and Mg₂Cu could be clearly observed with the presence of a weak peak at around 38~39°, which provides additional evidence to the formation of Mg₂Ni(Cu) during the initial dehydrogenation of MgH₂ catalyzed by Cu₁Ni₂ (Fig. S22d-e), consistent with the mechanism of MgH₂ catalyzed by Cu₁Ni₁ alloys.

We appreciate the reviewer's comment and all the above-mentioned results and the related discussion (highlighted in red color) have been provided in the revised manuscript as follows:

[Manuscript - Page 4]

“... By changing the appropriate amount of the precursors, the ratio of Cu and Ni atoms in the thus-obtained alloy could be facile controlled, resulting in the synthesis of Cu₁Ni_x (Cu: Ni atomic ratio of 1:x), Cu, and Ni, respectively. Inductively coupled plasma-optical emission spectroscopy (ICP-OES) indicated that the atomic ratios of Cu₁Ni₁, Cu₁Ni₂, and Cu₁Ni₃ are 1:1.03, 1:2.01, and 1:3.01 (Tab. S1), respectively, which demonstrates the successful

synthesis of CuNi alloys with controllable atomic ratios by facile adjusting the ratio of precursors.

According to X-ray diffraction (XRD) results (Fig. 1a), all the as-synthesized catalysts exhibit three distinct diffraction peaks, indicating the formation of typical face-centered-cubic (FCC) crystal phase with (111), (200), and (220) planes.”

[Manuscript - Page 5]

“... In addition, the morphology, particle size (Fig. S6a-f), and carbon content (Fig. S5b-c) of Cu_1Ni_2 , Cu_1Ni_3 , and Ni are similar to those of Cu_1Ni_1 , while the particle size of Cu is approximately 50 nm (Fig. S6g-h).”

[Manuscript - Page 6]

“After the ball-milling process of MgH_2 with the addition of catalysts with a weight percent of 10%, an obvious color difference could be observed (Fig. 2a). Pure MgH_2 is essentially white and it turns into yellowish-brown, and becomes darker and finally dark-black after the addition of Ni, CuNi alloys with the increasing Cu content, and Cu respectively, indicating the increase of light absorption potential. It could be directly supported by the ultraviolet-visible-near-infrared (UV-vis-NIR) absorption spectra (Fig. 2b), which demonstrates that MgH_2 catalyzed by Cu has the highest overall absorbance capability and the absorbance of MgH_2 after the addition of CuNi alloys increases as the relative Cu content in these catalysts increases, where Cu_1Ni_2 fall between those of Cu_1Ni_3 and Cu_1Ni_1 Herein, with the consideration of maximizing the differences brought by photothermal effects under similar catalytic performance, Cu_1Ni_1 and Cu_1Ni_3 are selected as representatives in the following. Correspondingly, under the same intensity (2.5 W/cm^2 and 3.0 W/cm^2) of solar irradiation, ...”

[Manuscript - Page 10]

“..., after the initial hydrogenation process, the color of MgH_2 catalyzed with Cu_1Ni_1 , Cu_1Ni_2 , and Cu_1Ni_3 turns deep black, similar to that with Cu, while MgH_2 catalyzed by Ni remains yellowish-brown (Fig. 3a inset).”

“The ball-milled MgH_2 catalyzed by Cu_1Ni_1 releases 5.39 wt.% H_2 within 30 minutes, while the time required to release the same H_2 capacity could be shortened to 25 minutes with more H_2 capacity for MgH_2 catalyzed by Cu_1Ni_1 after cycling (Fig. 3d, Fig. S18f). Similar phenomena of the increase of surface temperature (with a 3.7% increase) and the acceleration of H_2 desorption could also be observed for MgH_2 under the catalysis of Cu_1Ni_2 and Cu_1Ni_3 after cycling under the identical light intensity (Fig. S18a-b).”

[Manuscript - Page 25]

Fig. 1. a) XRD patterns of the as-synthesized of CuNi alloys, including Ni and Cu for comparison. High-resolution b) Cu 2p and c) Ni 2p XPS spectrum of Cu₁Ni₁, with Cu and Ni for comparison. d) SEM and e) HRTEM image and f) SAED pattern of Cu₁Ni₁. g) the corresponding particle size distribution. h-j) EDS elemental mapping images of Cu₁Ni₁ alloys.

[Manuscript - Page 26]

Fig. 2. a) The color, b) UV-vis-NIR absorption spectra, c) the response of temperature to light irradiation, d) the corresponding solar-driven H₂ desorption curves of MgH₂, and e) heat-driven isothermal dehydrogenation curves at 250 °C of MgH₂ under the catalysis of CuNi alloys, Ni, and Cu, respectively, including ball-milled MgH₂ (BM MgH₂) for comparison. f) H₂ desorption curves of MgH₂ under the catalysis of Cu₁Ni₁ under different light intensities. g) Completeness ratio over time using solar energy and direct heating with normalized H₂ release capacity of MgH₂ under the catalysis of Cu₁Ni₁ at varying light intensities (or corresponding temperatures). h) H₂ desorption curves of MgH₂ catalyzed by Cu₁Ni₁ using solar energy (light intensity: 2.8 W/cm², corresponding to a temperature of 250 °C) and electrical heating (250 °C), respectively.

[Manuscript - Page 27]

Fig. 3. a) UV-vis-NIR absorption spectra of MgH₂ under the catalysis of Cu₁Ni₁, Cu₁Ni₂, Cu₁Ni₃, Ni, and Cu after rehydrogenation. The inset shows the color of their cycled state. b) UV-vis-NIR absorption spectra of MgH₂ under the catalysis of Cu₁Ni₁, Cu₁Ni₂, and Cu₁Ni₃ after cycling, in comparison with their pristine ball-milled state (before cycling). The inset exhibits solar irradiation distribution of natural sunlight. c) The response of temperature to light irradiation of MgH₂ after cycling under the catalysis of Cu₁Ni₁, Cu₁Ni₂, Cu₁Ni₃, Ni, and Cu, respectively, with their pristine ball-milled state (before cycling) included for comparison. d) The response of temperature with corresponding H₂ desorption curves of MgH₂ under the catalysis of Cu₁Ni₁ under 2.6 W/cm² before and after cycling. e) Cycling H₂ desorption curves of MgH₂ under the catalysis of Cu₁Ni₁ using a light intensity of 3.5 W/cm².

[Supporting Information - Page 2]

Tab. S1. Elemental analysis of the CuNi alloys.

Sample name	Molar ratio of Cu and Ni salts	Composition (Cu:Ni) (w/w) ^a	Atomic ratio
Cu ₁ Ni ₁	1:1	46.51: 43.88	1: 1.03
Cu ₁ Ni ₂	1:2	31.56: 58.22	1: 2.01
Cu ₁ Ni ₃	1:3	23.89: 64.48	1: 3.01

^a Composition is measured by ICP-OES technique.

[Supporting Information - Page 3]

Fig. S2. a) XPS survey spectra and b) high-resolution C 1s XPS spectrum of CuNi alloys, including Cu and Ni for comparison. High-resolution c) Cu 2p and d) Ni 2p XPS spectrum of Cu₁Ni₂ and Cu₁Ni₃, including Cu and Ni for comparison.

[Supporting Information - Page 5]

Fig. S5. Thermogravimetric analysis (TGA) of a) Cu₁Ni₁, b) Cu₁Ni₂, and c) Cu₁Ni₃ alloys.

Similarly, it could be inferred that the content of CuNi metal in CuNi₂ and Cu₁Ni₃ alloy is 91% and 89%, respectively, and the carbon content is 8.4% and 10.6%, respectively.

[Supporting Information - Page 6]

Fig. S6. SEM images and the corresponding particle size distributions of **a-b) Cu_1Ni_2** , **c-d) Cu_1Ni_3** , **e-f) Ni**, and **g-h) Cu**.

[Supporting Information - Page 7]

Fig. S7. a) TPD curves and b) TPD derivative curves of MgH₂ under the catalysis of Cu₁Ni₁, Cu₁Ni₂, Cu₁Ni₃, Cu₃Ni₁, Ni, and Cu, including pure MgH₂ for comparison. c) TPD curves of MgH₂ under the catalysis of Ni with various ratios, including MgH₂ catalyzed by 10% CuNi and equimolar Cu and Ni for comparison. d) Isothermal dehydrogenation profiles of MgH₂ under the catalysis of Cu₁Ni₁ at various temperatures. e) Isothermal hydrogenation curves at 250 °C under 1 MPa of MgH₂ under the catalysis of Cu₁Ni₁, Cu₁Ni₂, Cu₁Ni₃, Cu₃Ni₁, Ni, and Cu, respectively, including ball-milled MgH₂ for comparison. f) Isothermal hydrogenation curves of MgH₂ under the catalysis of Cu₁Ni₁ at various temperatures.

A series of catalytic performance of MgH₂ under the catalysis of CuNi alloys with different components is first presented and analyzed, which provides a basis on the selection of CuNi alloys with suitable proportions for the subsequent solar-driven experiments.

Temperature-programmed desorption (TPD) curves and their differential curves (Fig. S7a-b) show that the introduction of 10% Cu₁Ni₁, Cu₁Ni₂, and Cu₁Ni₃ alloys lowers the onset temperature of H₂ desorption from MgH₂ to below 210 °C, and the H₂ desorption peak temperature is reduced to about 245 °C, which is more than 80 °C lower than the H₂ release temperature and peak temperature of ball-milled MgH₂ respectively. However, the

dehydrogenation peak temperature of MgH_2 catalyzed by Cu_3Ni_1 rises to 266°C with the Cu content in CuNi alloys exceeding 50%.

...However, the addition of Ni-based catalysts significantly improves the H_2 absorption kinetics of Mg. Mg catalyzed by Cu_1Ni_1 , Cu_1Ni_2 , and Cu_1Ni_3 absorb 6.1 wt.% H_2 in just 10 minutes after dehydrogenation, which is more than 95% of their maximum absorption capacity (Fig. S7e). It is worth noting that under the catalysis of Cu_1Ni_1 , the sample could reversibly store 6.0 wt.% H_2 in Mg within 60 minutes at 150°C (Fig. S7f), presenting the effective catalytic effect of CuNi alloys in improving the H_2 absorption kinetics of MgH_2 . However, the hydrogenation rate is also relatively slower in the case of MgH_2 catalyzed by Cu_3Ni_1 , leading to a deterioration in the catalytic effect, which is not conducive to achieving highly efficient solar-driven hydrogen storage. Therefore, Cu_3Ni_1 alloy is excluded and not further discussed in subsequent experiments.

[Supporting Information - Page 19]

Fig. S18. The response of temperature with corresponding H_2 desorption curves for MgH_2 under the catalysis before and after cycling of a) Cu_1Ni_2 , b) Cu_1Ni_3 , c) Ni, and d) Cu under 2.6 W/cm^2 ,

and e) Cu_1Ni_1 under 3.5 W/cm^2 . f) The H_2 capacity to light irradiation of MgH_2 after cycling under the catalysis of Cu_1Ni_1 , Cu_1Ni_2 , Cu_1Ni_3 , Ni, and Cu, respectively, with their pristine ball-milled state (before cycling) included for comparison.

[Supporting Information - Page 21]

Fig. S22. a) XRD patterns of MgH_2 under the catalysis of 10% Cu_1Ni_1 at different states. XRD patterns of MgH_2 b) after ball-milling, c) after rehydrogenation, and after dehydrogenation under the catalysis of Cu_1Ni_1 , Cu_1Ni_2 , Cu_1Ni_3 , Ni, and Cu, in the 2θ range of d) $20\sim 80$ degrees and e) an enlarged view of $38\sim 46$ degrees. XRD patterns of MgH_2 after dehydrogenation under the catalysis of equimolar Cu and Ni, in the 2θ range of f) $20\sim 80$ degrees and g) an enlarged view of $38\sim 46$ degrees, including Cu_1Ni_1 and Ni for comparison.

6. The author claims that it is the LSPR effect of CuNi alloy, but does not provide sufficient evidence. In Figure 2b, there is no characteristic absorption peak of Cu or CuNi.

Response: Thanks for the helpful suggestion. Due to the low concentration (only 10%) of Cu and CuNi alloy in MgH_2 , the LSPR peaks in Fig. 2b on that original wide coordinate scale is not sufficiently shown. We further zoomed in the UV-vis absorption spectra of MgH_2 catalyzed by Cu and Cu_1Ni_1 in the range of $400\text{-}800 \text{ nm}$, where characteristic peak due to the LSPR effect of Cu could be observed clearly (Fig. S12a).

LSPR is the collective resonant oscillation of conduction electrons of metallic nanoparticle in response to the electric field of incident light, and these resonances, associated with noble metal nanostructures, create sharp spectral absorption and scattering peaks as well as strong electromagnetic near-field enhancements²¹. To

further validate the LSPR effect of Cu and CuNi alloy, we conducted finite-difference time-domain (FDTD) simulations to investigate the radiation energy transformation of Cu, CuNi, and Ni. The FDTD simulation is performed by FDTD Package (Lumerical Solution, Inc.). Metal nanoparticles are set with a diameter of 20 nm. The simulation area is divided into uniform Yee cells with perfectly matched layer boundary condition²². An incident plane wave is propagated from the top along z axis and polarized along the x axis, and a wavelength of 580 nm light is used for simulation. The process is performed at 1 nm mesh resolution and 1000 fs time.

The dielectric constants of Cu and Ni nanoparticles are assumed to be the same as that of the bulk metal and were obtained from Johnson and Christy²³. The dielectric constant (ϵ) of CuNi alloy is rarely available and hence, to describe the plasmonic properties, the dielectric constant of the alloys is calculated by the composition-weighted^{24, 25} by $\epsilon_{alloy}(x_{Ni}, \omega) = x_{Ni}\epsilon_{Ni}(\omega) + (1 - x_{Ni})\epsilon_{Cu}(\omega)$, where x_{Ni} is the Ni volume fraction, as shown in Fig. S12c-d.

For Cu nanoparticles, plasmonic "hot spot" regions could be clearly observed around the particles (Fig. S12e), in contrast, Ni do not exhibit distinct "hot spot" regions near the periphery of the particles (Fig. S12g), which is related to the intrinsic electronic structure and optical properties of different metals²⁶. After forming CuNi alloy, ring-shaped "hot spot" regions could be also observed near the periphery of alloy particles, but with an intensity weaker than that of Cu (Fig. S12f). FDTD results validate the strong LSPR-induced localized electric field of Cu and CuNi, confirming the characteristic absorption peaks of Cu and CuNi in the absorption spectrum.

Furthermore, many studies have also pointed out that the introduction of LSPR in metal particles leads to an overall increase in absorbance^{27, 28}. This may be attributed to the fact that when the LSPR effect occurs, the free electrons in the metal nanoparticles undergo surface oscillations, causing a change in the electron density distribution near the surface. As a result, the effective optical cross-section of the metal particles increases, and metal nanoparticles could absorb more photons near the resonance frequency, thereby enhancing the overall absorbance^{21, 29}. This explains the enhanced overall absorption in the visible and NIR regions observed after the introduction of CuNi and Cu into the MgH₂ system, which increases with higher Cu content (Fig. 2b). When multi-metal nanoparticles are in close proximity, their LSPR effects couple with each other and form resonant states, leading to the scattering of the NIR by light interference with a more intense localized electric field and the increase of the overall absorbance³⁰. This is also evident in the improvement of absorption in the NIR region when increasing the catalyst content to 20% (Fig. S12b). Therefore, we believe that, after ball milling, the LSPR with the coupling effect of Cu and CuNi alloys are the primary factors for the overall enhancement of the absorbance of MgH₂ composites.

In addition, according to the comments 3 and 4 of Reviewer 3, we unravel the mechanism of CuNi alloying in improving photothermal effect of the composite after cycling based on the theoretical calculation of band structure of $\text{Mg}_2\text{Ni}(\text{Cu})\text{H}_4$ and Mg_2NiH_4 . Please refer to our detailed response to the comments 3 and 4 of Reviewer 3.

We appreciate the reviewer's comment and all the above-mentioned results and the related discussion (highlighted in red color) have been provided in the revised manuscript and supporting information as follows:

[Manuscript - Page 6]

“... Notably, MgH_2 under the catalysis of Cu exhibits a characteristic absorption peak from 400-800 nm due to the LSPR effect of Cu nanoparticles (Fig. S12a)³¹. By comparison, a slight red-shift of the LSPR characteristic peak of Cu is observed for MgH_2 catalyzed by Cu_1Ni_1 owing to the size changes of nanocrystals³². The FDTD simulations further demonstrate that plasmonic “hot spot” regions could be clearly observed around particles in both Cu and CuNi alloys, which validates the strong LSPR-induced localized electric-field of Cu and CuNi (Fig. S12e-g). Additionally, the relatively weaker absorption observed in the NIR region also increases with higher Cu content, which could be attributed to the scattering effect induced by the LSPR coupling of Cu particles (Fig. 2b, Fig. S12b)^{21, 29, 33}.”

[Supporting Information - Page 12]

Fig. S12. UV–vis–NIR absorption spectra of MgH₂ under the catalysis of a) Cu₁Ni₁ and Cu in the range of 400–800 nm and b) Cu₁Ni₁ with addition of 10% and 20%. c) Real and d) imaginary parts of dielectric functions of Cu, Cu₁Ni₁, and Ni metals. The dotted line represents the case where $\epsilon_r \approx -2\epsilon_m$, air. FDTD simulated localized electric field enhancement profiles of e) Cu , f) Cu₁Ni₁, and g) Ni.

The finite-difference time-domain (FDTD) simulation is performed by FDTD Package (Lumerical Solution, Inc.). Metal nanoparticles are set with a diameter of 20 nm. The simulation area is divided into uniform Yee cells with perfectly matched layer boundary condition²². An incident plane wave is propagated from the top along z axis and polarized along the x axis, and a wavelength of 580 nm light is used for simulation. The process is performed at 1 nm mesh resolution and 1000 fs time.

The dielectric constants of Cu and Ni nanoparticles are assumed to be the same as that of the bulk metal and were obtained from Johnson and Christy²³. The dielectric constant (ϵ) of CuNi alloy is rarely available and hence, to describe the plasmonic properties, the dielectric constant of the alloys is calculated by the composition-weighted average method^{24, 25}, where x_{Ni} is the Ni volume fraction, as given below:

$$\varepsilon_{\text{alloy}}(x_{\text{Ni}}, \omega) = x_{\text{Ni}}\varepsilon_{\text{Ni}}(\omega) + (1 - x_{\text{Ni}})\varepsilon_{\text{Cu}}(\omega) \quad (\text{Equ. S2})$$

For Cu nanoparticles, plasmonic "hot spot" regions could be clearly observed around the particles (Fig. S12e), in contrast, Ni do not exhibit distinct "hot spot" regions near the periphery of the particles (Fig. S12g), which is related to the intrinsic electronic structure and optical properties of different metals²⁶. After forming CuNi alloy, ring-shaped "hot spot" regions could be also observed near the periphery of alloy particles, but with an intensity weaker than that of Cu (Fig. S12f). FDTD results validate the strong LSPR-induced localized electric field of Cu and CuNi, confirming the characteristic absorption peaks of Cu and CuNi in the absorption spectrum.

Furthermore, many studies have also pointed out that the introduction of LSPR in metal particles leads to an overall increase in absorbance^{27,28}. This may be attributed to the fact that when the LSPR effect occurs, the free electrons in the metal nanoparticles undergo surface oscillations, causing a change in the electron density distribution near the surface. As a result, the effective optical cross-section of the metal particles increases, and metal nanoparticles could absorb more photons near the resonance frequency, thereby enhancing the overall absorbance^{21,29}. This explains the enhanced overall absorption in the visible and NIR regions observed after the introduction of CuNi and Cu into the MgH₂ system, which increases with higher Cu content (Fig. 2b). When multi-metal nanoparticles are in close proximity, their LSPR effects couple with each other and form resonant states, leading to the scattering of the NIR by light interference with a more intense localized electric field and the increase of the overall absorbance³⁰. This is also evident in the improvement of absorption in the NIR region when increasing the catalyst content to 20% (Fig. S12b). Therefore, we believe that, after ball milling, the LSPR with the coupling effect of Cu and CuNi alloys are the primary factors for the overall enhancement of the absorbance of MgH₂ composites.

Response to Reviewer #3

Reviewer #3: This paper describes reversible hydrogen storage on the Mg₂Ni(Cu)H₄ under solar light irradiation. By addition of Ni and Cu into the MgH₂, the samples showed better solar light absorption property, leading to higher surface temperature and better properties related to hydrogen storage. This is very interesting; it seems that relation between light absorption and increase of temperatures might be very important in this paper. Thus, the authors should discuss the relation and the mechanism of the light absorption more deeply. Especially, following points as listed below should be discussed carefully.

Response: We thank the reviewer for these invaluable comments which have greatly improved the quality of our manuscript, especially the mechanism of the light absorption. We have carefully addressed the reviewer's concerns point by point as follows.

1. Figure 4f shows DOS of the samples. To compare the density (peak intensity) among the orbitals, the authors should add the scale. The unit of a.u. is not suitable. More accurate unit should be used.

Response: We would appreciate the reviewer's careful review of our manuscript. We have added specific numerical values with units (states/eV) to all DOS plots in the manuscript and supporting information. All the updated figures (highlighted in red color) have been provided in the revised manuscript and supporting information as follows:

[Manuscript - Page 18]

“A plane wave energy cutoff of 520 eV and Gamma centered k-point meshes (3x3x1) with same density were applied to surface calculations. A 7x7x7 Gamma centered k-point meshes is used for the dos calculation of Cu, MgH₂, Mg₂NiH₄, Mg₂Ni(Cu)H₄, and 10x10x10 is used for the dos calculation of CuNi. In energy band calculation, the number of k-points is uniformly 20, and the k-path and energy band information are exported by the VASP post-processing tool VASPKIT. The Gaussian smearing method was used with a width of 0.05 eV, and spin polarization was considered in all calculations. The structures were relaxed until the forces and total energy on all atoms were converged to less than 0.05 eV Å⁻¹ and 1 × 10⁻⁵ eV. The MgH₂ dissociation and diffusions of H in various crystal lattice are simulated using the climbing-image nudged elastic band (CI-NEB) method³⁴. The climbing-image nudged elastic band (CI-NEB) method developed by Henkelman *et al.* was employed to locate the transition states of the MgH₂ dissociation and diffusions of H in various crystal lattice^{35,36}. ”

[Manuscript - Page 28]

Fig. 4. a) XRD patterns of MgH₂ under the catalysis of Cu₁Ni₁ at different states. b) contour plots of XRD patterns during re-dehydrogenation process of MgH₂ under the catalysis of Cu₁Ni₁. **Band structures of Ni(Cu)-orbitals and partial densities of states (PDOS) of corresponding atoms for c-d) Mg₂NiH₄ and e-f) Mg₂Ni(Cu)H₄.**

[Supporting Information - Page 27]

Fig. S29. a) Densities of states of Mg and H orbitals for MgH_2 with the calculated contribution percentages at the valence band and the conduction band. Partial densities of states of corresponding atoms for b) MgH_2 and c) MgCu_2 .

2. (L291) The authors described that in terms of pure MgH_2 , the valence band is mainly contributed by H(s) orbitals and the conduction band is composed of Mg(s) and Mg(p) orbitals. However, Figure 4d also suggested that Mg orbitals and H orbitals contribute to the conduction band and valence band, respectively. So, the authors should calculate the contribution percentages of H and Mg orbitals at the valence band and the conduction band. (Qualitative discussion is needed.)

Response: We would appreciate the reviewer's constructive comment. We have replotted the TDOS and PDOS of MgH_2 and calculated the contribution percentages of H(s) and Mg(sp) orbitals in the valence and conduction bands separately based on the PDOS (Fig. S29a).

The results indicate that the valence band is mainly contributed by H(s) (79.3%) with strong hybridization with Mg(sp) (20.7%) and the conduction band is composed of 63.1% Mg(sp) and 36.9% H(s). The band structure and PDOS reveal a significant

hybridization between the valence band H(s) and Mg(sp) (Fig. S30a, Fig. S31a-b), and strong ionic bonding occurs between Mg²⁺ and H⁻, resulting in electrons primarily distributed around H atoms (Fig. S30a). The electron distribution near the Fermi level approaches zero, indicating distinct insulating properties of MgH₂.

We appreciate the reviewer's comment and the above-mentioned discussion (highlighted in red color) have been provided in the revised manuscript and supporting information as follows:

[Manuscript - Page 12]

“To reveal the mechanism behind the enhancement of solar absorption of MgH₂ under the catalysis of CuNi alloy before and after cycling, the electronic density of states (DOS) is investigated. In terms of pure MgH₂, the valence band is mainly contributed by H(s) (79.3%) with strong hybridization with Mg(sp) (20.7%) and the conduction band is composed of 63.1% Mg(sp) and 36.9% H(s) (Fig. S29a). Strong ionic bonding occurs between Mg²⁺ and H⁻, resulting in electrons primarily distributed around H atoms (Fig. S30a), while the electron distribution near the Fermi level approaches zero, indicating distinct insulating properties (Fig. S29b, Fig. S31a-b).”

[Manuscript - Page 28]

Fig. 4. a) XRD patterns of MgH₂ under the catalysis of Cu₁Ni₁ at different states. b) contour plots of XRD patterns during re-dehydrogenation process of MgH₂ under the catalysis of Cu₁Ni₁. **Band structures of Ni(Cu)-orbitals and partial densities of states (PDOS) of corresponding atoms for c-d) Mg₂NiH₄ and e-f) Mg₂Ni(Cu)H₄.**

[Supporting Information - Page 27]

Fig. S29. a) Densities of states of Mg and H orbitals for MgH_2 with the calculated contribution percentages at the valence band and the conduction band. Partial densities of states of corresponding atoms for b) MgH_2 and c) MgCu_2 .

[Supporting Information - Page 28]

Fig. S31. Band structures of Mg(*sp*)-orbitals of MgH₂ a) without other orbitals and b) with H(*s*). Band structures of Mg(*sp*)-orbitals of Mg₂NiH₄ c) without other orbitals, d) with Ni(*d*), and e) H(*s*). Band structures of Mg(*sp*)-orbitals of Mg₂Ni(Cu)H₄ f) without other orbitals, g) with Ni(*d*) and Cu(*d*), and h) H(*s*).

3. (L296) The authors showed that strong absorption peaks in the visible light range could be observed after the addition of CuNi alloy into MgH₂, mainly attributed to the LSPR effect of the metal catalyst. However, from the UV-vis absorption spectra it is hard to conclude that strong absorption peaks in the visible light range was mainly attributed to the LSPR effect. Quantitative discussion is needed. The authors should do peak separation for good understanding of the absorption spectra; the absorption may be sum of d-d transition, LSPR, and other effects.

Response: Thanks for the helpful suggestion. Due to the low concentration (only 10%) of Cu and CuNi alloy in MgH₂, the LSPR peaks in Fig. 2b on that original wide coordinate scale is not sufficiently shown. We further zoomed in the UV-vis absorption spectra of MgH₂ catalyzed by Cu and Cu₁Ni₁ in the range of 400-800 nm, where characteristic peak due to the LSPR effect of Cu could be observed clearly (Fig. S12a). According to the band structure of Cu, this is ascribed to the LSPR effect due to intra-band absorption, which occurs by the excitation of free electrons of conduction

electrons near the Fermi surface from *sp-d* hybridized atomic orbitals of Cu. However, the *d*-orbitals are fully occupied, leading to a relatively weaker occurrence of *d-d* inter-band transitions (Fig. S32).

To further validate the LSPR effect of Cu and CuNi alloy, we conducted finite-difference time-domain (FDTD) simulations to investigate the radiation energy transformation of Cu, CuNi, and Ni. The FDTD simulation is performed by FDTD Package (Lumerical Solution, Inc.). Metal nanoparticles are set with a diameter of 20 nm. The simulation area is divided into uniform Yee cells with perfectly matched layer boundary condition²². An incident plane wave is propagated from the top along z axis and polarized along the x axis, and a wavelength of 580 nm light is used for simulation. The process is performed at 1 nm mesh resolution within the period of 1000 fs. The dielectric constants of Cu and Ni nanoparticles are assumed to be the same as that of the bulk metal and were obtained from Johnson and Christy²³. The dielectric constant (ϵ) of CuNi alloy is rarely available and hence, to describe the plasmonic properties, the dielectric constant of the alloys is calculated by the composition-weighted^{24, 25} by $\epsilon_{\text{alloy}}(x_{\text{Ni}}, \omega) = x_{\text{Ni}}\epsilon_{\text{Ni}}(\omega) + (1 - x_{\text{Ni}})\epsilon_{\text{Cu}}(\omega)$, where x_{Ni} is the Ni volume fraction, as shown in Fig. S12c-d. For Cu nanoparticles, plasmonic "hot spot" regions could be clearly observed around the particles (Fig. S12e), in contrast, Ni do not exhibit distinct "hot spot" regions near the periphery of the particles (Fig. S12g), which is related to the intrinsic electronic structure and optical properties of different metals²⁶. After the introduction of CuNi alloy, ring-shaped "hot spot" regions could be also observed near the periphery of alloy particles, but with an intensity weaker than that of Cu (Fig. S12f). FDTD results validate the strong LSPR-induced localized electric field of Cu and CuNi, confirming the characteristic absorption peaks of Cu and CuNi in the absorption spectrum.

Furthermore, many studies have also pointed out that the introduction of LSPR in metal particles leads to an overall increase in absorbance^{27, 28}. This may be attributed to the fact that when the LSPR effect occurs, the free electrons in the metal nanoparticles undergo surface oscillations, causing a change in the electron density distribution near the surface. As a result, the effective optical cross-section of the metal particles increases, and metal nanoparticles could absorb more photons near the resonance frequency, thereby enhancing the overall absorbance^{21, 29}. This explains the enhanced overall absorption in the visible and NIR regions observed after the introduction of CuNi and Cu into the MgH₂ system, which increases with higher Cu content (Fig. 2b). When multi-metal nanoparticles are in close proximity, their LSPR effects couple with each other and form resonant states, leading to the scattering of the NIR by light interference with a more intense localized electric field and the increase of the overall absorbance³⁰. This is also evident in the improvement of absorption in the NIR region when increasing the catalyst content to 20% (Fig. S12b).

Therefore, we believe that, after ball milling, the LSPR with the coupling effect of Cu and CuNi alloys are the primary factors for the overall enhancement of the absorbance of MgH₂ composites. Considering that there are often discrepancies between the theoretical prediction based on band structures and experimental observation, we refrained from further separating the absorption spectrum³⁷.

We appreciate the reviewer's comment and all the above-mentioned results and the related discussion (highlighted in red color) have been provided in the revised manuscript and supporting information as follows:

[Manuscript - Page 13]

“After adding CuNi alloy into MgH₂, strong absorption peaks in the visible light range could be ascribed to the LSPR effect due to intra-band absorption (Fig. S12a), which occurs by the excitation of free electrons of conduction electrons near the Fermi surface from *sp-d* hybridized atomic orbitals of Cu (Fig. S32), implying that the initial dehydrogenation process is driven by the LSPR effect of CuNi.”

[Supporting Information - Page 12]

Fig. S12. UV–vis–NIR absorption spectra of MgH₂ under the catalysis of a) Cu₁Ni₁ and Cu in the range of 400–800 nm and b) Cu₁Ni₁ with addition of 10% and 20%. c) Real and d) imaginary parts of dielectric functions of Cu, Cu₁Ni₁, and Ni metals. The dotted line represents the case where $\epsilon_r \approx -2\epsilon_m$, air. FDTD simulated localized electric field enhancement profiles of e) Cu , f) Cu₁Ni₁, and g) Ni.

The finite-difference time-domain (FDTD) simulation is performed by FDTD Package (Lumerical Solution, Inc.). Metal nanoparticles are set with a diameter of 20 nm. The simulation area is divided into uniform Yee cells with perfectly matched layer boundary condition²². An incident plane wave is propagated from the top along z axis and polarized along the x axis, and a wavelength of 580 nm light is used for simulation. The process is performed at 1 nm mesh resolution within the period of 1000 fs.

The dielectric constants of Cu and Ni nanoparticles are assumed to be the same as that of the bulk metal and were obtained from Johnson and Christy²³. The dielectric constant (ϵ) of

CuNi alloy is rarely available and hence, to describe the plasmonic properties, the dielectric constant of the alloys is calculated by the composition-weighted average method^{24, 25}, where x_{Ni} is the Ni volume fraction, as given below:

$$\varepsilon_{alloy}(x_{Ni}, \omega) = x_{Ni}\varepsilon_{Ni}(\omega) + (1 - x_{Ni})\varepsilon_{Cu}(\omega) \quad (\text{Equ. S2})$$

For Cu nanoparticles, plasmonic "hot spot" regions could be clearly observed around the particles (Fig. S12e), in contrast, Ni do not exhibit distinct "hot spot" regions near the periphery of the particles (Fig. S12g), which is related to the intrinsic electronic structure and optical properties of different metals²⁶. After the introduction of CuNi alloy, ring-shaped "hot spot" regions could be also observed near the periphery of alloy particles, but with an intensity weaker than that of Cu (Fig. S12f). FDTD results validate the strong LSPR-induced localized electric field of Cu and CuNi, confirming the characteristic absorption peaks of Cu and CuNi in the absorption spectrum.

Furthermore, many studies have also pointed out that the introduction of LSPR in metal particles leads to an overall increase in absorbance^{27, 28}. This may be attributed to the fact that when the LSPR effect occurs, the free electrons in the metal nanoparticles undergo surface oscillations, causing a change in the electron density distribution near the surface. As a result, the effective optical cross-section of the metal particles increases, and metal nanoparticles could absorb more photons near the resonance frequency, thereby enhancing the overall absorbance^{21, 29}. This explains the enhanced overall absorption in the visible and NIR regions observed after the introduction of CuNi and Cu into the MgH₂ system, which increases with higher Cu content (Fig. 2b). When multi-metal nanoparticles are in close proximity, their LSPR effects couple with each other and form resonant states, leading to the scattering of the NIR by light interference with a more intense localized electric field and the increase of the overall absorbance³⁰. This is also evident in the improvement of absorption in the NIR region when increasing the catalyst content to 20% (Fig. S12b). Therefore, we believe that after ball milling, the LSPR with the coupling effect of Cu and CuNi alloys are the primary factors for the overall enhancement of the absorbance of MgH₂ composites.

[Supporting Information - Page 28]

Fig. S32. Band structure of Cu nanoparticles with orbitals of *s*, *p*, and *d*.

4. (L304) The authors insisted that the electronic distribution of Mg(*s*) and Mg(*p*) in both Mg₂NiH₄, MgCu₂, and Mg₂Ni(Cu)H₄ tends to fill the originally vacant Fermi level, resulting in the uniform distribution of *s*-*p* band. The authors should discuss why Mg *s* and *p* orbitals in Mg₂Ni(Cu)H₄ fill the vacant Fermi level although the Mg orbitals in MgH₂ do not fill.

Response: We appreciate the reviewer's helpful advice. To further investigate the factors influencing the distribution of Mg(*sp*) orbitals near the Fermi level after the introduction of Ni and Cu, the calculated charge densities distributions and band structure for MgH₂, Mg₂NiH₄, and Mg₂Ni(Cu)H₄ have been included.

For MgH₂, Mg²⁺ and H⁻ form ionic bonds with robust interaction, and a significant hybridization could be observed between *p* and *s* orbitals of Mg and H in the valence band, resulting in the high stability of MgH₂ (Fig. S31a-b). Owing to the ionic bonding of MgH₂, electrons are primarily concentrated around H atoms (Fig. S30a).

Compared to the nature of ionic bonding of MgH₂, the bonding nature of Mg₂NiH₄ and Mg₂Ni(Cu)H₄ is mainly ionic with covalent bonds between Ni(Cu) metal atoms and their neighbor H atoms. Due to relatively stronger covalent bonds, clear electrons sharing between Ni(Cu) elements and H atoms could be observed. Simultaneously, part of electrons is also distributed around Mg atoms, indicating partial involvement of electrons from Mg(*sp*) orbitals in the formation of covalent bonds, which means the introduction of Ni and Cu weakens the Mg-H bond and leads to the uniform distribution of *s*-*p* band (Fig. S30b-c).

As for band structure, there is a relatively weak distribution of Ni(*d*), Mg(*sp*), and H(*s*) orbitals near the Fermi level in Mg₂NiH₄, demonstrating semiconductor properties (Fig. 4c-d, Fig. S31c-e). However, the solubility of Cu leads to the apparition of a more

flattened hybridized orbital, resembling a "flat band", which results in the formation of a continuous band structure filling the vacant Fermi level and metallic properties of $\text{Mg}_2\text{Ni}(\text{Cu})\text{H}_4$. The bonding peak near the Fermi energy is mainly contributed $\text{Ni}(d)$, accompanied by weaker contributions from $\text{Mg}(sp)$ and $\text{H}(s)$. This can be attributed to the robust hybridization of $\text{Cu}(d)$ and $\text{Ni}(d)$ orbitals within the valence band, resulting in a broader d -band width. The extension of the d -band facilitates hybridization of them with $\text{Mg}(sp)$ and $\text{H}(s)$ orbitals, further altering the electron distribution, and thereby providing facile pathway for inter-/intra-band transitions of electrons (including s - p and d orbitals) and leading to a broad absorption spectrum within the visible to near-infrared range (Fig. 4e-f, Fig. S31f-h) ^{38, 39, 40}.

In summary, the electronic behavior near the Fermi level between MgH_2 and $\text{Mg}_2\text{Ni}(\text{Cu})\text{H}_4$ arises from the differences in chemical bonding properties and orbital hybridization. We appreciate the reviewer's comment and all the above-mentioned results and the related discussion (highlighted in red color) have been provided in the revised manuscript and supporting information as follows:

[Manuscript - Page 13]

"As mentioned above, CuNi alloys would react with Mg during the dehydrogenation process, leading to the formation of $\text{Mg}_2\text{Ni}(\text{Cu})$ alloy and then the formation of Mg_2NiH_4 , MgCu_2 and $\text{Mg}_2\text{Ni}(\text{Cu})\text{H}_4$ upon reversible hydrogenation. In comparison to the ionic bonding of MgH_2 , covalent bonds between $\text{Ni}(\text{Cu})$ metal atoms and their neighbor H atoms exist in Mg_2NiH_4 and $\text{Mg}_2\text{Ni}(\text{Cu})\text{H}_4$ with clear electrons sharing (Fig.S30b-c), leading to the weakening of Mg-H bonds and uniform distribution of s - p band. Near the Fermi level, a relatively weak distribution of electrons could be observed in Mg_2NiH_4 with semiconductor properties (Fig. 4c-d, Fig. S31c-e), while, due to the solubility of Cu , a more flattened hybridized orbital, resembling a "flat band", appears, which is mainly $\text{Ni}(d)$ orbitals hybridized with weaker $\text{Mg}(sp)$ and $\text{H}(s)$ (Fig. 4e-f, Fig. S31f-h) ^{38, 39, 40}. As a result, a continuous band structure fills the vacant Fermi level in $\text{Mg}_2\text{Ni}(\text{Cu})\text{H}_4$ with metallic properties, together with MgCu_2 (Fig. S29c), which induces the facile excitation of electrons under light irradiation. Attributed to the robust hybridization of $\text{Cu}(d)$ and $\text{Ni}(d)$ orbitals within the valence band, the d -band is extended, which facilitates hybridization with $\text{Mg}(sp)$ and $\text{H}(s)$ orbitals, thereby providing facile pathway for inter-/intra-band transitions of electrons (including s - p and d orbitals) and leading to a broad absorption spectrum within the visible to near-infrared range ^{16, 40}."

[Manuscript - Page 28]

Fig. 4. a) XRD patterns of MgH₂ under the catalysis of Cu₁Ni₁ at different states. b) contour plots of XRD patterns during re-dehydrogenation process of MgH₂ under the catalysis of Cu₁Ni₁. **Band structures of Ni(Cu)-orbitals and partial densities of states (PDOS) of corresponding atoms for c-d) Mg₂NiH₄ and e-f) Mg₂Ni(Cu)H₄.**

[Supporting Information - Page 27]

Fig. S30. Isosurfaces of the calculated charge densities with an isovalue of $0.05 \text{ e } \text{Å}^{-3}$ for a) MgH_2 , b) Mg_2NiH_4 and c) $\text{Mg}_2\text{Ni}(\text{Cu})\text{H}_4$.

[Supporting Information - Page 28]

Fig. S31. Band structures of $\text{Mg}(sp)$ -orbitals of MgH_2 a) without other orbitals and b) with $\text{H}(s)$. Band structures of $\text{Mg}(sp)$ -orbitals of Mg_2NiH_4 c) without other orbitals, d) with $\text{Ni}(d)$, and e) $\text{H}(s)$. Band structures of $\text{Mg}(sp)$ -orbitals of $\text{Mg}_2\text{Ni}(\text{Cu})\text{H}_4$ f) without other orbitals, g) with $\text{Ni}(d)$ and $\text{Cu}(d)$, and h) $\text{H}(s)$.

5. (L306) It was described that this provides facile pathway for intra-band transitions of electrons (including s-p and d orbitals), leading to a broad absorption spectrum within the visible to near-infrared range and the localization of d electrons is capable of inducing stronger inter-band absorption than electrons in the s and p orbitals. The authors should more carefully discuss which orbitals contribute to the intra band and inter band transitions. Furthermore, the authors should explain why the localization of d electrons induces stronger inter-band absorption.

Response: Thanks for the constructive suggestion. The band structures of Mg_2NiH_4 and $\text{Mg}_2\text{Ni}(\text{Cu})\text{H}_4$ visually illustrate the orbitals distributions near the Fermi level , which contributes to the inter/intra-band transitions. Near the Fermi level, a relatively weak distribution of electrons could be observed in Mg_2NiH_4 with semiconductor properties (Fig. 4c-d, Fig. S31c-e), while, due to the solubility of Cu, a more flattened hybridized orbital, resembling a "flat band", appears, which is mainly contributed by Ni(*d*) orbitals hybridized with weaker Mg(*sp*) and H(*s*) (Fig. 4e-f, Fig. S31f-h). Additionally, the robust hybridization of Cu(*d*) and Ni(*d*) orbitals within the valence band, resulting in a broader *d*-band width. The extension of the *d*-band facilitates hybridization of them with Mg(*sp*) and H(*s*) orbitals, further altering the electron distribution, and thereby providing facile pathway for inter-/intra-band transitions of electrons (including *s-p* and *d* orbitals) and leading to a broad absorption spectrum within the visible to near-infrared range.

Furthermore, due to the unique spatial distribution and electronic properties, the localization of d electrons leads to stronger inter-band absorption^{26, 41}. Compared to *s* and *p* orbitals, *d* orbitals are more complex in shape and have multiple lobes, which can be highly localized near the nucleus and experience stronger electron-electron interactions. Stronger interactions make it more likely the transition of a *d* electron to a higher energy state when it absorbs a photon. In addition, the localized nature of *d* orbitals increases the overlap, making inter-band transitions more probable.

We appreciate the reviewer's comment and all the above-mentioned discussion (highlighted in red color) have been provided in the revised manuscript as follows:

[Manuscript - Page 13]

"Near the Fermi level, a relatively weak distribution of electrons could be observed in Mg_2NiH_4 with semiconductor properties (Fig. 4c-d, Fig. S31c-e), while, due to the solubility of Cu, a more flattened hybridized orbital, resembling a "flat band", appears, which is mainly Ni(*d*) orbitals hybridized with weaker Mg(*sp*) and H(*s*) (Fig. 4e-f, Fig. S31f-h). As a result, a continuous band structure fills the vacant Fermi level in $\text{Mg}_2\text{Ni}(\text{Cu})\text{H}_4$ with metallic properties, together with MgCu_2 (Fig. S29c), which induces the facile excitation of electrons under light irradiation. Attributed to the robust hybridization of Cu(*d*) and Ni(*d*) orbitals within the valence

band, the *d*-band is extended, which facilitates hybridization of them with Mg(*sp*) and H(*s*) orbitals, thereby providing facile pathway for inter-/intra-band transitions of electrons (including *s-p* and *d* orbitals) and leading to a broad absorption spectrum within the visible to near-infrared range ^{38, 39, 40}. In addition, the *d* electrons may undergo transitions to a higher energy state when absorbing photons with more probability due to the overlap of the localization of *d* orbitals, thereby inducing stronger inter-band absorption ^{41, 42}. More importantly, the localization of *d*-electrons introduced by the presence of Ni and Cu could be well preserved even after the formation of Mg₂Ni(Cu) alloy after dehydrogenation.”

[Manuscript - Page 28]

Fig. 4. a) XRD patterns of MgH₂ under the catalysis of Cu₁Ni₁ at different states. b) contour plots of XRD patterns during re-dehydrogenation process of MgH₂ under the catalysis of Cu₁Ni₁. **Band structures of Ni(Cu)-orbitals and partial densities of states (PDOS) of corresponding atoms for c-d) Mg₂NiH₄ and e-f) Mg₂Ni(Cu)H₄.**

[Supporting Information - Page 28]

Fig. S31. Band structures of Mg(*sp*)-orbitals of MgH₂ a) without other orbitals and b) with H(*s*). Band structures of Mg(*sp*)-orbitals of Mg₂NiH₄ c) without other orbitals, d) with Ni(*d*), and e) H(*s*). Band structures of Mg(*sp*)-orbitals of Mg₂Ni(Cu)H₄ f) without other orbitals, g) with Ni(*d*) and Cu(*d*), and h) H(*s*).

References

1. Ji L, Zhang L, Yang X, Zhu X, Chen L. The remarkably improved hydrogen storage performance of MgH₂ by the synergetic effect of an FeNi/rGO nanocomposite. *Dalton Transactions* **49**, 4146-4154 (2020).
2. Hou Q, Zhang J, Yang X, Ding Z. Ni₃Fe/BC nanocatalysts based on biomass charcoal self-reduction achieves excellent hydrogen storage performance of MgH₂. *Dalton Transactions* **51**, 14960-14969 (2022).
3. Ding Z, *et al.* Improve hydrogen sorption kinetics of MgH₂ by doping carbon-encapsulated iron-nickel nanoparticles. *Journal of Alloys and Compounds* **843**, 156035 (2020).
4. Guemou S, *et al.* Exceptional catalytic effect of novel rGO-supported Ni-Nb nanocomposite on the hydrogen storage properties of MgH₂. *Journal of Materials Science & Technology*, (2023).
5. Huang T, *et al.* Enhancing hydrogen storage properties of MgH₂ through addition of Ni/CoMoO₄ nanorods. *Materials Today Energy* **19**, 100613 (2021).
6. Liang H, *et al.* Facile synthesis of nickel-vanadium bimetallic oxide and its catalytic effects on the hydrogen storage properties of magnesium hydride. *International Journal of Hydrogen Energy* **47**, 32969-32980 (2022).
7. Zang J, *et al.* Ni, beyond thermodynamic tuning, maintains the catalytic activity of V species in Ni₃(VO₄)₂ doped MgH₂. *Journal of Materials Chemistry A* **9**, 8341-8349 (2021).
8. Zhao Y, *et al.* Enhancing hydrogen storage properties of MgH₂ by core-shell CoNi@C. *Journal of Alloys and Compounds* **862**, 158004 (2021).
9. Huang X, *et al.* Synergistic catalytic activity of porous rod-like TMTiO₃ (TM= Ni and Co) for reversible hydrogen storage of magnesium hydride. *The Journal of Physical Chemistry C* **122**, 27973-27982 (2018).
10. Xu N, Wang K, Zhu Y, Zhang Y. PdNi biatomic clusters from metallene unlock record - low onset dehydrogenation temperature for bulk - MgH₂. *Advanced Materials*, 2303173.
11. Chen M, *et al.* Synergy between metallic components of MoNi alloy for catalyzing highly efficient hydrogen storage of MgH₂. *Nano Research* **13**, 2063-2071 (2020).
12. Wang Z, *et al.* In situ formed ultrafine NbTi nanocrystals from a NbTiC solid-solution MXene for hydrogen storage in MgH₂. *Journal of Materials Chemistry A* **7**, 14244-14252 (2019).
13. Zhang L, *et al.* Highly active multivalent multielement catalysts derived from hierarchical porous TiNb₂O₇ nanospheres for the reversible hydrogen storage of MgH₂. *Nano Research* **14**, 148-156 (2021).
14. Xin Y, *et al.* Copper - Based Plasmonic Catalysis: Recent Advances and Future Perspectives. *Advanced Materials* **33**, 2008145 (2021).
15. Wang F, *et al.* Photothermal-enhanced catalysis in core-shell plasmonic hierarchical Cu₇S₄ microsphere@zeolitic imidazole framework-8. *Chemical science* **7**, 6887-6893 (2016).
16. Chen G, *et al.* Alumina - supported CoFe alloy catalysts derived from layered - double - hydroxide nanosheets for efficient photothermal CO₂ hydrogenation to

- hydrocarbons. *Advanced Materials* **30**, 1704663 (2018).
17. Lv X, *et al.* Electron - deficient Cu sites on Cu₃Ag₁ catalyst promoting CO₂ electroreduction to alcohols. *Advanced Energy Materials* **10**, 2001987 (2020).
 18. Qiu HJ, *et al.* Nanoporous graphene with single - atom nickel dopants: an efficient and stable catalyst for electrochemical hydrogen production. *Angewandte Chemie International Edition* **54**, 14031-14035 (2015).
 19. Yu W, *et al.* Laser-controlled tandem catalytic sites of CuNi alloy with ampere-level electrocatalytic nitrate to ammonia activity for Zn–nitrate battery. *Energy & Environmental Science*, (2023).
 20. Han S, *et al.* Ultralow overpotential nitrate reduction to ammonia via a three-step relay mechanism. *Nature Catalysis*, 1-13 (2023).
 21. Mayer KM, Hafner JH. Localized surface plasmon resonance sensors. *Chemical reviews* **111**, 3828-3857 (2011).
 22. He Z, Zhao J, Lu H. Tunable nonreciprocal reflection and its stability in a non-PT-symmetric plasmonic resonators coupled waveguide systems. *Applied Physics Express* **13**, 012009 (2019).
 23. Johnson PB, Christy R-W. Optical constants of the noble metals. *Physical review B* **6**, 4370 (1972).
 24. Lee K-S, El-Sayed MA. Gold and silver nanoparticles in sensing and imaging: sensitivity of plasmon response to size, shape, and metal composition. *The Journal of Physical Chemistry B* **110**, 19220-19225 (2006).
 25. Sekhon JS, Malik HK, Verma S. DDA simulations of noble metal and alloy nanocubes for tunable optical properties in biological imaging and sensing. *RSC advances* **3**, 15427-15434 (2013).
 26. Ma X, Sun H, Wang Y, Wu X, Zhang J. Electronic and optical properties of strained noble metals: Implications for applications based on LSPR. *Nano Energy* **53**, 932-939 (2018).
 27. Fang R, *et al.* Transition metal tuned g-C₃N₄ induce highly efficient photocatalytic of ammonia borane to hydrogen evolution and mechanism investigation. *Fuel* **334**, 126707 (2023).
 28. Ji G, Wu S, Tian J. High fuel production rate and excellent durability for photothermocatalytic CO₂ reduction achieved via the surface plasma effect of NiCu alloy nanoparticles. *Catalysis Science & Technology* **13**, 2500-2507 (2023).
 29. Ding X, *et al.* Surface plasmon resonance enhanced light absorption and photothermal therapy in the second near-infrared window. *Journal of the American Chemical Society* **136**, 15684-15693 (2014).
 30. Petryayeva E, Krull UJ. Localized surface plasmon resonance: Nanostructures, bioassays and biosensing-A review. *Analytica chimica acta* **706**, 8-24 (2011).
 31. Marimuthu A, Zhang J, Linic S. Tuning selectivity in propylene epoxidation by plasmon mediated photo-switching of Cu oxidation state. *Science* **339**, 1590-1593 (2013).
 32. Singh M, Sinha I, Singh A, Mandal R. LSPR and SAXS studies of starch stabilized Ag–Cu alloy nanoparticles. *Colloids and Surfaces A: Physicochemical and Engineering Aspects* **384**, 668-674 (2011).
 33. Bobinger M, Angeli D, Colasanti S, La Torraca P, Larcher L, Lugli P. Infrared, transient

- thermal, and electrical properties of silver nanowire thin films for transparent heaters and energy - efficient coatings. *Physica Status Solidi (a)* **214**, 1600466 (2017).
34. Heyden A, Bell AT, Keil FJ. Efficient methods for finding transition states in chemical reactions: comparison of improved dimer method and partitioned rational function optimization method. *The Journal of Chemical Physics* **123**, 224101 (2005).
 35. Henkelman G, Jónsson H. Improved tangent estimate in the nudged elastic band method for finding minimum energy paths and saddle points. *The Journal of chemical physics* **113**, 9978-9985 (2000).
 36. Henkelman G, Uberuaga BP, Jónsson H. A climbing image nudged elastic band method for finding saddle points and minimum energy paths. *The Journal of chemical physics* **113**, 9901-9904 (2000).
 37. Shangguan W, *et al.* Molecular-level insight into photocatalytic CO₂ reduction with H₂O over Au nanoparticles by interband transitions. *Nature Communications* **13**, 3894 (2022).
 38. Li X, *et al.* Accessing parity-forbidden d-d transitions for photocatalytic CO₂ reduction driven by infrared light. *Nature Communications* **14**, 4034 (2023).
 39. Yang B, *et al.* Flatband λ -Ti₃O₅ towards extraordinary solar steam generation. *Nature*, 1-8 (2023).
 40. Aslam U, Rao VG, Chavez S, Linic S. Catalytic conversion of solar to chemical energy on plasmonic metal nanostructures. *Nature Catalysis* **1**, 656-665 (2018).
 41. Jian C-c, Zhang J, Ma X. Cu–Ag alloy for engineering properties and applications based on the LSPR of metal nanoparticles. *RSC advances* **10**, 13277-13285 (2020).
 42. Li Y, *et al.* High-entropy-alloy nanoparticles with enhanced interband transitions for efficient photothermal conversion. *Angewandte Chemie International Edition* **60**, 27113-27118 (2021).

REVIEWER COMMENTS

Reviewer #1 (Remarks to the Author):

The authors have well addressed all my questions. I support the publication of this revised manuscript in the current version.

Reviewer #2 (Remarks to the Author):

The author provided a detailed response to the review comments, and some of the answers were correct. In terms of innovation in the article, the author still needs further refinement. The authors should further strengthen the introduction section of the manuscript. The manuscript might be accepted for publication with major reversion after addressing the following issues.

1. I think the author still needs to further strengthen the innovation aspect of the article. The advantage of photothermal catalysis is not only that it does not require external energy input for heating, but also that it can solve problems that cannot be solved by thermal catalysis.
2. About the role of light in this work. The author explains that the better photothermal performance than pure thermal performance is a localized thermal effect. We believe that such an explanation is insufficient. Local thermal effects are essentially thermal catalysis, and their functions are the same except for different heating methods. In addition, the activation energy (Figure S14f) under photothermal and pure thermal conditions is the same, which further explains the light only provides a thermal effect during the reaction process. However, why the performance is better than under pure thermal conditions. A detailed explanation was needed.
3. About temperature testing under light irradiation. (Figure S11) At present, the use of infrared thermometers in the field of photothermal catalysis is not accepted. The author should use a thermocouple to test the temperature of the catalyst surface in contact with the catalyst. In the article, the reason why the performance of photothermal catalysis is better than pure thermal performance is due to inaccurate temperature testing of the catalyst surface under photothermal conditions. As far as we know, using an infrared thermometer to test the temperature is 20-50°C lower than the actual temperature. Therefore, the surface temperature of the catalyst needs to be remeasured, and the comparison experiment between photothermal and pure thermal needs to be redone. (Figure 2c, 2g, S14b S14e,f , S16a,c and d)
4. In Figure 1, the evidence is insufficient and needs to be further proved by line scanning two adjacent particles.

Reviewer #3 (Remarks to the Author):

This paper describes reversible hydrogen storage on the $\text{Mg}_2\text{Ni}(\text{Cu})\text{H}_4$ under solar light irradiation. Based on the reviewer's comments, the authors added many data and discussed the data soundly. All of the listed problems were solved. Thus, it seems that this paper can be recommended for publication in Nature Communications without further revision.

Response to Reviewers

Dear Reviewers,

We sincerely thank you for taking the time to review the manuscript and giving positive comments. We are very grateful for your comments on the influence of light and corresponding mechanism of CuNi alloy in this work. Your professional and academic comments really help us to further improve the quality of the work. We respond to the comments by adding more comprehensive experiments and theoretical calculations.

Accordingly, based on these constructive comments and suggestions, we have made corresponding revisions in the manuscript and highlighted the revised parts **in red** for easy tracking. With these improvements, we believe that the revised manuscript can better meet the high standards of *Nature Communications*. The point-by-point responses highlighted **in blue** are as follows:

Response to Reviewer #1

Reviewer #1: The authors have well addressed all my questions. I support the publication of this revised manuscript in the current version.

Response: Thank you very much for your approval and guidance on our manuscript.

Response to Reviewer #2

Reviewer #2: The author provided a detailed response to the review comments, and some of the answers were correct. In terms of innovation in the article, the author still needs further refinement. The authors should further strengthen the introduction section of the manuscript. The manuscript might be accepted for publication with major reversion after addressing the following issues.

Response: We sincerely thank the reviewer for these valuable comments for strengthening our manuscript. We have carefully addressed the reviewer's concerns point by point as follows.

1. I think the author still needs to further strengthen the innovation aspect of the article. The advantage of photothermal catalysis is not only that it does not require external energy input for heating, but also that it can solve problems that cannot be solved by thermal catalysis.

Response: We thank the reviewer for this constructive comment. In order to further

explore the role of light irradiation in improving hydrogen storage performance of MgH₂ besides local heat, we conducted *in-situ* irradiated XPS (ISI-XPS) on CuNi and Mg₂Ni(Cu) alloy. To our surprise, the binding energy peaks of Cu 2*p* and Ni 2*p* exhibited positive and negative shifts, respectively, compared to the dark conditions, and the uneven electron distribution is consistently maintained upon light irradiation (Fig. S37a-b). This implies that, attributed to the smaller work function of Cu (4.65 eV) than that of Ni (5.15 eV), the hot electrons generated by the LSPR effect of Cu are transferred to Ni, leading to an increase in electron density around Ni during light irradiation. This phenomenon has been observed in other previously reported results^{1, 2, 3}. After the *in-situ* formation of Mg₂Ni(Cu) alloy during the initial dehydrogenation, the relative transformation of hot electrons between Cu and Ni is well preserved upon light irradiation. However, no observable changes in the binding energy of Mg atoms are observed, attributed to the lower work function of Mg (3.68 eV) than that of Cu (Fig. S37c-e). This indicates an additional effect of photogenerated charge migration in the designed CuNi alloy and *in-situ* formed Mg₂Ni(Cu) under light irradiation besides local heat.

Furthermore, theoretical calculations were performed to further investigate the effect of imbalanced electron distribution in CuNi and Mg₂Ni(Cu) alloy in improving their catalytic performance. The differential charge density based on all electrons reveals an electron accumulation at the Ni sites in both CuNi and Mg₂Ni(Cu) alloy (Fig. S38 a-b), which enables Ni sites to contribute more electrons H bonded to Mg atoms of MgH₂ (Fig. S39 a-d), resulting in the weakening of Mg-H bonds (Fig. S39 e-h). As a result, compared to pure Ni with uniform electron distribution, the electron enrichment at the Ni sites in CuNi and Mg₂Ni(Cu) alloys facilitates the elongation of Mg-H bonds from 1.87 Å to 1.91 Å and 2.00 Å, respectively, which directly demonstrates the weakening of Mg-H bonds of MgH₂ and hence the superior catalytic effect induced by the electron enrichment at the Ni sites. However, the role of Cu sites in the dissociation of MgH₂ bonds remains unchanged under various electron distribution.

As stated in the original manuscript, the “hydrogen pump” effect of Mg₂Ni(Cu)/Mg₂Ni(Cu)H₄ improves cycling hydrogenation and dehydrogenation of MgH₂ and hence results in the significant decrease of the apparent activation energy required for driving hydrogen storage of MgH₂. Based on this mechanism, we provide additional evidences that, upon light irradiation, the presence of hot electrons induces internal disparities in electron distribution within CuNi and Mg₂Ni(Cu) alloy, influencing the distribution of active catalytic sites, which is conducive to promoting the dissociation of Mg-H bonds of MgH₂ and enhancing the “hydrogen pump” effect of Mg₂Ni(Cu)/Mg₂Ni(Cu)H₄. Additionally, Mg does not affect the electron transfer of CuNi under light irradiation due to its positive charge, ensuring the promotion of Mg-H bond

dissociation by the *in-situ* stable $Mg_2Ni(Cu)$ during cycling.

These hot electrons could be generated continuously under light irradiation and, however, cannot be directly participate in the MgH_2 dehydrogenation process, which subsequently would decay into localized thermal energy ⁴, acting on catalytic sites in the form of heat. This process would elevate the local temperature of catalytic sites and enhance the catalytic effect of both CuNi and $Mg_2Ni(Cu)$ alloys. The *in-situ* atomic reconstruction of $Mg_2Ni(Cu)$ enables the integration of photothermal and catalytic roles in a single-component phase, which allows the direct action of localized photothermal heat on the catalytic sites without any heat loss.

Therefore, compared to traditional thermal catalysis, photothermal catalysis not only achieves the ideal coupling of photothermal and catalytic sites, but also induces internal uneven electron distribution within CuNi and $Mg_2Ni(Cu)$ alloy, leading to a light-enhanced "hydrogen pump" effect. This enables a faster dehydrogenation during cycling under solar irradiation than heating and does not require external energy input for heating.

We appreciate the reviewer's comment and all the above-mentioned results and the related discussion (highlighted in red color) have been provided in the revised manuscript and supporting information as follows:

[Manuscript - Page 1]

Abstract: "...Moreover, in comparison with MgH_2 , the hydrogen storage reaction of $Mg_2Ni(Cu)$ is thermodynamically and kinetically favored resulting from the weakening of Mg-H bonds and the low migration barrier of hydrogen atoms in both $Mg_2Ni(Cu)$ and $Mg_2Ni(Cu)H_4$. Interestingly, the imbalanced distribution of the light-induced hot electrons within CuNi and $Mg_2Ni(Cu)$ alloy facilitates the effective weakening of Mg-H bonds of MgH_2 , which contributes to promoting the dissociation of Mg-H bonds and hence enhancing the "hydrogen pump" effect of $Mg_2Ni(Cu)/Mg_2Ni(Cu)H_4$. This phenomenon would improve cycling hydrogenation and dehydrogenation of MgH_2 and hence result in the significant decrease of the apparent activation energy required for driving hydrogen storage of MgH_2 ..."

[Manuscript - Page 4]

"On the other hand, in comparison with MgH_2 , the hydrogen storage reaction of $Mg_2Ni(Cu)$ is thermodynamically and kinetically favored resulting from the weakening of Mg-H bonds and the low migration barrier of hydrogen atoms in both $Mg_2Ni(Cu)$ and $Mg_2Ni(Cu)H_4$, which in turn provides a facile pathway for the spontaneous breaking of Mg-H bonds of MgH_2 . More importantly, attributed to the smaller work function of Cu than that of Ni and the larger work function of Cu than that of Mg, the hot electrons generated by the LSPR effect of Cu could be transferred to Ni inside of both CuNi and $Mg_2Ni(Cu)$ alloys, leading to an increase in electron density around Ni during light irradiation. The uneven distribution of light-induced hot electrons

within CuNi and Mg₂Ni(Cu) alloy contributes to the effective weakening of Mg-H bonds of MgH₂ and hence enhances the “hydrogen pump” effect of Mg₂Ni(Cu)/Mg₂Ni(Cu)H₄, which improves cycling hydrogenation and dehydrogenation of MgH₂ and significantly reduces the apparent energy required for driving hydrogen storage in MgH₂. As a result, the *in-situ* atomic reconstruction of Mg₂Ni(Cu) enables the integration of photothermal and catalytic roles in a single-component phase, which allows the direct action of localized photothermal heat on the catalytic sites without any heat loss, resulting in the complete dehydrogenation of MgH₂ with a reversible capacity of 6.1 wt.% within 15 minutes under 3.5 W/cm².”

[Manuscript - Page 5]

“Particularly, the binding energies of Cu⁰ and Ni⁰ in CuNi alloys exhibit shifts compared to pure Cu and Ni due to the electronegativity difference, indicating the electronic interactions inside of CuNi alloys ^{5, 6}.”

[Manuscript - Page 15]

“Furthermore, upon light irradiation, the presence of hot electrons induces internal disparities in electron distribution within CuNi and Mg₂Ni(Cu) alloy as observed *via in-situ* irradiated XPS (ISI-XPS) (Fig.S37a-e). Attributed to the work function difference, the hot electrons generated by the LSPR effect of Cu transferred to Ni, leading to an increase in electron density around Ni (Fig. S38 a-b), which enables Ni sites to contribute more electrons to H bonded to Mg atoms of MgH₂ (Fig. S39 a-d), and hence leads to the weakening Mg-H bonds (Fig. S39 e-h). As a result, compared to pure Ni with uniform electron distribution, the electron enrichment at the Ni sites in CuNi and Mg₂Ni(Cu) alloys facilitates the elongation of Mg-H bonds from 1.87 Å to 1.91 Å and 2.00 Å, respectively, while the impact of Cu sites remains unchanged. The internal uneven electron distribution within CuNi and Mg₂Ni(Cu) alloys is conducive to promoting the dissociation of Mg-H bonds and hence improving the “hydrogen pump” effect of Mg₂Ni(Cu)/Mg₂Ni(Cu)H₄. Additionally, Mg does not affect the electron transfer between CuNi under light irradiation due to its positive charge, ensuring the promotion of Mg-H bonds dissociation by the *in-situ* well preserved Mg₂Ni(Cu) during cycling. These hot electrons could be generated continuously under light irradiation and, however, cannot be directly participate in the MgH₂ dehydrogenation process, which subsequently would decay into localized thermal energy ⁴, acting on catalytic sites in the form of heat, thereby elevating the local temperature of catalytic sites and enhancing their catalytic effect.”

[Manuscript - Page 16]

Discussion: “...More importantly, in comparison with MgH₂, the hydrogen storage reaction of Mg₂Ni(Cu) is thermodynamically and kinetically favored resulting from the weakening of Mg-H bonds and the low migration barrier of hydrogen atoms in both Mg₂Ni(Cu) and Mg₂Ni(Cu)H₄, which in turn provides a facile pathway for the spontaneous breaking of Mg-H bonds of MgH₂.

The uneven distribution of the light-induced hot electrons within CuNi and Mg₂Ni(Cu) alloy also contributes to the effective weakening of Mg-H bonds of MgH₂ and the accelerated formation of Mg₂Ni(Cu)/Mg₂Ni(Cu)H₄, hence resulting in a light-enhanced “hydrogen pump” effect in improving cycling hydrogenation and dehydrogenation of MgH₂ and the significant decrease of the apparent energy required for driving hydrogen storage of MgH₂....”

[Manuscript - Page 15]

“...X-ray photoelectron spectroscopy (XPS) results were obtained on a Thermo Scientific K-Alpha system equipped with a dual X-ray source, adopting an Al K α (1486.6 eV) anode with a hemispherical energy analyzer. *In-situ* irradiated XPS (ISI-XPS) were performed on a Thermofisher ESCALAB 250Xi under Xe lamp irradiation....”

[Manuscript - Page 19]

“...A 7x7x7 Gamma centered k-point meshes is used for the dos calculation of Cu, MgH₂, Mg₂NiH₄, Mg₂Ni(Cu)H₄, and 10x10x10 is used for the dos calculation of CuNi. For all slab structures, a 20 Å vacuum layer in the z-direction is added to prevent surface interactions between the upper and lower surfaces. In energy band calculation, the number of k-points is uniformly 20, and the k-path and energy band information are exported by the VASP post-processing tool VASPKIT. The Gaussian smearing method was used with a width of 0.05 eV, and spin polarization was considered in all calculations.... ”

[Supporting Information - Page 30]

Fig. S37. High-resolution a) Cu 2*p* and b) Ni 2*p* XPS spectra of Cu₁Ni₁ in the dark and under solar irradiation of 10 and 20 minutes. High-resolution c) Mg 1*s*, d) Cu 2*p*, and e) Ni 2*p* XPS spectra of Mg₂Ni(Cu), the dehydrogenation state of MgH₂ catalyzed by 30% Cu₁Ni₁ alloy, in the dark and under solar irradiation.

After the *in-situ* formation of Mg₂Ni(Cu) alloy during the initial dehydrogenation, the relative transformation of hot electrons between Cu and Ni persists under light irradiation. However, no observable changes in the binding energy of Mg atoms are observed, attributed to the lower work function of Mg (3.68 eV) compared to Cu (Fig. S37c-e).

[Supporting Information - Page 31]

Fig. S38. Crystal model and selected plane of a) CuNi and c) Mg₂Ni(Cu) alloy. Distribution of atoms and corresponding differential charge densities of b) CuNi alloy and d) Mg₂Ni(Cu) alloy. All the results are plotted with an isovalue of 0.001 e/Bohr³.

[Supporting Information - Page 32]

Fig. S39. The charge density differences of MgH₂ under the catalysis of a) Ni, b) Cu, c) CuNi and d) Mg₂Ni(Cu). The blue area represents electron migration, while the yellow area represents electron aggregation, respectively. The length of Mg-H bond under the catalysis of d) Ni, e) Cu, f) CuNi and g) Mg₂Ni(Cu).

2. About the role of light in this work. The author explains that the better photothermal performance than pure thermal performance is a localized thermal effect. We believe that such an explanation is insufficient. Local thermal effects are essentially thermal catalysis, and their functions are the same except for different heating methods. In addition, the activation energy (Figure S14f) under photothermal and pure thermal conditions is the same, which further explains the light only provides a thermal effect during the reaction process. However, why the performance is better than under pure thermal conditions. A detailed explanation was needed.

Response: Thanks for the invaluable comment. In the response to the Comment 1, employing *in-situ* irradiated XPS and DFT calculations, we elucidate the uneven

distribution of the light-induced hot electrons transferring inside of CuNi and Mg₂Ni(Cu) alloys, which contributes to the effective weakening of the Mg-H bonds of MgH₂ and enhances “hydrogen pump” effect of Mg₂Ni(Cu)/Mg₂Ni(Cu)H₄ during the dehydrogenation cycling. This is another important factor contributing to the better photothermal performance than pure thermal performance besides localized thermal effect (please refer to our detailed response to the Comment 1).

As for activation energy, the comparable results under photothermal and pure thermal conditions suggests that, the LSPR-induced hot electrons couldn't participate in the hydrogen release process of MgH₂, and therefore, they do not play a role in lowering the reaction energy barrier for MgH₂ dehydrogenation. However, these hot electrons that fail to injected into the MgH₂ dehydrogenation process are converted into localized heat by thermalization for initiating catalytic effects of CuNi and Mg₂Ni(Cu) alloys.

However, it is noteworthy that, upon light irradiation, the pre-exponential factor for the activation energy is increased compared to that driven by thermal heating (Fig.S14f). This aligns with the conclusion of previous studies in thermal catalysis, *i.e.*, when two samples exhibit almost the same apparent activation energy, the difference in the reaction rates is attributed to the variations in the number of active sites, that is, the different pre-exponential factors in the Arrhenius equation ^{7, 8}. This also corroborates the viewpoint stated in our response to Comment 1 that, although the light-induced hot electrons couldn't be directly transferred into the MgH₂, they play an additional role in realizing the formation of electron-rich Ni sites, and further promoting the dissociation of Mg-H bonds of MgH₂ during H₂ desorption.

Therefore, we believe that, the better photothermal performance than pure thermal performance is attributed to both the light-enhanced “hydrogen pump” effect with the effective weakening of Mg-H bonds of MgH₂ owing to the non-uniform electron distribution of CuNi alloys under light irradiation and the localized thermal energy generated by the unreleased hot electrons that could directly act on catalytic sites.

We appreciate the reviewer's comment and all the related supplement with discussion (highlighted in red color) have been provided in the revised manuscript as follows:

[Manuscript - Page 2]

[Manuscript - Page 30]

Fig. 5. a) Schematic illustration of H₂ desorption and H diffusion pathway of MgH₂ on the surface of Mg₂Ni(Cu) alloy and b) the corresponding energy profiles. c) Schematic diagram of the ideal integration of excellent photothermal and catalytic effect via continuous *in-situ* atomic reconstruction upon repeated dehydrogenation process.

3. About temperature testing under light irradiation. (Figure S11) At present, the use of

infrared thermometers in the field of photothermal catalysis is not accepted. The author should use a thermocouple to test the temperature of the catalyst surface in contact with the catalyst. In the article, the reason why the performance of photothermal catalysis is better than pure thermal performance is due to inaccurate temperature testing of the catalyst surface under photothermal conditions. As far as we know, using an infrared thermometer to test the temperature is 20-50 °C lower than the actual temperature. Therefore, the surface temperature of the catalyst needs to be remeasured, and the comparison experiment between photothermal and pure thermal needs to be redone. (Figure 2c, 2g, S14b S14e,f, S16a,c and d)

Response: We sincerely appreciate the reviewer for providing this valuable question. In order to compare the differences between the infrared thermometer and the contact-type thermocouple and their impact on the comparison experiments between photothermal and pure thermal conditions, a home-made glass tube reactor (Fig. S11c-d) that allows the direct insertion of a contact-type thermocouple is employed for hydrogen release tests without high pressure. We remeasured a series of surface temperature and dehydrogenation curves of MgH₂ under the catalysis of Cu₁Ni₁ using various light intensities, and reevaluated the comparison between photothermal and pure thermal, including the completeness ratio over time and the activation energy. Simultaneously, we supplemented temperature and dehydrogenation curves data for MgH₂ catalyzed by Cu, Ni, and Cu₁Ni₃ as a further validation.

It is well-known that precise temperature detection in photothermal catalysis is still a big challenge^{9, 10}. Two common approaches for detecting reaction temperature in photothermal reactions are the contact-type thermocouple based on the Seebeck effect¹¹ and the non-contact-type infrared thermometer based on the blackbody radiation law¹². Consistent with the reviewer's comments, our experimental results also indicate that temperatures measured by contact-type thermocouples are indeed slightly lower than those obtained by infrared thermometer. The disparity between the two testing methods for all samples in our experiment falls within the range of 5~10 °C (Fig. S15a-c, Fig. S16a-c). The contact-type thermocouple takes approximately 10 minutes to reach a stable temperature, while the infrared thermometer responds more rapidly, which may contribute to the difference in measurement principle and sensitivity to some extent.

Taking MgH₂ under the catalysis of Cu₁Ni₁ as an example, the dehydrogenation performance and temperature differences are compared using two temperature measurement methods under the light intensity of 2.5~2.8 W/cm² (Fig. S15a-f). The dehydrogenation rates of powder samples in the glass tube under varying light intensities are slower compared to the sample pellets in the high-pressure reactor. However, whether in the glass tube or the high-pressure reactor, the dehydrogenation

rate of MgH_2 catalyzed by Cu_1Ni_1 alloy under solar irradiation is faster and exhibits a more rapid response compared to heating (Fig. S15g). Additionally, we further fitted the activation energy under light irradiation using the glass tube with contact-type thermocouple, resulting in a value of 80.4 ± 5.57 kJ/mol, which remains comparable to that obtained in the high-pressure reactor (81.23 ± 3.35 kJ/mol) (Fig. S15h). Furthermore, we retested the temperature and dehydrogenation performance of MgH_2 catalyzed by Cu, Ni, and Cu_1Ni_3 under the light intensity of 3.0 W/cm^2 and compared the results with the corresponding heating performance. All of these experiments led to similar conclusions as observed for MgH_2 under the catalysis of Cu_1Ni_1 (Fig. S16a-f). These results further validate the advantages of solar-driven dehydrogenation measured by two methods over heat-driven H_2 desorption in realizing fast dehydrogenation for MgH_2 under the catalysis of CuNi alloy.

It is important to note that, due to the stringent requirements for reactor airtightness and pressure resistance in testing hydrogen storage performance of MgH_2 , despite our initial efforts to improve the reactor design earlier, a satisfactory coupling between the contact-type thermocouple and the high-pressure reactor has not been achieved at present. Therefore, differences in thermal management, sample form, and other aspects still persist between the high-pressure reactor with an infrared thermometer and the glass tube with contact-type thermocouple testing methods. For example, in terms of sample form, about 30 mg samples are pressed into pieces to ensure the stability of the test and the comparability of the results in high-pressure reactor, while samples are placed in the reactor with the same mass as powder to ensure that the thermocouple is fully buried. Due to the accelerated dehydrogenation rate of MgH_2 under light irradiation, powder samples in the glass tube experience uncontrollable splattering. In addition, regarding thermal management, glass fiber sheets with low thermal conductivity are placed at the bottom inside of the high-pressure reactor to minimize material heat dissipation to the outside. In contrast, the glass tube can only mitigate heat loss through an external shield filled with thermal insulation cotton. The difference of contact area (powder vs. pellet) with light between the samples, as well as the overall thermal management of the reactor, leads to the slower dehydrogenation rates of powder samples in the glass tube compared to the sample pellets in the high-pressure reactor. Additionally, powder splattering results in a portion of the powder not being maintained within a more uniformly illuminated range (Fig. S11c-2), which also could contribute to the decrease in H_2 capacity.

More importantly, the cycling performance is crucial for the practical application of MgH_2 . In our original manuscript, we demonstrated the enhancement of photothermal performance and its catalytic impact due to the *in-situ* formation of $\text{Mg}_2\text{Ni}(\text{Cu})$ ternary alloy after cycling. Unfortunately, the testing method using a glass tube cannot

withstand high hydrogenation pressures, making it impractical for stable cycling tests. Therefore, considering the stringent requirements for airtightness, stability, and accessibility in MgH₂ hydrogen storage system, the use of a high-pressure reactor with real-time temperature measurement by an infrared thermometer may be the most suitable and reliable approach for testing in the current stage of this work. Therefore, the comparative temperature data from the glass tube measurements mentioned above are provided in the Supporting Information for reference to the readers. We will consistently make efforts to enhance the integration of reactors and temperature measurement devices and actively upgrade our testing equipment to achieve more accurate temperature measurements.

We appreciate the reviewer's comment, which have enhanced the rigor of the manuscript, and all the above-mentioned results and the related discussion (highlighted in red color) have been provided in the revised supporting information as follows:

[Manuscript - Page 8]

“...Particularly, the superiority of fast response and rate of H₂ desorption under solar irradiation over direct heating could be consistently observed for MgH₂ within the range of 2.5~3.0 W/cm² irradiance (Fig. 2g, Fig. S15g). In strong contrast, H₂ desorption kinetics of MgH₂ using direct heating strongly depends on adopted temperatures induced by the low thermal conductivity of MgH₂ (< 1 W/m K)¹³. The advantages of solar-driven dehydrogenation over heat-driven H₂ desorption in realizing fast dehydrogenation are also measured and evaluated using the glass tube with contact-type thermocouple method (Fig. S11c-d, Fig. S15a-f), and demonstrated in MgH₂ catalyzed by Cu₁Ni₃, Ni, and Cu, respectively (Fig. S16a-f).”

[Supporting Information - Page 13]

Fig. S11. The photograph of solar-driven H₂ absorption and desorption apparatus and schematic diagram of the customized reactor: a-b) high-pressure reactor with infrared thermometer used in this work, and c-d) glass-tube reactor with contact-type thermocouple used for temperature validation.

For temperature testing, the surface temperature during the H₂ absorption and desorption processes of the sample is continuously monitored in real-time *via* a short-wave infrared thermometer, which could receive the signal of infrared radiation from the sample's surface through the sapphire, and subsequently converts it into temperature data. **Additionally, to minimize the influence of the unavoidable non-uniformity of photon flux of Xe lamp on temperature accuracy, we have adjusted the height of the infrared thermometer to nearly cover the entire sample range and ensure that the temperature measurement center aligns with the light source center, which provides a more accurate measurement of the sample surface's average temperature.**

Another detailed description for the hydrogen storage measurement and sample preparation of **hydrogen-pressure reactor** is provided in the "Hydrogen Storage Measurement" section of Method.

To compare the differences between the infrared thermometer and the contact-type thermocouple, a home-made glass tube reactor (Fig. S11c-d) that allows the direct insertion of a contact-type thermocouple is employed for hydrogen release tests without high pressure. Approximately 30 mg samples are placed in the reactor as powder to ensure that the thermocouple is fully covered by the materials to accurately measure the temperature of MgH₂ composite. During the testing process, a shield filled with thermal insulation cotton is used to

minimize heat loss through thermal conduction from glass tube to the outside (Fig. S11c-1).

[Supporting Information - Page 17]

Fig. S15. a-c) Temperature and d-f) corresponding dehydrogenation curves of MgH₂ catalyzed by Cu₁Ni₁ via two different measurement method under varying light intensity, in comparison of that driven by heating.

The difference of contact area (powder vs. pellet) with light between the samples, as well as the overall thermal management of the reactor, leads to the slower dehydrogenation rates of powder samples in the glass tube compared to the sample pellets in the high-pressure reactor. Additionally, powder splattering results in a portion of the powder not being maintained within a more uniformly illuminated range (Fig. S11c-2), which also could contribute to the decrease in H₂ capacity.

The disparity of surface temperature between the two testing methods for all samples in our experiment falls within the range of 5~10 °C. The contact-type thermocouple takes approximately 10 minutes to reach a stable temperature, while the infrared thermometer responds more rapidly, which may contribute to the difference in measurement principle and sensitivity to some extent.

[Supporting Information - Page 18]

Fig. S16. Temperature and corresponding dehydrogenation curves of MgH_2 catalyzed by a,e) Cu, b,d) Ni, and c,f) Cu_1Ni_3 via two different measurement method under varying light intensity, in comparison of that driven by heating.

4. In Figure 1, the evidence is insufficient and needs to be further proved by line scanning two adjacent particles.

Response: We would appreciate the reviewer's careful review of our manuscript. In response to this comment, we have reconduted TEM measurements with line scanning of two adjacent Cu_1Ni_1 alloy particles. The figures have been replaced in Fig. 1h-j and supplemented in Fig. S3h-j as evidences to reinforce the assertion that CuNi is an alloy structure

We appreciate the reviewer's comment and the above-mentioned figures have been provided in the revised supporting information as follows:

[Manuscript - Page 6]

"Furthermore, energy dispersive X-ray spectroscopy (EDS) elemental mapping and line scanning of two adjacent particles (Fig. 1h-j, Fig. S3e-j) result confirms the homogeneous distribution of Cu and Ni, which coincides well with the morphology of particles, further validating the uniform synthesis of CuNi alloys."

[Manuscript - Page 26]

Fig. 1. a) XRD patterns of the as-synthesized of CuNi alloys, including Ni and Cu for comparison. High-resolution b) Cu 2p and c) Ni 2p XPS spectrum of Cu₁Ni₁, with Cu and Ni for comparison. d) SEM and e) HRTEM image and f) SAED pattern of Cu₁Ni₁. g) the corresponding particle size distribution. h-j) EDS elemental mapping images with line scanning of Cu₁Ni₁ alloys.

[Supporting Information - Page 4]

Fig. S1. a-d) TEM images of Cu₁Ni₁ alloys. e-j) EDS elemental mapping images with line scanning of Cu₁Ni₁ alloys.

Response to Reviewer #3

Reviewer #3: This paper describes reversible hydrogen storage on the Mg₂Ni(Cu)H₄ under solar light irradiation. Based on the reviewer's comments, the authors added many data and discussed the data soundly. All of the listed problems were solved. Thus, it seems that this paper can be recommended for publication in Nature Communications without further revision.

Response: We thank the reviewer for the positive comment.

References

1. Shao T, *et al.* A Stacked Plasmonic Metamaterial with Strong Localized Electric Field Enables Highly Efficient Broadband Light - Driven CO₂ Hydrogenation. *Advanced Materials* **34**, 2202367 (2022).
2. Li X, *et al.* PdCu nanoalloy decorated photocatalysts for efficient and selective oxidative coupling of methane in flow reactors. *Nature Communications* **14**, 6343 (2023).
3. Zhang S, *et al.* Visible-light-driven multichannel regulation of local electron density to accelerate activation of O–H and B–H bonds for ammonia borane hydrolysis. *ACS Catalysis* **10**, 14903-14915 (2020).
4. Zhang J, Chen H, Duan X, Sun H, Wang S. Photothermal catalysis: From fundamentals to practical applications. *Materials Today* **68**, 234-253 (2023).
5. Yu W, *et al.* Laser-controlled tandem catalytic sites of CuNi alloy with ampere-level electrocatalytic nitrate to ammonia activity for Zn–nitrate battery. *Energy & Environmental Science*, (2023).
6. Han S, *et al.* Ultralow overpotential nitrate reduction to ammonia via a three-step relay mechanism. *Nature Catalysis*, 1-13 (2023).
7. Xie X, Li Y, Liu Z-Q, Haruta M, Shen W. Low-temperature oxidation of CO catalysed by Co₃O₄ nanorods. *Nature* **458**, 746-749 (2009).
8. Jiang H, *et al.* Light-driven CO₂ methanation over Au-grafted Ce_{0.95}Ru_{0.05}O₂ solid-solution catalysts with activities approaching the thermodynamic limit. *Nature Catalysis* **6**, 1-12 (2023).
9. Mascaretti L, Schirato A, Montini T, Alabastri A, Naldoni A, Fornasiero P. Challenges in temperature measurements in gas-phase photothermal catalysis. *Joule* **6**, 1727-1732 (2022).
10. Bian X, Zhao Y, Zhou C, Zhang T. Minimizing Temperature Bias through Reliable Temperature Determination in Gas-Solid Photothermal Catalytic Reactions. *Angewandte Chemie International Edition*, e202219340 (2023).
11. Ghuman KK, Wood TE, Hoch LB, Mims CA, Ozin GA, Singh CV. Illuminating CO₂ reduction on frustrated Lewis pair surfaces: investigating the role of surface hydroxides and oxygen vacancies on nanocrystalline In₂O_{3-x}(OH)_y. *Physical Chemistry Chemical Physics* **17**, 14623-14635 (2015).
12. Qi Y, *et al.* Fabrication of black In₂O₃ with dense oxygen vacancy through dual functional carbon doping for enhancing photothermal CO₂ hydrogenation. *Advanced Functional Materials* **31**, 2100908 (2021).
13. Xia G, *et al.* Monodisperse magnesium hydride nanoparticles uniformly self - assembled on graphene. *Advanced Materials* **27**, 5981-5988 (2015).

REVIEWERS' COMMENTS

Reviewer #2 (Remarks to the Author):

The authors have well addressed all my questions. All the listed problems were solved. Thus, it seems that this paper can be recommended for publication in Nature Communications without further revision.